# BIN1 knockdown rescues systolic dysfunction in aging male mouse hearts

Maartje Westhoff[1,5], Silvia G. del Villar[1,5], Taylor L. Voelker[1], Phung N. Thai [2], Heather C. Spooner [1], Alexandre D. Costa [1], Padmini Sirish [2], Nipavan Chiamvimonvat[2,3,4], Eamonn J. Dickson [1] & Rose E. Dixon [1] ✉

Cardiac dysfunction is a hallmark of aging in humans and mice. Here we report that a two-week treatment to restore youthful Bridging Integrator 1 (BIN1) levels in the hearts of 24-month-old mice rejuvenates cardiac function and substantially reverses the aging phenotype. Our data indicate that age-associated overexpression of BIN1 occurs alongside dysregulated endosomal recycling and disrupted trafficking of cardiac Ca$_V$1.2 and type 2 ryanodine receptors. These deficiencies affect channel function at rest and their upregulation during acute stress. In vivo echocardiography reveals reduced systolic function in old mice. BIN1 knockdown using an adeno-associated virus serotype 9 packaged shRNA-mBIN1 restores the nanoscale distribution and clustering plasticity of ryanodine receptors and recovers Ca$^{2+}$ transient amplitudes and cardiac systolic function toward youthful levels. Enhanced systolic function correlates with increased phosphorylation of the myofilament protein cardiac myosin binding protein-C. These results reveal BIN1 knockdown as a novel therapeutic strategy to rejuvenate the aging myocardium.

Old age is a major independent risk factor for cardiovascular disease which remains the leading cause of death globally[1,2]. As humans advance into old age, regardless of overall health, certain intrinsic and progressive changes in cardiac structure and function occur including enhanced atrial and ventricular fibrosis, left ventricular hypertrophy, and diastolic dysfunction[3-5]. Maximal systolic function is also impaired due to reduced responsiveness to sympathetic nervous system activation and subsequent β-adrenergic receptor (β-AR)[3-5]. The decline of this reactive stimulus can precipitate exercise intolerance in the elderly and render the heart more vulnerable to damage during acutely stressful events when hypoxia and metabolic stress ensue[4,6]. In mice, similar intrinsic cardiac changes occur over a much shorter lifespan and in the absence of other risk factors like smoking or diabetes, making mice a useful model to understand the molecular mechanisms of cardiac aging[3].

The force of myocardial contraction and the rate and extent of the subsequent relaxation are tuned by both Ca$^{2+}$ and myofilament-dependent processes. Examination and comparison of these points of regulation and their integration in the in vivo function of young and old mice may identify targets to improve the cardiac aging phenotype. Weakened myocardial contractile responses to β-AR stimulation have been linked to an age-associated decrease in the ability of the β-AR-signaling cascade to augment intracellular Ca$^{2+}$ transient amplitude[7,8]. This has been attributed to a reduced capacity to increase Ca$_V$1.2 activity and availability, and thus to diminished Ca$^{2+}$ influx during the action potential (AP)[5]. Activation of cAMP-dependent protein kinase A (PKA), downstream of β-AR stimulation, leads to phosphorylation of the channel complex[9,10], removing an inhibitory brake[11,12] and resulting in increased channel open probability ($P_o$) due to potentiation of longer-duration 'mode

[1]Department of Physiology and Membrane Biology, University of California Davis, Davis, CA, USA. [2]Division of Cardiovascular Medicine, Department of Internal Medicine, University of California, Davis, Davis, CA, USA. [3]Department of Veterans Affairs, Northern California Health Care System, Mather, CA, USA. [4]Department of Pharmacology, University of California Davis, Davis, CA, USA. [5]These authors contributed equally: Maartje Westhoff, Silvia G. del Villar. ✉e-mail: redickson@ucdavis.edu

2' openings[13], and an increased number of functional channels[14,15]. We recently reported that $\beta$-AR activation also mobilizes a sub-sarcolemmal pool of $Ca_V1.2$ channels, triggering a PKA-dependent, dynamic increase in sarcolemmal $Ca_V1.2$ expression in ventricular myocytes[15,16]. Consequent enlargement of $Ca_V1.2$ channel clusters facilitates cooperative channel interactions[16,17], and contributes to an amplification of $Ca^{2+}$ influx that rapidly tunes excitation-contraction (EC)-coupling to meet increased hemodynamic and metabolic demands. We hypothesized that loss of this rheostatic mechanism in aging myocytes could impose a narrow inotropic dynamic range on the system, limiting the response to $\beta$-AR stimulation.

Nanoscale redistribution and augmentation of cardiac $Ca^{2+}$ channel clustering in response to $\beta$-AR stimulation also extends to RyR2. Accordingly, acute treatment with the $\beta$-AR-agonist iso-proterenol (ISO) or a phosphorylation-inducing cocktail promotes enhanced RyR2 cluster sizes[18,19], as recently reviewed[20]. This dynamic redistribution of RyR2 into larger clusters is orchestrated by bridging integrator 1 (BIN1)[18] and is linked to increased $Ca^{2+}$ spark frequency which summate to produce larger $Ca^{2+}$ transients[19,21]. There is widespread agreement that $\beta$-AR-stimulated $Ca^{2+}$ transient augmentation is impaired in the aging myocardium[5,8], however the underlying mechanisms are not understood.

Here, we present a comprehensive examination of cardiac $Ca^{2+}$ channel function, their nanoscale distribution, and ISO-stimulated redistribution to test the central hypothesis that age-dependent disruption in the basal and on-demand trafficking and clustering of these channels restricts EC-coupling plasticity during myocardial aging and contributes to the pathophysiology of cardiac aging. We report that old myocytes exhibit basal super-clustering of both $Ca_V1.2$ and RyR2, with no additional dynamic response to ISO. We link these deficits in cardiac $Ca^{2+}$ channel recycling and mobility to an age-associated upregulation in BIN1 and the development of endo-somal trafficking deficits or "endosomal traffic jams". Crucially, we find that shRNA-mediated knockdown of BIN1 in old animals restores a young phenotype by re-establishing RyR2 organization, $Ca^{2+}$ transient amplitude, and their $\beta$-AR-stimulated augmentation, and reju-venating myofilament protein phosphorylation levels to recover youthful systolic function.

## Results

### $\beta$-AR-stimulated $I_{Ca}$ augmentation is diminished in aging

We began our study with an investigation into the $Ca_V1.2$ channel response to $\beta$-AR stimulation by recording whole-cell $Ca^{2+}$ currents ($I_{Ca}$) from 3-month-old and 24-month-old (henceforth referred to as young and old, respectively) ventricular myocytes under control and 100 nM ISO-stimulated conditions. In young myocytes, ISO elicited a 1.57-fold increase in peak $I_{Ca}$ density (Fig. 1a, b, Supplementary Fig. 1a, c, d, and Supplementary Table 1). However, old myocytes exhibited a blunted response displaying only a 1.19-fold increase with ISO and significantly larger control currents compared to young myocytes (Fig. 1a–d; Supplementary Fig. 2). Control $I_{Ca}$ in old myocytes

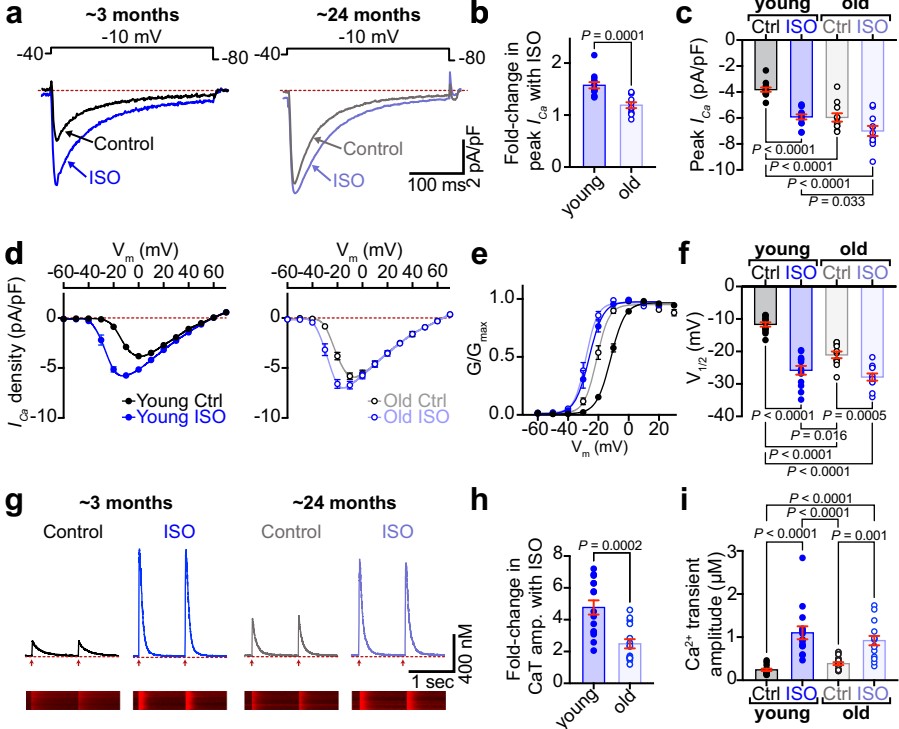

**Fig. 1 | $\beta$-AR-stimulated augmentation of $I_{Ca}$ and $Ca^{2+}$ transients is diminished in aging. a** Representative whole-cell currents elicited from young and old ven-tricular myocytes before (control; black) and during application of ISO (blue). **b** Dot-plots showing the fold-change in peak current with ISO in young and old myocytes. **c** Dot-plots showing peak $I_{Ca}$ density before and after ISO. **d** Plots showing the voltage dependence of $I_{Ca}$ density for both groups before and after ISO. **e** Voltage dependence of the normalized conductance ($G/G_{max}$) fit with Boltzmann functions and **f** dot-plots showing the $V_{1/2}$ of activation for each group. N-numbers for patch clamp data in (**b–f**) is as follows: young ($N = 9$, $n = 13$) and old ($N = 5$, $n = 11$). **g** Representative $Ca^{2+}$ transients recorded before (black) and after ISO (blue) from paced young and old myocytes. **h** Dot-plots showing the fold increase in $Ca^{2+}$ transient amplitude after ISO and **i** $Ca^{2+}$ transient amplitude before and after ISO. N-numbers for $Ca^{2+}$ transient data in (**h, i**) is as follows: young ($N = 5$, $n = 15$) and old ($N = 3$, $n = 15$). Unpaired two-tailed Student's $t$-tests were performed on data sets displayed in (**b, h**). Two-way ANOVAs with multiple comparison post-hoc tests were performed on data displayed in (**c, f, i**). Young data in (**b–f**) is pooled from NIA young and JAX young myocytes, as no significant differences were found in $I_{Ca}$ when NIA young and JAX young myocytes were compared (see Supplementary Fig. 1a–e). Young data in (**h, i**) is from NIA young. Data are presented as mean ± SEM. Source data are provided in the Source Data file.

exhibited a significantly left-shifted voltage dependence relative to that of young myocytes (Fig. 1e, f). This characteristic of PKA-regulation of cardiac Ca$_V$1.2 channels suggests enhanced basal phosphorylation of Ca$_V$1.2 in old myocytes. If a larger proportion of the channels are phosphorylated at rest, this could leave less unspent functional reserve to disburse during ISO stimulation. Accordingly, we observed reduced ISO-stimulated current augmentation indicative of reduced $\beta$-adrenergic responsiveness in old mice versus young.

### Reduced capacity of $\beta$-AR stimulation to tune EC-coupling in aging myocytes

Influx of Ca$^{2+}$ through Ca$_V$1.2 channels during AP-mediated depolarization stimulates Ca$^{2+}$-induced Ca$^{2+}$-release from cross-dyad RyR2 to initiate EC-coupling. The amplitude of the resultant Ca$^{2+}$ transient largely dictates the magnitude of myocardial contraction and the fraction of blood ejected, assuming constant Ca$^{2+}$ sensitivity of the myofilaments. $\beta$-AR stimulation enhances both Ca$_V$1.2 and RyR2-mediated Ca$^{2+}$ influx to produce positive inotropy and a greater ejection fraction. We thus examined Ca$^{2+}$ transients in young and old myocytes to investigate the effects of aging on $\beta$-AR tuning of EC-coupling. Transients evoked from myocytes paced at 1 Hz exhibited a $4.8 \pm 0.4$-fold enhancement in amplitude with ISO in young cells (Fig. 1g, h). In a parallel with the $I_{Ca}$ results, basal Ca$^{2+}$ transients in aging myocytes were larger than in young myocytes (Fig. 1g, i; unpaired Student's $t$-test $P = 0.002$) and the fold increase with ISO was halved ($2.5 \pm 0.3$-fold change; Fig. 1h; Supplementary Fig. 3a, c). Ca$^{2+}$ transient decay rate is an indicator of the effectiveness of Ca$^{2+}$ extrusion mechanisms, mainly SERCA2a, that reinstate Ca$^{2+}$ gradients between beats. PKA-mediated phosphorylation of phospholamban relieves an inhibitory brake on SERCA2a and accelerates Ca$^{2+}$ extrusion and lusitropy during $\beta$-AR stimulation. Accordingly, in young myocytes, ISO significantly accelerated $\tau_{decay}$ (Supplementary Fig. 3b) within 1 min of application however old myocytes exhibited an already enhanced $\tau_{decay}$ that displayed little change with ISO (Supplementary Fig. 3d). These results suggest that old myocytes may exhibit enhanced basal phosphorylation of phospholamban as has been reported by others[22], and further support a reduced $\beta$-adrenergic responsiveness in old hearts and diminished myocardial responses to acute stress.

### Age-dependent alterations in nanoscale distribution and clustering of Ca$_V$1.2 channels

Increased sarcolemmal Ca$_V$1.2 channel expression and clustering at rest in aging cells could explain the age-dependent increase in basal $I_{Ca}$. Furthermore, the reduced responsivity to ISO could be explained by an impaired capacity to mobilize/recycle additional endosomal Ca$_V$1.2 channels in aging myocytes. In young cells, the result of ISO-stimulated channel recycling can be observed as Ca$_V$1.2 super-clustering in single-molecule localization microscopy (SMLM)[15]. To determine whether ISO-stimulated recycling was still functional in aging myocytes, we examined the nanoscale distribution and clustering of Ca$_V$1.2 in t-tubule regions of young and old myocytes using super-resolution SMLM with and without ISO stimulation. These experiments confirmed that the super-clustering response to ISO was intact in the t-tubules of young myocytes, where Ca$_V$1.2 channel cluster areas in ISO-treated (8 min) young myocytes were 23.8% larger than unstimulated controls (Fig. 2a, b). This Ca$_V$1.2 super-clustering response, was rapid and detectable after just 1 min of ISO exposure, with cluster areas plateauing over 3–8 min ISO (Supplementary Fig. 4a, b). In old myocytes, channels were already basally super-clustered, showed no augmentation in area with ISO, and were similarly sized to clusters in ISO-stimulated young cells (Fig. 2a, b). Overall, the age-dependent increase in basal Ca$_V$1.2 channel cluster area and failure to elicit any additional increase with ISO suggests that a larger number of channels are already present in old myocyte t-tubule sarcolemma and that the ISO-stimulated channel insertion response is impaired in aging.

### Age-dependent alterations in nanoscale distribution and clustering of RyR2 channels

Other groups have reported a similar phosphorylation-induced cluster size expansion of the cross-dyad RyR2s in young ventricular myocytes[18,19]. In agreement with those reports, SMLM performed on young myocytes immunostained against RyR2, revealed that average RyR2 cluster size grew by 56.2% in young myocytes after acute ISO treatment (Fig. 2c, d). ISO-stimulated RyR2 cluster expansion in young cells occurred on a similar timescale to the Ca$_V$1.2 super-clustering (Supplementary Fig. 4c, d). Furthermore, mean RyR2 cluster areas in

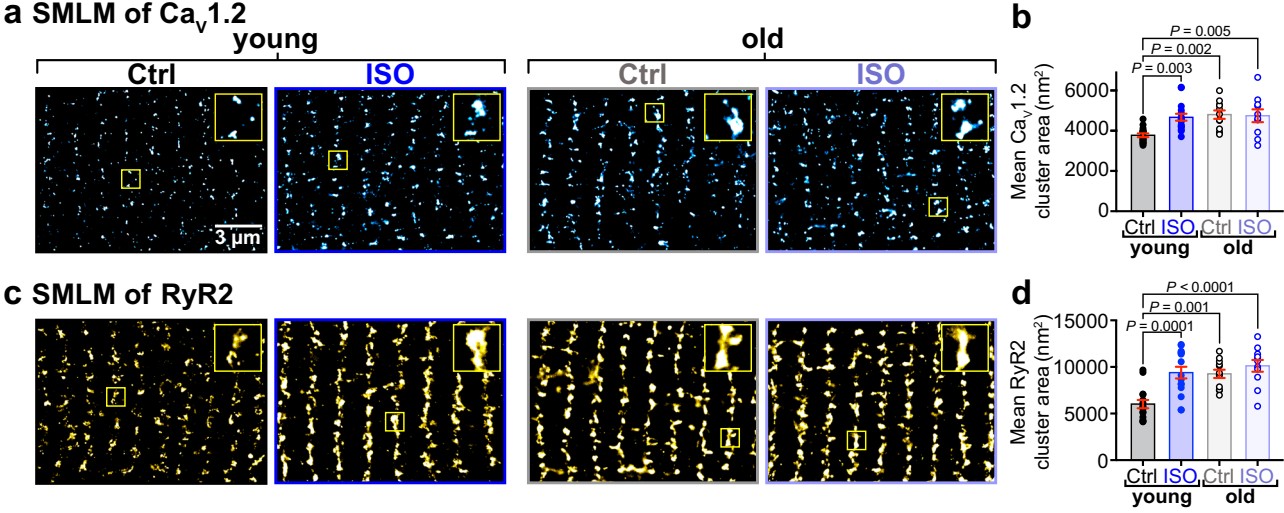

**Fig. 2 | Basal super-clustering and impaired $\beta$-AR responsiveness of Ca$_V$1.2 and RyR2 in aged myocytes. a** Single-molecule localization microscopy (SMLM) map showing Ca$_V$1.2 channel localization and distribution in the t-tubules of young and old ventricular myocytes with or without ISO stimulation. Yellow boxes indicate the location of the regions of interest magnified in the top right of each image. **b** Dot-plots showing mean Ca$_V$1.2 channel cluster areas in young (control: $N = 3$, $n = 16$;

ISO: $N = 3$, $n = 16$) and old (control: $N = 4$, $n = 11$; ISO: $N = 4$, $n = 10$) myocytes. **c, d** show the same for RyR2 in young (control: $N = 3$, $n = 15$; ISO: $N = 3$, $n = 15$) and old (control: $N = 3$, $n = 12$; ISO: $N = 3$, $n = 11$) myocytes. Statistical analyses on data summarized in (**b, d**) were performed using two-way ANOVAs with multiple comparison post-hoc tests. Young data in b and d are from NIA young. Data are presented as mean $\pm$ SEM. Source data are provided in the Source Data file.

unstimulated old myocytes were similarly sized to the expanded clusters observed in ISO-stimulated young myocytes and did not undergo any further expansion with ISO (Fig. 2c, d). These findings suggest that the ISO-stimulated nanoscale redistribution of RyR2 is also impaired in aging.

### Dynamic TIRF imaging reveals reduced ISO-stimulated Ca$_V$1.2 insertion in aging

The basal super-clustering and absence of an additional response to ISO in old myocytes may indicate: (i) the endosomal reservoir is empty after already being recycled and inserted; and/or (ii) the trafficking and mobilization of channels are somehow impaired by aging. To interrogate these hypotheses, we visualized real-time dynamics of the channels utilizing a Ca$_V$1.2 "biosensor" approach pioneered by our lab. In this technique, adeno-associated virus serotype 9 (AAV9)-Ca$_V$β$_{2a}$-paGFP auxiliary subunits are transduced via retro-orbital injections and then visualized in isolated myocytes using TIRF microscopy as previously described[15,16]. The relative balance between insertion and endocytosis dictates the overall expression of any membrane protein. An insertion-heavy mismatch between them will lead to increased membrane expression of the protein, while an endocytosis-heavy mismatch will favor reduced expression over time. We used image math to calculate the relative population of channels that were inserted during the ISO treatment, removed/endocytosed, or that remained static during the whole recording. As expected based on our previous studies[15,16], young myocytes responded to ISO by inserting more Ca$_V$1.2 channels into the sarcolemma than they endocytosed, resulting in a net change in Ca$_V$β$_{2a}$-paGFP in the TIRF footprint that on average amounted to 18% (Fig. 3a, c, e, f). However, old myocytes exhibited a comparatively blunted response with fewer ISO-stimulated insertions (Fig. 3b, c, e), reduced endocytosis (Fig. 3d), and a larger population of static channels in the TIRF footprint suggesting impaired mobility (Fig. 3e; Supplementary Movie 1). Consequently, the ISO-induced change in TIRF footprint Ca$_V$β$_{2a}$-paGFP amounted to only 9.5% (Fig. 3f). Overall, these data reveal an age-dependent deficit in ISO-triggered insertion of Ca$_V$1.2, and a shift towards reduced channel mobility or "stranding" at the plasma membrane.

### Endosomal traffic jams impede mobilization of endosomal Ca$_V$1.2 reservoirs in aging

We further investigated the effects of aging on the endosomal reservoir and capacity to mobilize Ca$_V$1.2 channel cargo in response to ISO by examining the EEA1-positive early endosome pool of Ca$_V$1.2 in young and old myocytes. Accordingly, Airyscan super-resolution imaging on young myocytes revealed immunostained Ca$_V$1.2 on 14.89% of EEA1-positive pixels (Fig. 4a, b). As we have previously reported, β-AR activation stimulates the recycling of a portion of the early endosome Ca$_V$1.2 to the plasma membrane via the Rab4-choreographed fast endosomal recycling pathway[15]. This was evidenced by a significant decrease in the % colocalization between EEA1 and Ca$_V$1.2 after ISO in young myocytes. In old myocytes, there was a trending increase in the percentage of EEA1/Ca$_V$1.2 colocalization in unstimulated cells and no ISO-induced endosomal emptying, suggesting trafficking alterations (Fig. 4b). Supporting this idea, EEA1-positive early endosomes were discernibly and quantifiably larger in old myocytes than in young (Fig. 4a, c).

Transferrin recycling assays further confirmed the significant slowing of endosomal recycling in old cells compared to young (Supplementary Fig. 5).

### BIN1 protein and transcript is overexpressed in old myocytes

Endosomal enlargement has been linked to endosomal dysfunction and deficiencies in membrane protein recycling in Alzheimer's disease (AD)[23]. Endosome swelling results from an imbalance in cargo coming into and leaving endosomes, sometimes referred to as endosomal traffic jams[24]. BIN1/amphiphysin II has been implicated as playing a role in the development of endosomal traffic jams in AD-stricken neurons[24]. In the heart, BIN1 is better known for its role in targeted delivery of Ca$_V$1.2[25,26], t-tubule biogenesis[27], micro-folding[28], and maintenance[29], and in dyad formation[30]. However, in neurons, where there are no

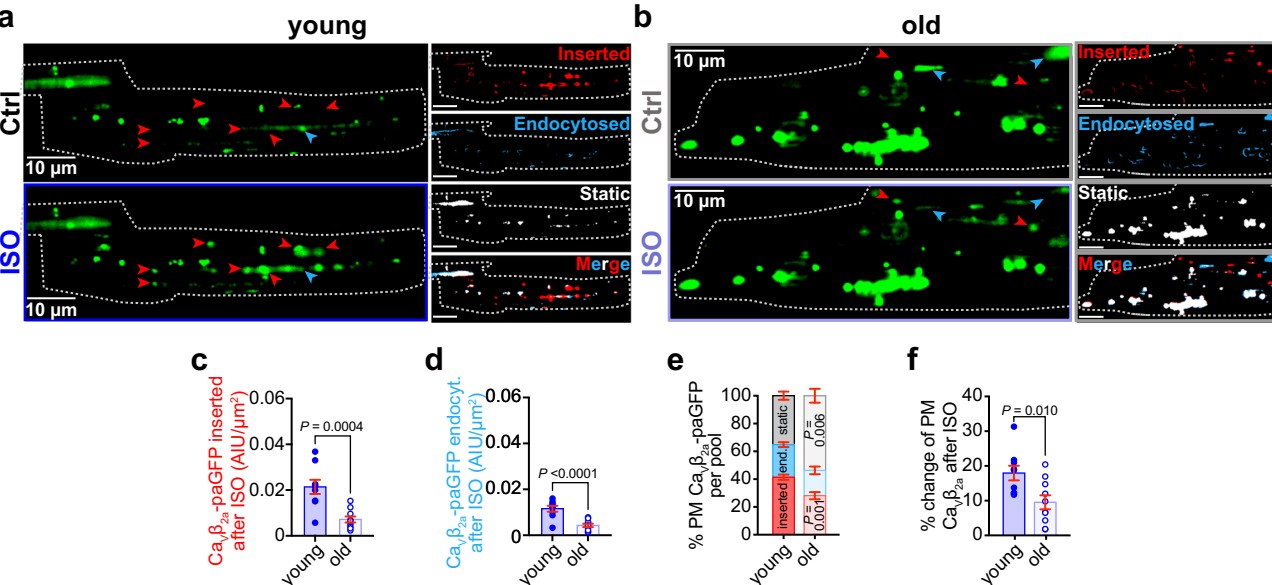

**Fig. 3 | Dynamic TIRF imaging reveals age-associated trafficking deficits of Ca$_V$1.2. a, b** Representative TIRF images of transduced Ca$_V$β$_{2a}$-paGFP young (**a**) and old (**b**) ventricular myocytes before (*top*) and after ISO (*bottom*). Channel populations that were inserted (red), endocytosed (blue), or static (white) during the ISO treatment are represented to the *right*. **c, d** Dot-plots summarizing the quantification of inserted (**c**) and endocytosed (**d**) Ca$_V$β$_{2a}$-paGFP populations. **e** is a compilation of the data to show the relative % of the channel pool that is inserted, endocytosed, and static. **f** is a summary plot showing the net % change of plasma membrane Ca$_V$β$_{2a}$-paGFP after ISO for each cohort of young and old myocytes. N-numbers for data in (**c–f**) are as follows: young ($N = 3$, $n = 9$) and old ($N = 3$, $n = 10$). Statistical analysis was performed on data in (**c–f**) using unpaired two-tailed Student's *t*-tests. Young data in (**c–f**) are from JAX young. Data are presented as mean ± SEM. Source data are provided in the Source Data file.

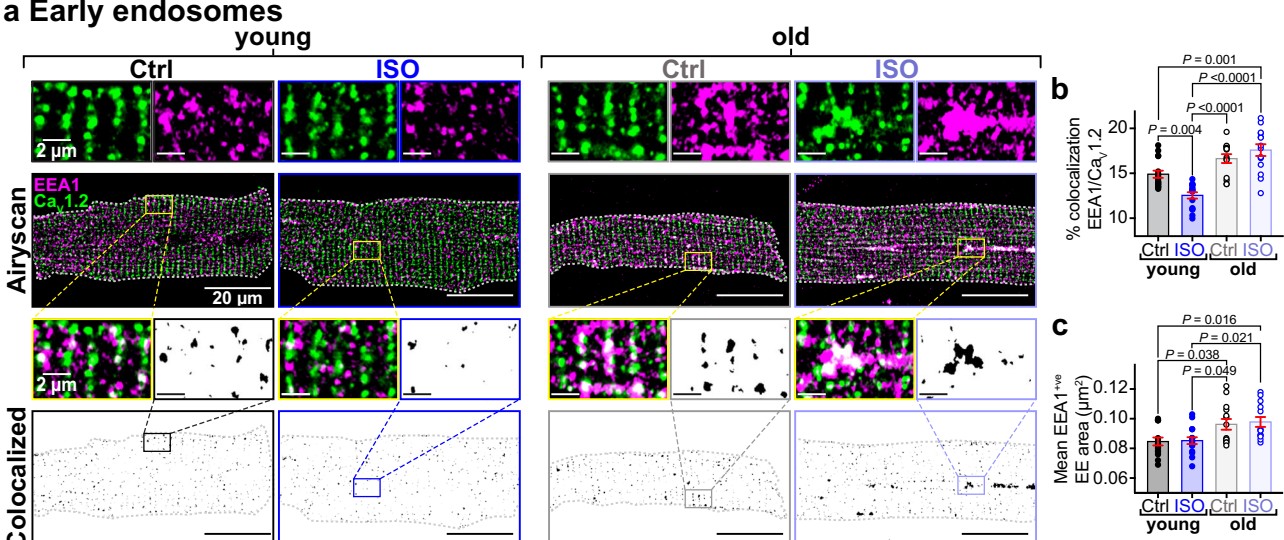

**Fig. 4 | Aging impairs $\beta$-AR-stimulated Ca$_V$1.2 recycling. a** Airyscan super-resolution images of Ca$_V$1.2 (green) and EEA1 (magenta) immunostained myocytes with and without ISO. *Bottom:* Binary colocalization maps show pixels in which Ca$_V$1.2 and EEA1 completely overlapped. **b** dot-plots summarizing % colocalization between EEA1 and Ca$_V$1.2 young (control: $N = 3$, $n = 16$; ISO: $N = 3$, $n = 16$) and old (control: $N = 3$, $n = 14$; ISO: $N = 3$, $n = 14$) myocytes, and **c** EEA1-positive endosome areas in young (control: $N = 3$, $n = 15$; ISO: $N = 3$, $n = 16$) and old (control: $N = 3$, $n = 14$;

ISO: $N = 3$, $n = 13$) myocytes. Data were analyzed using two-way ANOVAs with multiple comparison post-hoc tests. Young data in (**b**, **c**) are from JAX young. Note no significant differences in EEA1/Ca$_V$1.2 colocalization, responsivity to ISO, or endosome size was detected when young JAX and young NIA myocytes were compared (Supplementary Fig. 1h, i). Data are presented as mean ± SEM. Source data are provided in the Source Data file.

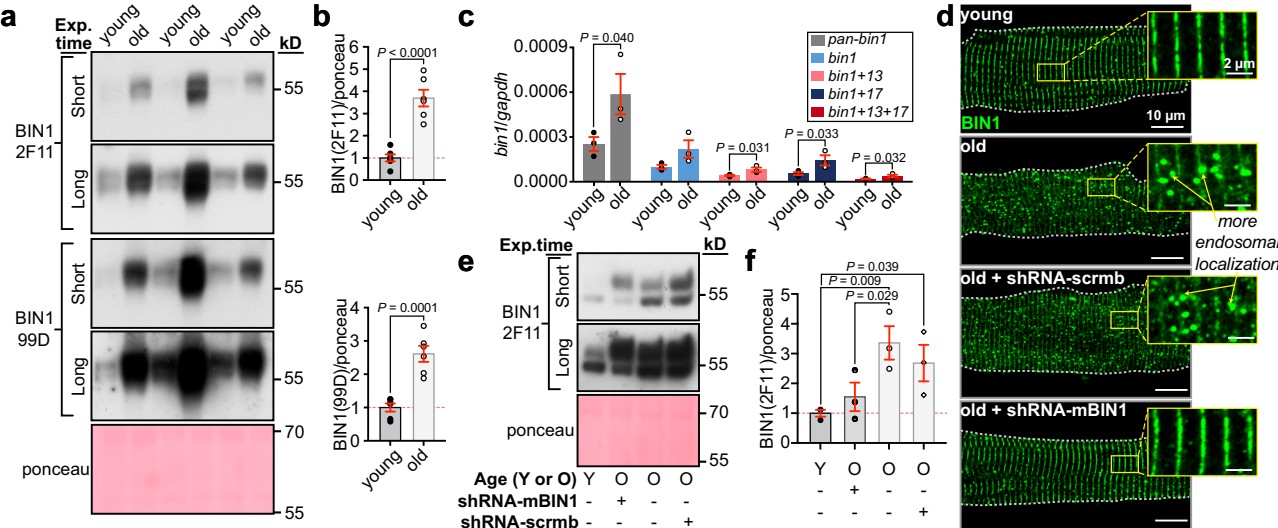

**Fig. 5 | shRNA-mediated BIN1 knockdown restores youthful expression levels and localization patterns. a** Western blot of BIN1 expression in whole heart lysates from young and old mice probed with 2F11 (*top*) and 99D (*bottom*). Short and long exposures are displayed. Total ponceau was used for normalization. **b** Histogram showing normalized BIN1 levels in old relative to young for 2F11 and 99D ($N = 6$ biological replicates). **c** Quantitative RT-PCR analysis of *pan-bin1*, *bin1*, *bin1 + 13*, *bin1 + 17*, and *bin1 + 13 + 17* transcripts for young and old myocytes are displayed normalized to *gapdh* ($N = 3$ per group, samples from each $N$ were ran in triplicate through three separate PCR runs). **d** Representative Airyscan images of young ($N = 3$, $n = 20$), old ($N = 3$, $n = 15$), old shRNA-scrmb ($N = 3$, $n = 16$) and old shRNA-mBIN1 ($N = 3$, $n = 19$) myocytes immunostained against BIN1. **e** Western blot of BIN1 expression in whole heart lysates from: young, old, shRNA-mBIN1, and shRNA-

scrmb transduced mice probed with 2F11. Short and long exposures are displayed. Total ponceau was used for normalization. **f** Histogram showing normalized BIN1 levels relative to young ($N = 3$ biological replicates). Statistical analysis was performed on data in (**b**) using unpaired two-tailed Student's *t*-tests, on data in (**c**) using unpaired one-tailed Student's *t*-tests, and on data in (**f**) using one-way ANOVAs with multiple comparison post-hoc tests. Young data in (**a**, **b**, **e**, **f**) are from JAX young, and young data in (**c**, **d**) are from NIA young myocytes. Note there was no significant difference in BIN1 expression when JAX and NIA young mice were compared (see Supplementary Fig. 1j–l). Data are presented as mean ± SEM. See Supplemental Information for full scans and number of technical replicates for western blots. Source data are provided in the Source Data file.

t-tubules, BIN1 is known to play a role in mediating endosomal membrane curvature and tubule formation required for cargo exit and recycling from endosomes[31,32]. BIN1 overexpression has been linked to aging and neurodegeneration in the brain[33–35] and has been seen to produce endosomal expansion and traffic jams[24,31,32]. Furthermore,

BIN1 expression levels are known to affect ion channel trafficking in both neurons[36] and ventricular myocytes[25,26]. We thus examined BIN1 expression and localization in young and old myocytes and found an age-associated upregulation (Fig. 5a, b). Four BIN1 splice variants are known to be expressed in the hearts of young mice[28,37] (Supplementary

Fig. 6a). We observed each of those variants in young and old hearts. Probing western blots with a pan-BIN1 antibody (2F11) revealed three bands in young and old mice and based on their molecular weight these are believed to represent BIN1 (~48 kDa), a combination of BIN1 + 13 and BIN1 + 17 (~58 kDa), and BIN1 + 13 + 17 (~65 kDa) (Supplementary Fig. 6b). Probing with an exon 17 specific anti-BIN1 (99D) revealed the presence of two exons 17 containing isoforms, surmised to be BIN1 + 17 (~58 kDa) and BIN1 + 13 + 17 (~65 kDa) (Supplementary Fig. 6c). Unfortunately, the relatively high BIN1 expression levels in old lysates made it impossible to capture well-separated bands for young and old mice using our approach and thus the relative isoform expression profile of each cohort could not be defined.

To examine *Bin1* transcriptional expression, total RNA was extracted from ventricular myocytes isolated from young and old mouse hearts and reverse transcribed into cDNA. Quantitative RT-PCR was performed using primers against total *Bin1* transcript, the known cardiac isoforms *Bin1*, *Bin1 + 13*, *Bin1 + 17*, and *Bin1 + 13 + 17*[37], and *gapdh*. Using a pan-*Bin1* primer pair against the ubiquitous exon 2, we observed a $2.32 \pm 0.53$-fold increase in total *Bin1* transcript expression in old cells compared to young (Fig. 5c). Isoform-specific primer pairs confirmed the expression of all four established *Bin1* isoforms and similar fold changes were observed in each of the *Bin1* isoforms with aging (*Bin1*: $2.20 \pm 0.58$; *Bin1 + 13*: $1.88 \pm 0.33$; *Bin1 + 17*: $2.57 \pm 0.60$; *Bin1 + 13 + 17*: $2.03 \pm 0.38$). In both young and old cells mRNA for *Bin1* and *Bin1 + 17* were most abundantly expressed while *Bin1 + 13* and *Bin1 + 13 + 17* transcripts were scarcer. Taken together, these results indicate that BIN1 protein and *Bin1* transcript are upregulated in aging ventricular myocytes compared to young.

## BIN1 knockdown in old mice rejuvenates BIN1 localization, RyR2 plasticity, and β-AR responsiveness

While BIN1 was observed to undergo upregulation with aging, those experiments did not provide any information as to the spatial localization of the excess protein. To examine that we immunostained fixed young and old ventricular myocytes against BIN1. These experiments revealed a mislocalization or redistribution of BIN1 in old myocytes where a vesicular, endosome-like pattern of BIN1-staining was observed alongside the expected z-line population (Fig. 5d; *top*).

Since BIN1 became upregulated and mislocalized with aging with a more endosome-like pattern of expression, we hypothesized that knocking down BIN1 to young levels might resolve the endosomal traffic jams and restore β-AR responsivity to aging hearts. To test this, we transduced live mice with AAV9-GFP-mBIN1-shRNA (shRNA-mBIN1) or AAV9-GFP-scramble-shRNA (shRNA-scrmb; control) via retro-orbital injection. Two weeks post-injection, hearts were harvested, and successful shRNA-mediated knockdown of BIN1 was confirmed via both immunostaining and western blot analysis. BIN1 knockdown restored youthful localization of BIN1 to the t-tubules (Fig. 5d; *bottom*) and reduced BIN1 expression levels in old hearts to young heart levels (Fig. 5e, f). Control shRNA-scrmb transduced old hearts had similar BIN1 expression levels to non-transduced old hearts (Fig. 5e–f) and in all experiments cells isolated from shRNA-scrmb transduced hearts maintained an old phenotype (Figs. 5d, 6a–h, Supplementary Figs. 7 and 8, and Supplementary Table 1).

Given the role of BIN1 in t-tubule development[29,37], we also examined the t-tubule network in isolated young, old, shRNA-scrmb treated old, and shRNA-mBIN1 treated old ventricular myocytes, staining them with the membrane dye di-8-ANEPPS (Supplementary

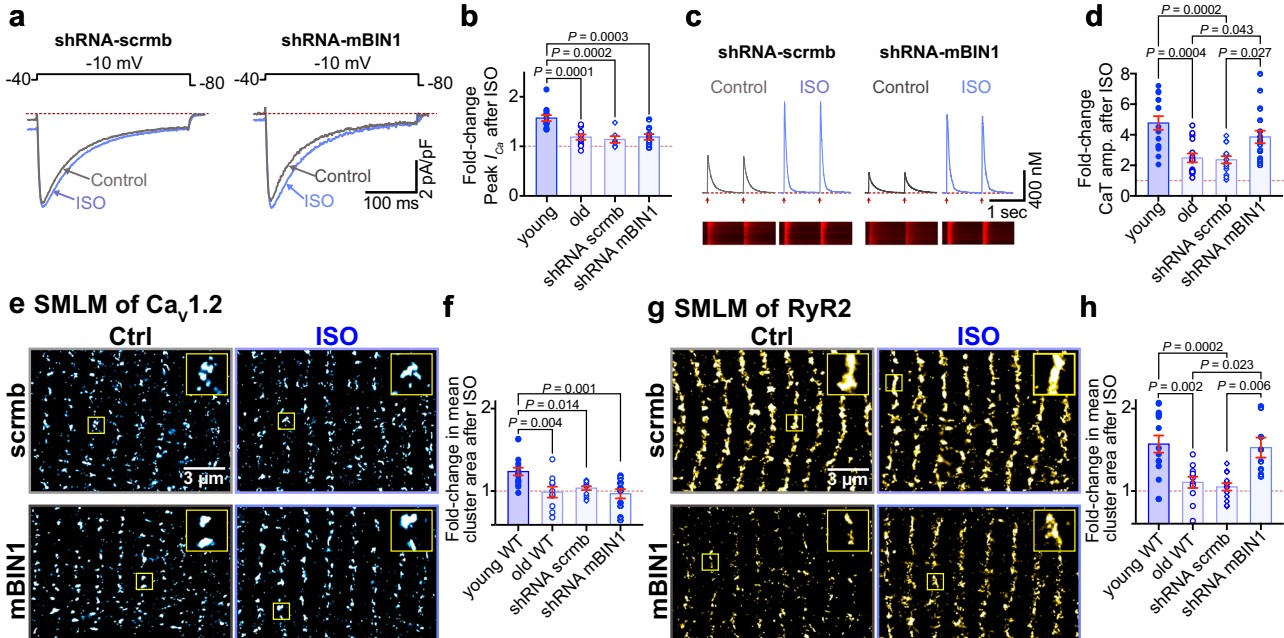

**Fig. 6 | BIN1 knockdown restores β-AR augmentation of Ca²⁺ transients and RyR2 clustering dynamics. a** Representative whole-cell currents from shRNA-scrmb and shRNA-mBIN1 myocytes before and during ISO application. **b** fold change in peak $I_{Ca}$ with ISO in young, old, shRNA-scrmb ($N = 3$, $n = 7$), and shRNA-mBIN1 ($N = 3$, $n = 9$) myocytes. **c** Representative Ca²⁺ transients recorded from old shRNA-scrmb and shRNA-mBIN1 myocytes before and after ISO. **d** Fold increase after ISO from young, old, shRNA-scrmb ($N = 3$, $n = 14$), and shRNA-mBIN1 ($N = 5$, $n = 18$) myocytes. **e** SMLM localization maps showing Ca$_V$1.2 channel localization on t-tubules of myocytes from old shRNA-scrmb and shRNA-mBIN1, with or without ISO stimulation. Regions of interest are highlighted by yellow boxes. **f** Fold change in mean Ca$_V$1.2 channel cluster area with ISO in the young, old, old shRNA-scrmb

(control: $N = 3$, $n = 14$; ISO: $N = 3$, $n = 13$) and shRNA-mBIN1 myocytes (control: $N = 3$, $n = 12$; ISO: $N = 3$, $n = 11$). **g, h** show the same layout for RyR2 immunostained old shRNA-scrmb (control: $N = 3$, $n = 12$; ISO: $N = 3$, $n = 13$) and shRNA-mBIN1 myocytes (control: $N = 3$, $n = 9$; ISO: $N = 3$, $n = 8$). Old and young data points in (**b, d, f, h**) are reproduced from data in Figs. 1b, h, and 2b, d respectively. Statistical analysis was performed on data in (**b, d, f, h**) using one-way ANOVAs with multiple comparison post-hoc tests. Young data in (**b**) are pooled from NIA young and JAX young myocytes, data in (**d, f, h**) are from NIA young myocytes. Note there was no significant difference in $I_{Ca}$, Ca$_V$1.2, and RyR2 cluster areas when JAX and NIA young mice were compared (see Supplementary Fig. 1a–g). Data are presented as mean ± SEM. Source data are provided in the Source Data file.

Fig. 8). T-tubule density and organization decreased with aging with many tubules adopting a longitudinal orientation. Interestingly, myocytes isolated from old mice two weeks post-retro-orbital injection with AAV9-shRNA-mBIN1 displayed youthful t-tubule organization and density with fewer longitudinally arranged tubules. This effect was not observed in shRNA-scrmb transduced old myocytes and thus this rejuvenation of the t-tubule network appears to occur as a direct result of the BIN1 knockdown.

Contrary to our hypothesis, $I_{Ca}$ recordings from shRNA-mBIN1 transduced old myocytes did not indicate restoration of ISO responsivity (Fig. 6a, b, Supplementary Fig. 7a). In line with that finding, BIN1 knockdown did not reverse the endosomal swelling seen with aging, suggesting that it is not an essential contributor to endosomal dysfunction in cardiomyocytes (Supplementary Fig. 9). However, basal $Ca^{2+}$ transient amplitude in shRNA-mBIN1 transduced old myocytes resembled that of young myocytes and $\beta$-adrenergic responsiveness was fully restored to young levels (Fig. 6c, d, Supplementary Fig. 7b).

The recovery of $Ca^{2+}$ transient amplitude, and not $I_{Ca}$, suggested that BIN1 knockdown may predominantly impact RyR2 function and/or distribution and not $Ca_V1.2$. To investigate that possibility, we performed SMLM on shRNA-mBIN1 transduced myocytes finding the nanoscale distribution of $Ca_V1.2$ was unaltered compared to non-transduced old myocytes in that they remained basally super-clustered

and did not recover the super-clustering response to ISO (Fig. 6e, f, and Supplementary Fig. 7d). In contrast, RyR2 appeared rejuvenated and exhibited a young cell-like nanoscale distribution and response to ISO (Fig. 6g, h, and Supplementary Fig. 7e). Overall, these results suggest that knockdown of BIN1 in aging hearts recovers cardiac $\beta$-adrenergic responsiveness by restoring RyR2 clustering plasticity.

### BIN1 knockdown in old animals rescues systolic but not diastolic function

Since all our experiments were performed in isolated cells, the question remained, would BIN1 knockdown improve old heart contractile function in vivo? To address this, we performed echocardiography in young, old, and old shRNA-transduced mice. In conscious mice, fractional shortening (FS) was reduced in old compared to young (Fig. 7a, b) suggesting an age-dependent systolic dysfunction. To examine the effects of shRNA-scrmb or shRNA-mBIN1 on old heart function, we performed echocardiograms on old mice to record their initial function prior to treatment with shRNA (Fig. 7c, d; *left*). Those mice were then retro-orbitally injected with either shRNA-scrmb (control) or shRNA-mBIN1 and after a two-week transduction period, repeat echocardiograms were obtained (Fig. 7c, d; *right*). In agreement with the rejuvenating effects of BIN1 knockdown observed in the in vitro functional analysis of $Ca^{2+}$

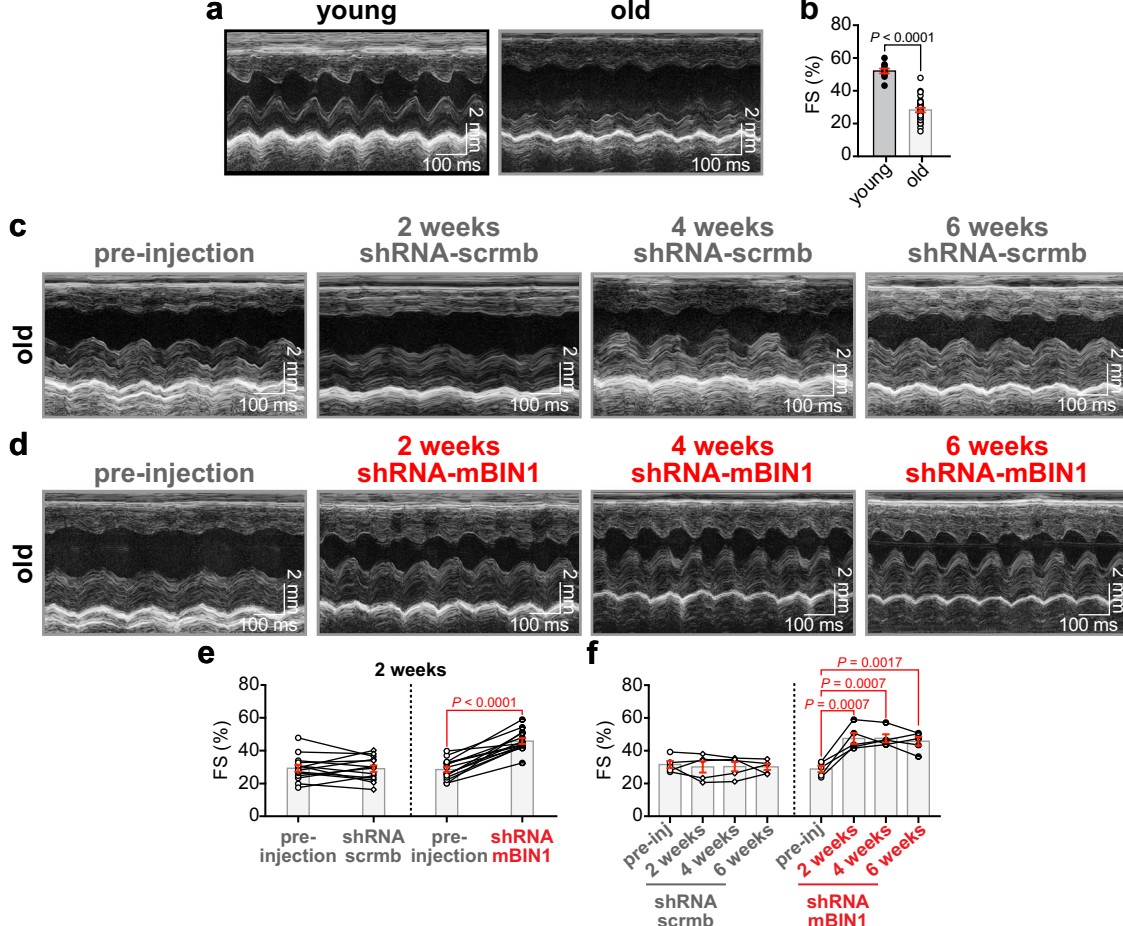

**Fig. 7 | BIN1 knockdown improves cardiac contractility in old mice.**
**a** Representative M-mode echocardiogram images from conscious young and old mice. **b** Summary dot-plots showing fractional shortening (FS) in young ($N = 10$) and old ($N = 30$) mice are shown. **c**, **d** Representative M-mode echocardiogram images from conscious old mice before and two, four, and six weeks after RO-injection of shRNA-scrmb (**c**) and shRNA-mBIN1 (**d**). **e** Summary dot-plots for FS showing paired results before and two weeks after RO-injection for shRNA-scrmb

($N = 14$) and shRNA-mBIN1 ($N = 14$). **f** summary dot-plots for FS showing paired results before and two, four, and six weeks after RO-injection for shRNA-scrmb ($N = 5$) and shRNA-mBIN1 ($N = 5$). Statistical analysis was performed on data in (**b**) using unpaired two-tailed Student's $t$-tests, on data in (**e**) using paired two-tailed Student's $t$-tests, and on data in (**f**) using one-way ANOVAs with multiple comparison post-hoc tests. Young data are from NIA young mice. Data are presented as mean ± SEM. Source data are provided in the Source Data file.

transients, basal ventricular function was restored to a more youthful phenotype upon BIN1 knockdown, with shRNA-mBIN1 treated old mice displaying improved FS (Fig. 7e). To investigate the longevity of the rejuvenating effect of BIN1 knockdown a subset of old mice were subsequently subjected to repeat echocardiograms at the four- and six-week post-injection timepoints with the beneficial effects on FS persisting over this extended timeframe (Fig. 7f). The restoration of youthful FS by BIN1 knockdown suggests this treatment rejuvenates systolic function in old mice.

Old mice exhibited signs of left ventricular (LV) hypertrophy including increased LV mass (Supplementary Fig. 10a). Two weeks of BIN1 knockdown significantly rejuvenated LV mass in paired old mice pre- and post-shRNA-mBIN1 that was not apparent in the shRNA-scrmb cohort (Supplementary Fig. 10b). Two weeks of shRNA-mBIN1 transduction also resulted in a slight rejuvenation in heart rate suggesting that pacemaker function may also be improved by BIN1 knockdown (Supplementary Fig. 10d–f), although further studies are required to confirm that. BIN1 knockdown also significantly rejuvenated several LV wall dimensions (Supplementary Fig. 11).

We further measured the in vivo effects of ISO on ventricular contractile function by performing echocardiograms on anesthetized young and old mice before and after intraperitoneal injection of ISO. As above, repeat echocardiograms were performed on unconscious old mice two weeks after retro-orbital injection of shRNA-scrmb or shRNA-mBIN1 (Supplementary Figs. 12 and 13). In agreement with the in vivo data from conscious mice, basal ventricular contractile function was rejuvenated by BIN1 knockdown with FS (Supplementary Figs. 12 and 13) improving toward youthful levels. Importantly this effect was not observed in shRNA-scrmb transduced mice suggesting a specific effect of BIN1 knockdown.

To assess changes in diastolic function with aging, we performed pulsed-wave Doppler imaging on unconscious mice. Old mice displayed an enhanced isovolumetric relaxation time (IVRT) and reduced mitral valve early to late ventricular filling velocity ratio (MV E/A), indicative of diastolic dysfunction, which was not remedied with BIN1 knockdown (Fig. 8). Altogether, these results suggest that BIN1 knockdown may represent a new therapeutic strategy to improve age-related deficits in systolic function.

## Age-dependent decrease in myofilament $Ca^{2+}$ sensitivity is rescued by BIN1 knockdown

$Ca^{2+}$ sensitivity of the myofilaments can also influence systolic and diastolic function thus we examined the expression and phosphorylation state of two cardiac myofilament proteins that are known to impact $Ca^{2+}$ sensitivity in a phosphorylation-dependent manner, specifically cardiac troponin I (cTnI)[38,39] and cardiac myosin binding protein-C (cMyBP-C)[40,41]. We probed for these myofilament proteins in heart lysates isolated from each cohort. We began with cTnI, finding its total expression increased with aging but relative expression of the phosphorylated form (cTnI(pS23/24)) reduced (Fig. 9a–c). shRNA-mBIN1 treatment displayed a trend toward a more youthful cTnI(pS23/24) expression but remained significantly different from that of young lysates. Total expression of cMyBP-C was unaltered with aging but its phosphorylation at S273 was significantly decreased (Fig. 9d–f). shRNA-mBIN1 treatment restored youthful cMyBP-C pS273 expression levels. Our results support a model where reduced phosphorylation of cMyBP-C in old hearts contributes to deficits in contractile function while reduced cTnI phosphorylation favors longer relaxation and diastolic dysfunction. BIN1 knockdown appears to restore systolic but not diastolic function by improving phosphorylation of cMyBP-C but not cTnI.

## Discussion

We report six novel findings (summarized in Fig. 10) that provide insight into the mechanistic basis of age-related cardiac dysfunction and reveal BIN1 as a new therapeutic target to rejuvenate the aging heart and extend cardiovascular health into old age: (1) patch clamp and $Ca^{2+}$ transient recordings revealed an age-associated loss of cardiac $\beta$-adrenergic responsiveness measured by the functional augmentation of $Ca_V1.2$ and RyR2 channel activity; (2) super-resolution imaging revealed an age-associated dysregulation of $Ca_V1.2$ and RyR2 clustering plasticity; (3) examination of early endosome pools and $Ca_V1.2$ dynamics revealed ISO-stimulated $Ca^{2+}$ channel mobility and recycling is impaired in aging due to trafficking deficits that cause endosomal traffic jams; (4) western blots and super-resolution imaging revealed that BIN1, important for t-tubule biogenesis[27], and $Ca_V1.2$, RyR2[18,25,26] and sarcomere[42] organization, is overexpressed in aging hearts; (5) shRNA-mediated BIN1 knockdown to youthful levels rejuvenated RyR2 clustering plasticity, basal $Ca^{2+}$ transient amplitudes and their functional augmentation by $\beta$-AR stimulation, and myofilament protein phosphorylation; and finally (6) in vivo echocardiography and Doppler imaging demonstrated that BIN1 knockdown in old mice restored youthful systolic (but not diastolic) function and substantially reversed the aging phenotype. Our knockdown approach using AAV9 packaged shRNA against BIN1 is rapid, occurring within two weeks of retro-orbital injection, and has translational potential as AAVs are approved for use in humans[43].

Prior studies have shown that basal $Ca_V1.2$ activity and $\beta$-AR-mediated $I_{Ca}$ enhancement are altered with aging[44]. However, the effects on basal $Ca_V1.2$ currents appear to vary depending on sex, species, and possibly even strain with various groups reporting enhanced[44–46], unchanged[47], or reduced[48–50] currents in aged models. In our hands, myocytes from ~24-month-old C57BL/6 mice had enhanced basal $Ca_V1.2$ currents compared to young (~3-month-old). Old myocytes also exhibited basally enhanced $Ca^{2+}$ transient amplitudes. Prior findings on $Ca^{2+}$ transient alterations in aging cardiomyocytes range from basal amplitude augmentation[45,51] and faster decay kinetics[52] to no or little reported change[50], to basal reduction in $Ca^{2+}$ transient amplitude and slower rate of decay[46,53]. This echoes the variability in age-associated alterations of $I_{Ca}$ and reflects the complexity of aging. Importantly though, and in line with our results, there is agreement that the ability of $\beta$-AR signaling to augment $Ca^{2+}$ transient amplitude and tune EC-coupling and inotropy is diminished with aging[7,8,54–59]. We cannot rule out that our 20-min loading of cells with 10 μM Rhod-2-AM at room temperature may have affected cytosolic calcium buffering or $Ca^{2+}$ handling by organelles such as the SR or mitochondria. However, the same loading protocol was used in all $Ca^{2+}$ transient experiments so all cells across all cohorts experienced the same conditions.

This study provides the first evidence linking age-associated $\beta$-AR hypo-responsivity to alterations in $Ca^{2+}$ channel clustering and organization. Super-resolution microscopy studies of the nanoscale arrangement of $Ca_V1.2$ and RyR2 have fueled the emerging concept that tunable insertion, recycling, and clustering of these channels physiologically during receptor signaling cascades or pathologically in disease, affects $Ca^{2+}$ release efficiency and myocardial contractility[15,16,19,20,60–62]. Larger $Ca_V1.2$ clusters facilitate enhanced $Ca_V1.2$-$Ca_V1.2$ physical interactions and cooperative gating that amplifies $I_{Ca}$[16,17,63] while larger RyR2 clusters favor enhanced $Ca^{2+}$ spark frequency and larger $Ca^{2+}$ transients[21]. We previously reported that acute $\beta$-AR activation in young ventricular myocytes promotes rapid, dynamic augmentation of sarcolemmal $Ca_V1.2$ channel abundance as channels are mobilized from endosomal reservoirs via fast and slow recycling pathways[15,16]. Here, super-resolution microscopy and dynamic channel tracking experiments demonstrated that this response is impaired with aging, such that $Ca_V1.2$ are basally super-clustered to a similar extent to that seen in young ISO-stimulated myocytes. These results, correlated with the basally elevated $I_{Ca}$ and impaired current augmentation with ISO, hint that the endosomal reservoir of $Ca_V1.2$ cargo has already been expended and/or

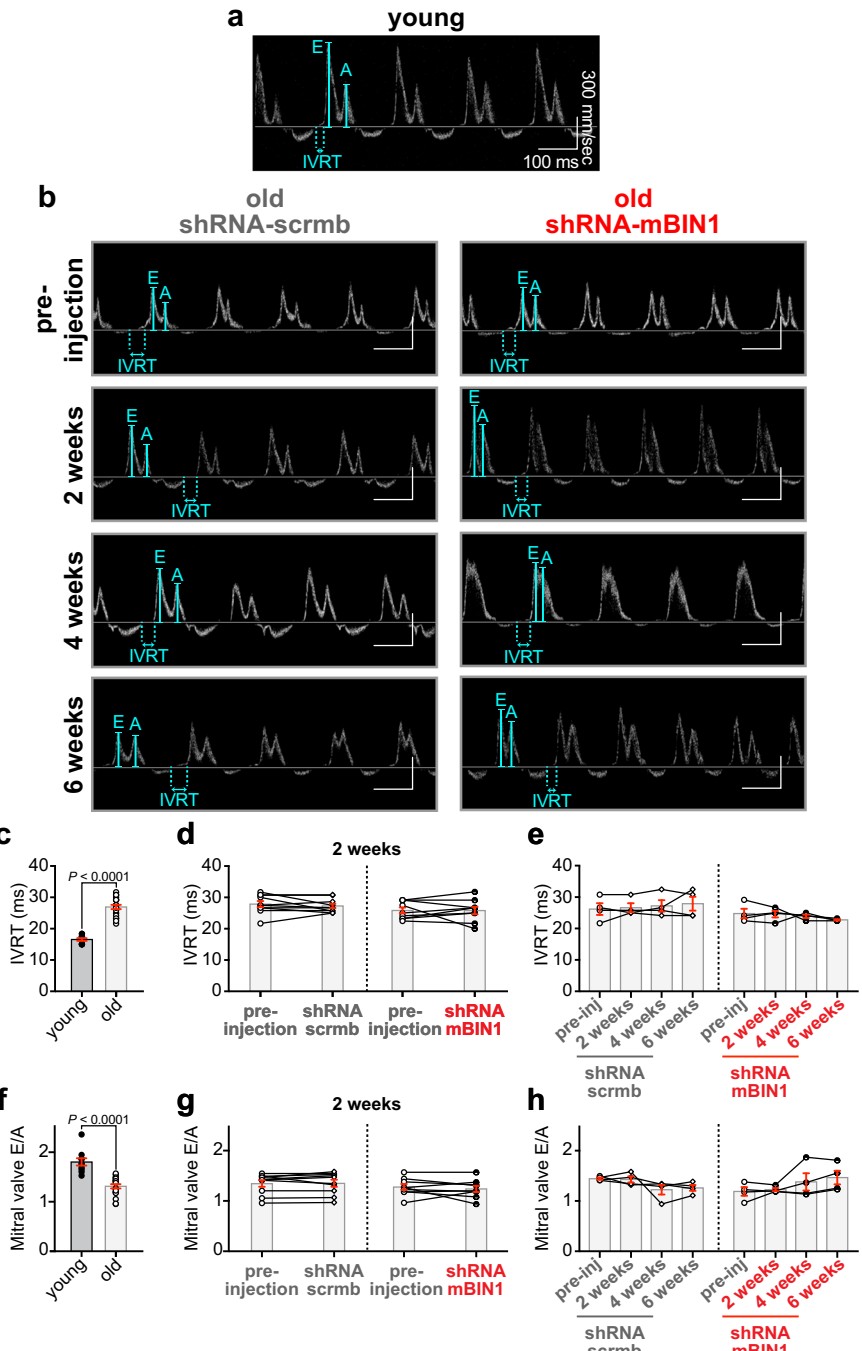

**Fig. 8 | BIN1 knockdown does not recover age-related diastolic dysfunction.**
**a**, **b** Representative pulsed-wave (PW) Doppler images from unconscious young (**a**) and old (**b**) mice before and two, four, and six weeks after RO-injection of shRNA-scrmb (*left*) and shRNA-mBIN1 (*right*). Summary dot-plots for young (*N* = 10) and old (*N* = 19) mice, paired results before and two weeks after RO-injection of old mice with shRNA-scrmb (*N* = 10) and shRNA-mBIN1 (*N* = 9), and paired results before and two, four and six weeks after RO-injection of old mice with shRNA-scrmb (*N* = 4) and shRNA-mBIN1 (*N* = 4) for the following diastolic measurements are displayed: **c**−**e** isovolumetric relaxation time (IVRT) and, **f**−**h** mitral valve E/A ratio (MV E/A). Statistical analysis was performed on data in (**c**, **f**) using unpaired two-tailed Student's *t*-tests, on data in (**d**, **g**) using paired two-tailed Student's *t*-tests, and on data in (**e**, **h**) using one-way ANOVAs with multiple comparison post-hoc tests. Young data are from NIA young mice. Data are presented as mean ± SEM. Source data are provided in the Source Data file.

endosomal recycling pathways are impaired. Our data suggest the latter is true, revealing reduced ISO-induced insertions and endocytosis in old myocytes, an aggregation of channels on swollen endosomes, and a larger pool of static channels that appear stranded in the sarcolemma. Whether this pool of static channels reflects an increased lifetime of Ca$_V$1.2 in the membrane remains to be established but it is reminiscent of our prior finding of an enhanced static population in myocytes treated with the actin depolymerizing agent latrunculin-A[15]. Actin polymerization is an essential aspect of endocytosis, generating the lateral force to facilitate the scission of the endocytic vesicle[64]. It is possible that loss of endocytic processes is an adaptive response in old myocytes to preserve sarcolemmal Ca$_V$1.2 in the face of impaired endosomal recycling. The main conclusion from this data is that the endosomal pathway and homeostatic regulation of Ca$_V$1.2 channel expression at the sarcolemma is impaired with aging.

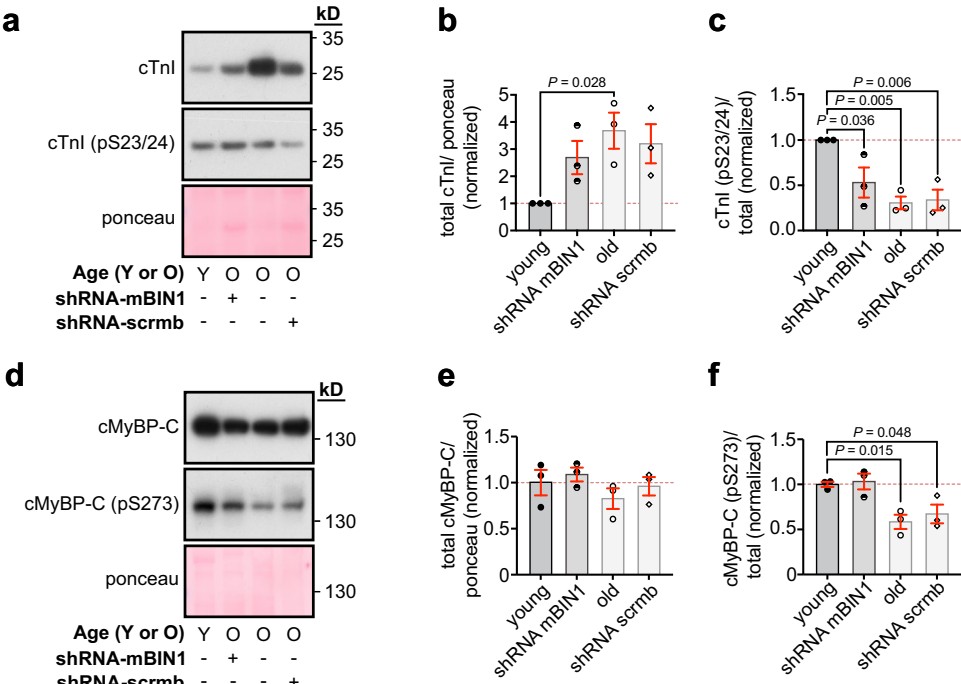

**Fig. 9 | Myofilament Ca²⁺ sensitivity is altered in aging. a** Western blot of total (*top*) and pS23/24 (*bottom*) cTnI expression in whole heart lysates from young, old, shRNA-mBIN1 or shRNA-scrmb transduced mice. Ponceau was used for normalization. **b** Histogram showing normalized total cTnI levels relative to young ($N = 3$ biological replicates). **c** Histogram showing cTnI pS23/24 normalized to total cTnI relative to young ($N = 3$ biological replicates). **d** Western blot of total (*top*) and pS273 (*bottom*) cMyBP-C expression in whole heart lysates from young, old, shRNA-mBIN1 or shRNA-scrmb transduced mice. Ponceau was used for normalization. **e** Histogram showing normalized total cMyBP-C levels relative to young ($N = 3$

biological replicates). **f** Histogram showing cMyBP-C pS273 normalized to total cMyBP-C relative to young ($N = 3$ biological replicates). Statistical analysis was performed on data in (**b, c, e, f**) using one-way ANOVAs with multiple comparisons post-hoc test. Young data are from JAX young mice. Note there was no significant difference in cTnI (total and pS23/24) or cMyBP-C (total and pS273) expression when JAX and NIA-sourced young mice were compared (see Supplementary Fig. 1m–r). Data are presented as mean ± SEM. See Supplemental Information for uncropped scans and number of technical replicates for western blots. Source data are provided in the Source Data file.

Deficiencies in recycling and endosomal enlargement are familiar sights in Alzheimer's disease (AD) where endosomal enlargement is the first neuro-cytopathological hallmark of AD, visible even before neurofibrillary tangles or amyloid β plaques[23]. An imbalance in cargo trafficking into and out of endosomes is thought to result in endosome swelling and recycling pathway deficiencies termed endosomal traffic jams[24]. As mentioned earlier, alterations in BIN1 expression are thought to contribute to endosomal dysregulation in AD neurons. Thus, when we found BIN1 expression was significantly increased in aging, and appeared to assume a more endosomal localization pattern, we thought, since it was in the "right place at the right time" that knockdown of BIN1 would restore youthful endosomal trafficking and improve Ca$_V$1.2 channel cargo mobilization, recovering β-AR responsivity and Ca$_V$1.2 channel functional augmentation. That hypothesis was thoroughly disproven with results showing BIN1 knockdown did not in fact accomplish any of those things. Several other proteins have been implicated as sources or contributors to endosomal dysfunction and swelling in AD neurons as reviewed[24]. Future studies should investigate whether any of those proteins underlie age-associated endosomal dysfunction and impaired Ca$_V$1.2 recycling in ventricular myocytes. Another possibility that should be pursued is whether endosomal pathway disruption also affects the sarcolemmal expression and/or recovery from desensitization of β-ARs. Old hearts exhibit reduced expression of β-ARs and impaired recovery from desensitization compared to young[7,54]. Normally, agonist binding to β-ARs triggers G-protein signaling and subsequent receptor phosphorylation, and recruitment of β-arrestin which halts receptor signaling by reducing functional coupling to heterotrimeric G-proteins[65,66]. β-ARs are then endocytosed, dephosphorylated, β-arrestin dissociates, they

resensitize, and recycle[65,66]. If this process becomes impaired due to age-associated endosomal traffic jams this would prolong β-AR desensitization, potentially contributing to the age-dependent hyporesponsivity to β-AR-agonists.

On the other side of the dyadic cleft, RyR2 also displays enlarged basal clusters that fail to undergo augmentation with ISO in aging myocytes. Other groups have reported phosphorylation-stimulated clustering of RyR2[18,19] but this is the first report of disruption of this dynamic response with aging. Cardiac RyR2 trafficking and mobility is poorly understood and understudied but one report has linked RyR2 redistribution upon phosphorylation to BIN1, evidenced by reduced recruitment of phosphorylated RyR2 to dyadic regions in cardiac-specific *BIN1* knockout cardiomyocytes[18]. Interestingly, BIN1 knockdown in old mice had a more profound effect on RyR2 distribution, Ca²⁺ transient amplitude, and inotropy than it had on Ca$_V$1.2. To that end, RyR2 nanoscale distribution was rejuvenated by BIN1 knockdown in old hearts but Ca$_V$1.2 still exhibited the old super-clustered phenotype and heightened basal $I_{Ca}$. So, while our initial rationale for knocking down BIN1 in the aging heart was flawed, serendipitously we discovered that BIN1 knockdown has a rejuvenating effect on RyR2.

Echocardiography confirmed the benefits of BIN1 knockdown on the aging heart with systolic function substantially improved after just a two-week knockdown of BIN1. In humans, resting systolic function is generally preserved with aging, however, it becomes significantly impaired during exercise and acute stress[3,67,68]. The resting systolic dysfunction observed in the current study may reflect the difference between human and mouse heart basal autonomic balance. Mice have relatively more sympathetic nervous system activity at rest than humans. With aging, autonomic balance

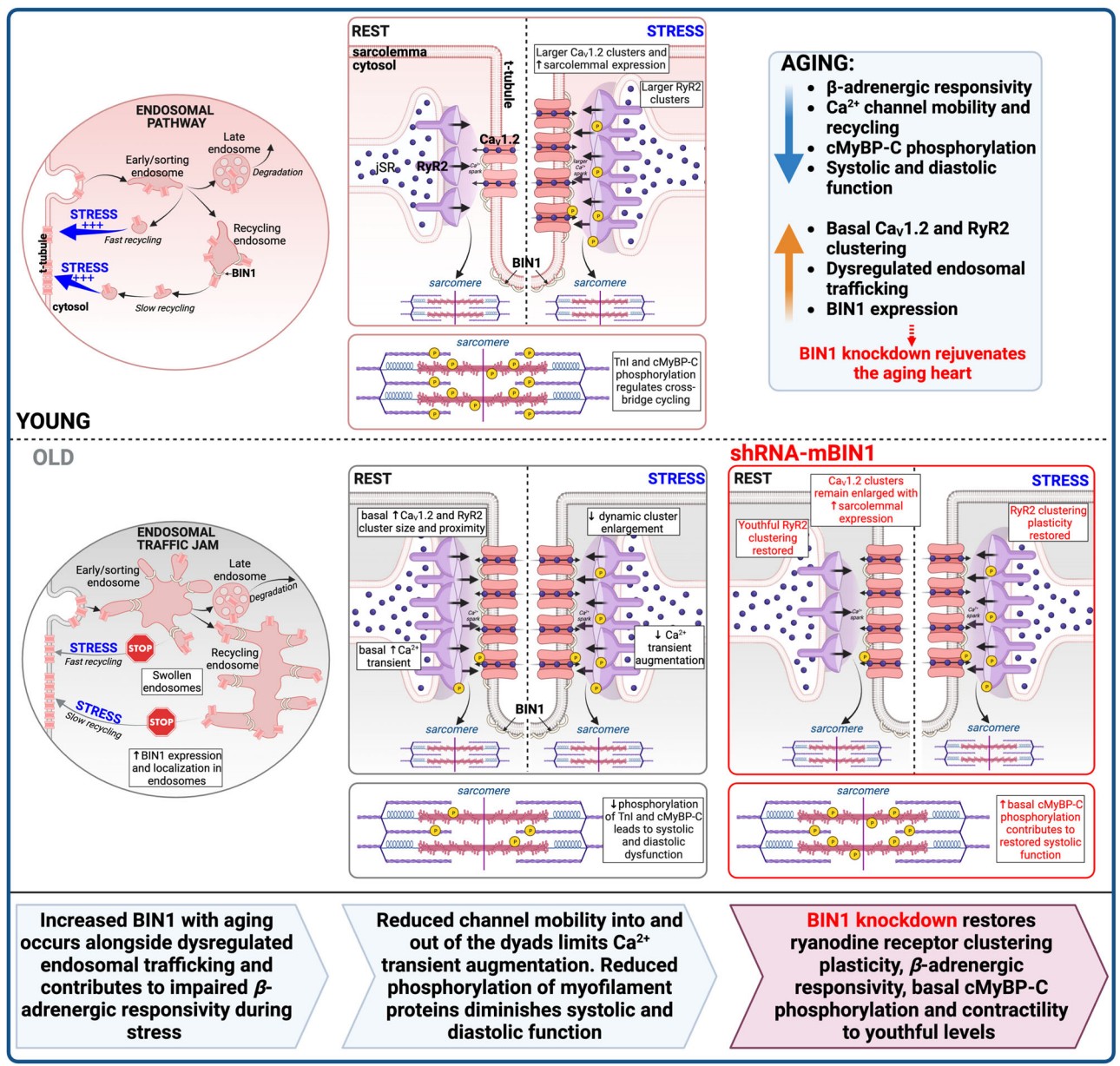

**Fig. 10 | BIN1 knockdown rejuvenates the aging heart.** The main findings of our study graphically illustrated and summarized. *Top*: In healthy young cells, Ca$_V$1.2 channels undergo endosomal recycling, where channels on endosomes are either marked for degradation through the late endosome pathway or are recycled to the sarcolemma through the fast and slow recycling pathways. Following *β*-adrenergic receptor (*β*-AR) stimulation, a pool of channels localized to endosomes are mobilized to the membrane, resulting in larger Ca$_V$1.2 clusters along t-tubules. Across the dyad, RyR2 clusters on the sarcoplasmic reticulum also increase following βAR stimulation ensuring efficient Ca$^{2+}$-induced Ca$^{2+}$-release. This increase in cytosolic Ca$^{2+}$, along with increased phosphorylation of cardiac Troponin I (cTnI) and cardiac myosin binding protein-C (cMyBP-C) within the sarcomere, allows for enhanced contractility under acute stress to cope with elevated hemodynamic and metabolic demands. *Bottom*: In aging, Bridging Integrator 1 (BIN1) protein levels are increased, accompanied by a swelling of endosomes and subsequent dysregulation of endosomal trafficking of Ca$_V$1.2. Ca$_V$1.2 and RyR2 channels are basally super-clustered at the dyads and lose *β*-AR responsivity. Reduced phosphorylation of cTnI and cMyBP-C result in systolic and diastolic dysfunction. BIN1 knockdown in aging recovers RyR2 clustering plasticity and Ca$^{2+}$ transient responsivity to *β*-AR stimulation. Phosphorylation of cMyBP-C is basally restored, and contractility is recovered to youthful levels. Thus, BIN1 knockdown rejuvenates the aging heart. Created with BioRender.com.

becomes even more tilted toward sympathetic activity as para-sympathetic input appears reduced[69,70]. In this way, a resting mouse heart is not necessarily a fair comparison to a resting human heart and may instead be a better reflection of an exercising human heart. Thus, BIN1 knockdown may present a therapeutic option to enhance systolic function in aging humans during acute exercise and stress.

Diastolic dysfunction is a more universal finding with aging in both humans and mice[4,71] and our findings of a prolonged IVRT with aging and reduced mitral valve E/A ratio agree with prior studies reporting impaired relaxation. However, BIN1 knockdown did not remedy diastolic dysfunction. This is perhaps expected as enhanced fibrosis[71] with aging is one of the main causes of the enhanced stiffness that underlies diastolic issues in the aging heart, and there is no reason to suspect that BIN1 knockdown would improve fibrosis.

Reduced Ca$^{2+}$ sensitivity of the myofilaments can also contribute to diastolic dysfunction. Enhanced phosphorylation of the thin myo-filament protein cTnI by PKA at ser-23 and −24 accelerates myocardial relaxation by reducing myofilament Ca$^{2+}$ sensitivity, favoring Ca$^{2+}$

dissociation from troponin-C, and leading to faster actin-myosin detachment[38,39]. Our results indicate that aging hearts display significantly lower levels of cTnI phosphorylation favoring slower relaxation. This deficit was not improved or corrected by BIN1 knockdown and diastolic dysfunction prevailed. Given the reciprocal relationship between cTnI phosphorylation and myofilament $Ca^{2+}$ sensitivity, one would predict that the age-associated reduction in cTnI pS23/24 levels would result in increased $Ca^{2+}$ sensitivity in old mice compared to young, leading to increased $Ca^{2+}$-dependent force generation. However, our in vitro finding of larger amplitude $Ca^{2+}$ transients in old myocytes accompanied by reduced in vivo contractile function in old mice suggests quite the opposite. Specifically, our functional results suggest an age-associated reduction in the $Ca^{2+}$ sensitivity of the myofilaments that is remedied by BIN1 knockdown. Thus, we broadened our search to examine cMyBP-C, another myofilament protein. Reduced phosphorylation of cMyBP-C causes decreased myofilament $Ca^{2+}$ sensitivity[41] and contractile dysfunction in heart failure patients[40]. Furthermore, in the dephosphorylated state, cMyBP-C inhibits actin-myosin interactions leading to reduced force development[41]. We found old mouse hearts had lower cMyBP-C pS273[72] expression levels than young hearts. A similar age-associated decline in cMyBP-C phosphorylation has been reported by others and found to correlate with deterioration of cardiac function[73]. Furthermore, phosphorylation-mimetic cMyBP-C mutant mice exhibit preserved cardiac function and enhanced longevity suggesting phosphorylation of cMyBP-C is cardioprotective[73]. We report that BIN1 knockdown rescues cMyBP-C pS273 phosphorylation levels which potentially explains the improved contractile function and $Ca^{2+}$ sensitivity in shRNA-mBIN1 transduced mice. But how can one link BIN1 expression levels to sarcomeric proteins? While we do not yet have the full mechanistic picture of how this occurs, there is evidence that BIN1 binds to and influences sarcomeric proteins. For example, in skeletal muscle, BIN1 is said to act as an essential adapter protein and scaffold that promotes mature sarcomere assembly[42]. Increased BIN1 expression is also known to promote enhanced expression of myosin heavy chain[74] and to associate with and stabilize actin filaments[75]. Future studies should examine the mechanism of BIN1 regulatory influences over cMyBP-C but it is an intriguing idea first proposed by Fernando et al.[42], that a single protein (BIN1) could orchestrate sarcolemmal and sarcomeric architecture and regulation.

It is important to mention that although our study demonstrates the detrimental effects of BIN1 upregulation on systolic function with aging, prior work has revealed that BIN1 *downregulation* is associated with heart failure[26,29] and established that cardiomyocyte-specific loss of BIN1 causes dilated cardiomyopathy[76]. These findings are not in conflict but instead suggest that BIN1 levels must be tightly controlled so that deviation in expression, whether that be too much or too little, has negative consequences for heart health and function. In Alzheimer's disease, such a model has been presented where too much or too little BIN1 expression are both associated with enhanced risk of disease[35,77]. Exogenous supplementation of a cardiac-specific isoform of BIN1 has thus been proposed as a heart failure therapeutic and has been demonstrated to improve cardiac function in mice with pressure overload-induced heart failure[78]. Our findings provide an important cautionary note that warns of a potentially tight therapeutic window for this approach where too much BIN1 supplementation may trigger further deterioration of systolic function and worsen the disease.

We acknowledge a limitation of our study lies in the lack of isoform-specific analysis of BIN1 expression with aging and subsequent knockdown. It is possible and indeed likely that the age-associated alterations in BIN1 expression we report here will not uniformly apply to all isoforms. This should be investigated in future studies using antibodies specific to single isoforms such as those described in recent work[37]. We further acknowledge that a subset of our young mouse data was collected from non-source-matched mice. In all instances where

we pooled results from JAX and NIA-sourced age-matched young C57Bl/6 mice, a statistical comparison was first performed to assess differences between the two datasets. No source-dependent statistical differences were ever detected in any of the measured parameters in this study (Supplementary Fig. 1). Accordingly, pooling was deemed permissible and akin to pooling data across sexes when no sex-dependent differences were detected. However, just as human populations age differently it is possible that C57Bl/6 mice sourced from JAX may age differently than those obtained from the NIA, we thus acknowledge this as a limitation of our study. Critically, the benefits of BIN1 knockdown reported herein were observed in gold-standard paired experiments by examining cardiac function in individual age- and source-matched mice before and after BIN1 knockdown using shRNA. This robust approach allowed us to track the cardiac function of individual two-year-old mice before, and subsequently at two, four, and six weeks after shRNA-mediated knockdown of BIN1 (or scrambled control) to unequivocally demonstrate an association between BIN1 knockdown and improved systolic function both in individual mice and on average across the population.

In summary, we find that BIN1 knockdown restores youthful $Ca^{2+}$ transient amplitude, $\beta$-AR-mediated transient augmentation, and in vivo systolic function in the aging myocardium. This study sets the groundwork for future work and raises the hope that a similar approach could improve cardiac function, restore exercise tolerance, and increase the capacity to endure acute stress in elderly patients.

## Methods

### Ethical approval
University of California Davis Institutional Animal Care and Use Committee (IACUC) approved all procedures involving mice which were conducted in strict compliance with the Guide for the Care and Use of Laboratory Animals[79] (protocols: 22848 and 21182). Mice were kept in a vivarium with a standard 12/12 light-dark cycle, a temperature range of 68–79 °F, and humidity between 30–70%. They received standard chow (Laboratory Rodent Diet 5001, LabDiet, St Louis, MO, USA) and water *ad libitum*. Mice were euthanized using an intraperitoneal injection of pentobarbital solution (>100 mg/kg; B euthanasia-D Special; Merck & Co., Inc., Rahway, NJ, USA).

### Isolation of mouse ventricular myocytes
All experiments were performed on male mice. C57Bl/6 3–5-month-old (referred to as "young") and 21–25-month-old mice (referred to as "old") were sourced from the NIA Aged Rodent Colony (Charles River Laboratories) unless otherwise stated. In some experiments, young C57Bl/6 mice sourced from The Jackson Laboratory (JAX; Sacramento, CA, USA) were utilized. Data from JAX-sourced young and NIA-sourced young mice are statistically compared in Supplementary Fig. 1 and no significant differences were detected in any of the measured parameters. Figure legends specify which data were obtained from NIA-sourced or JAX-sourced young mice.

After intraperitoneal injection of pentobarbital solution, hearts were surgically removed and repeatedly plunged into chilled digestion buffer (130 mM NaCl, 5 mM KCl, 3 mM Na-pyruvate, 25 mM HEPES, 0.5 mM $MgCl_2$, 0.33 mM $NaH_2PO_4$, 22 mM glucose, and 150 µM EGTA) to encourage replacement of blood in the chambers and ventricular myocytes were isolated using the Langendorff technique[15,16]. Briefly, aortic cannulation was performed and used to retrogradely perfuse the heart with 37 °C digestion buffer supplemented with 50 µM $CaCl_2$ (Thermo Fisher Scientific, Rockford, IL, USA), 0.04 mg/ml protease (Sigma–Aldrich, Inc., St. Louis, MO, USA) and 1.4 mg/ml type 2 collagenase (Worthington Biochemical, Lakewood, NJ, USA). The decision that digestion was complete was made by monitoring the color, shape, and texture of the perfused heart, removing it from the Langendorff system when it appeared pale, flaccid, and soft to the touch. Ventricles were then cut away from the rest of the heart, chopped into smaller

pieces, and myocytes were dissociated by gently pipetting the pieces in 37 °C digestion buffer supplemented with 0.96 mg/mL type 2 collagenase, 0.04 mg/mL protease, 100 μM $CaCl_2$ and 10 mg/ml BSA (Sigma–Aldrich). After 2 min centrifugation at $9 \times g$, the supernatant was discarded, and cells were resuspended in a wash buffer consisting of the digestion buffer supplemented with 10 mg/ml BSA and 250 μM $CaCl_2$. A final 2 min centrifugation step was performed, the supernatant wash buffer was aspirated away, and the cells were resuspended in the appropriate solution for the planned experiments.

Strict standards for isolation and cell quality were adhered to. Accordingly, cells were considered "healthy" and pursued for experiments when they fulfilled the following criteria:

(1) Cell isolation: at least 75% cell survival was required to consider isolation successful and to proceed with further evaluation and experiments.

(2) Cell morphology: "rod-shaped"/"brick-shaped" cells with clear striations, no blebbing, and no bunching at the cell ends.

(3) Cell stability: cells were required to be quiescent in a physiological (1.8 mM $Ca^{2+}$-containing) solution, with no spontaneous action potentials and accompanying contractions.

(4) When electrophysiology was performed, cells were pursued if they were found capable of holding stable seals with no more than 200 pA leak (but usually <50 pA) for at least 10 min.

## Patch clamp electrophysiology

Whole-cell patch clamp was performed at room temperature on freshly isolated ventricular myocytes to record $Ca_V1.2$ channel currents ($I_{Ca}$). Borosilicate glass pipettes (Sutter instrument Novato, CA, USA) were fire polished to 1–3 MΩ resistance, and filled with an internal solution containing 87 mM Cs-aspartate, 20 mM CsCl, 1 mM $MgCl_2$, 10 mM HEPES, 10 mM EGTA and 5 mM MgATP (pH adjusted to 7.2 with CsOH). The MgATP was added to the solution on the day of the experiments. 50 μM escin was added to the patch pipette for perforated patch clamp experiments, and a successful perforated configuration was assumed when series resistance fell below 15 MΩ.

Initially, myocytes were perfused with an external Tyrode's solution containing 140 mM NaCl, 5 mM KCl, 10 mM HEPES, 10 mM Glucose, 1 mM $MgCl_2$, and 2 mM $CaCl_2$ (pH adjusted to 7.4 with NaOH). Once a whole-cell/perforated configuration was obtained, perfusion was switched to a solution containing 5 mM CsCl, 10 mM HEPES, 10 mM Glucose, 140 mM NMDG, 1 mM $MgCl_2$ and 2 mM $CaCl_2$ (pH adjusted to 7.3 with HCl).

To minimize the effects of $I_{Ca}$ rundown, cells were held at a holding potential of −80 mV for 5 min prior to the recording of control currents. Next, cells were held at −80 mV and, to inactivate $Na^+$ currents, stepped to −40 mV for 100 ms, followed by 300 ms steps to voltages ranging from −60 to +90 mV. Recordings were obtained in control conditions and after 2–3 min of perfusion with 100 nM isoproterenol (ISO; Sigma–Aldrich) in the external bath solution. For perforated patch clamp recordings, cells were held at −80 mV, stepped to −40 mV for 100 ms, followed by a 300 ms step to −10 mV. 100 nM ISO in the external bath solution was added to the bath after 6 stable sweeps and perfused for 3.5 min to observe the time course of $I_{Ca}$ response to ISO.

Currents were sampled at a frequency of 10 kHz and low-pass-filtered at 2 kHz using an Axopatch 200B amplifier (Molecular Devices, Sunnyvale, CA, USA), digitized using a Digidata 1550B plus Humsilencer (Molecular Devices) and acquired using pClamp (Molecular Devices). Analysis was performed using Clampfit software (Molecular Devices). Membrane potentials were corrected for a liquid junction potential of −10 mV.

## Calcium transient recordings

Freshly isolated cardiomyocytes were loaded with 10 μM Rhod-2 AM (Thermo Fisher Scientific) for 20 min in the dark. After loading, myocytes were liberated from excess indicator via centrifugation for 2 min at $9 \times g$ and were subsequently resuspended in Tyrode's solution (140 mM NaCl, 5 mM KCl, 10 mM HEPES, 10 mM Glucose, 1 mM $MgCl_2$, and 2 mM $CaCl_2$, pH adjusted to 7.4 with NaOH) where they de-esterified for a further 20 min prior to commencement of experiments.

Myocytes pacing was performed at room temperature at 1 Hz using a 12 V square-wave stimulus evoked by a Myopacer electric field stimulator (IonOptix, LLC., Westwood, MA), and the resulting $Ca^{2+}$ transients were visualized by exciting Rhod-2 (with 594-nm laser light) and capturing a sequence of line-scans across the length of the cell at a rate of 560–813 lines per second using a Zeiss LSM 880 line-scanning confocal microscope (Carl Zeiss Microscopy, LLC., White Plains, NY, USA) equipped with a Plan-Apochromat 63×/1.40 N.A. oil immersion objective. Cells were continuously perfused with Tyrode's solution until a steady state was achieved when control transients were recorded. The same cell was then perfused with 100 nM ISO-containing Tyrode's, capturing transients at 1, 2, 3, and 5 min post-perfusion to assess the ISO-induced effects.

Rhod-2 fluorescent signals were converted into intracellular $Ca^{2+}$ concentration using the pseudo-ratiometric approach[80] and the equation: $[Ca^{2+}]_i = K_d(F/F_0)/(K_d/[Ca^{2+}]_{i\text{-rest}} + 1 - F/F_0)$[63,81] where $K_d$ was 720 nM[82]. $Ca^{2+}$ transient amplitude and decay were measured using Clampfit software (Molecular Devices).

## Single-molecule localization microscopy

Coverslips (#1.5; VWR, Radnor, PA, USA) were sonicated for 20 min in 1 M NaOH to remove any contaminants, followed by several washes in de-ionized water and stored in 70% ethanol until use. Freshly isolated ventricular myocytes were plated onto poly-$_L$-lysine (0.01%; Sigma–Aldrich); and laminin (20 μg ml$^{-1}$; Life Technologies, Carlsbad, CA, USA) coated coverslips and left for 45 min in a 37 °C incubator to adhere. Adhered cells were then treated for 8 min with either PBS (control) or 100 nM ISO in PBS at 37 °C. For time course experiments, cells were treated for 0, 1, 2, 3, 5, and 8 min with 100 nM ISO in PBS at 37 °C. The cells were then fixed and permeabilized in 100% ice-cold methanol (Thermo Fisher Scientific) for 5 min at −20 °C, followed by several washing steps in PBS. Fixed cells were blocked and permeabilized for 1 h at room temperature in 20% SEA BLOCK blocking buffer (Thermo Fisher Scientific) and 0.25% v/v Triton X-100 (Sigma–Aldrich) in PBS.

Cells were incubated overnight at 4 °C in rabbit polyclonal IgG anti-$Ca_V1.2$ (CACNA1C, ACC-003, Alomone Labs, Jerusalem, Israel; 1:300 dilution) or mouse monoclonal IgG1 anti-RyR2 (C3-33, MA3-916, Invitrogen, Waltham, MA, USA; 1:50 dilution) in blocking buffer. Excess primary antibodies were washed off with PBS before subsequent incubation in relevant secondary antibodies [Alexa Fluor 647-conjugated goat-anti-rabbit (A-21245, Invitrogen) or Alexa Fluor 647-conjugated goat-anti-mouse IgG1 (A-21240, Invitrogen) (both at a 1:1000 dilution in blocking buffer)] for 1 h at room temperature. After multiple washes in PBS, coverslips were mounted onto glass depression slides (neoLab, Heidelberg, Germany) with a cysteamine (MEA)-catalase/glucose/glucose oxidase (GLOX) imaging buffer containing TN buffer (50 mM Tris pH 8.0, 10 mM NaCl), a GLOX oxygen scavenging system (0.56 mg mL$^{-1}$ glucose oxidase, 34 μg mL$^{-1}$ catalase, 10% w/v glucose) and 100 mM MEA.

To exclude oxygen and hold coverslips in place, Twinsil silicone glue (Picodent, Wipperfürth, Germany) and aluminum tape (T204-1.0-AT205; Thorlabs Inc., Newton, NJ, USA) were used. Fixed cells were imaged on a Leica DMi8 microscope (Leica Microsystems, Wetzlar, Germany) in HiLo TIRF mode, using an HC PL APO 160 × 1.43 oil CORR GSD objective (Leica Microsystems). Ground state depletion was performed, and dye-blinking was elicited using a 638 nm/150 mW laser. Photon emission was detected with a Hamamatsu Flash 4.0 camera. Raw blinking images were collected with an exposure time of 10 ms for 50,000 frames using LAS X Life Science Software (Leica

Microsystems). Localization maps with 10 nm pixel size were generated using a detection threshold of 30 and a camera gain factor of 2.2. Mean $Ca_V1.2$ and RyR2 cluster areas were quantified using the "analyze particles" function in ImageJ/FIJI[83].

## Total internal reflection fluorescence (TIRF) imaging of live transduced cardiomyocytes

Isoflurane-anesthetized mice were retro-orbitally injected with AAV9-$Ca_V\beta_{2a}$-paGFP at a concentration of $4 \times 10^{12}$ vg/ml 4–6 weeks before the planned experiment when transduced myocytes were isolated and plated onto poly-L-lysine coated coverslips as described above. Photoactivation of the transduced $Ca_V\beta_{2a}$-paGFP was achieved by illumination with 405 nm LED light and photoactivated GFP was subsequently excited with 488 nm laser light at a TIRF penetration depth of 153 nm on an Olympus IX83 inverted microscope equipped with a Cell-TIRF MITICO and a 60×/1.49 N.A. TIRF objective lens and an iXon Ultra 888 back-thinned EM-CCD camera (Andor, Belfast, Northern Ireland). Images were acquired at room temperature at a rate of 10.34 FPS. After a control period of 300 frames with Tyrode's solution perfusion, cells were stimulated with 100 nM ISO for 3 min (1861 frames).

ImageJ/FIJI was used to analyze and quantify $Ca_V\beta_{2a}$-paGFP populations. Images were bleach-corrected before application of a 20-pixel rolling ball background subtraction and a 10-frame moving average. Maximum intensity z-projections of the first 300 frames of each experiment were used to represent the control period and the final 300 frames of the ISO perfusion were used to represent the ISO-stimulated period. These control and ISO period z-projections were then subjected to thresholding to binarize them and image math was performed to visualize and quantify the following channel populations, those that were: (i) inserted/recycled during the ISO period (ISO − Ctrl), (ii) endocytosed during the course of the experiment (Ctrl − ISO); and (iii) those that remained static throughout (Ctrl * ISO)[15].

## Transferrin receptor recycling assay

Freshly isolated cardiomyocytes were resuspended in plating media containing MEM-Gibco (Thermo Fisher Scientific, 11090-081) supplemented with 1% penicillin/streptomycin, 2 mM Glutamax (Gibco Thermo Fisher Scientific), 4 mM NaHCO$_3$ (Thermo Fisher Scientific), 10 nM HEPES (Thermo Fisher Scientific) and 0.2% BSA (Sigma–Aldrich). Cells were plated on poly-l-lysine and laminin-coated coverslips and incubated with 10 μg/ml transferrin conjugated to Alexa Fluor 568 nm (Thermo Fisher Scientific) for one hour at 37 °C. After that, media was exchanged to plating media with 5% FBS and cells were imaged on a Zeiss LSM 880 super-resolution microscope equipped with an Airyscan detector, a Plan-Apochromat 63×/1.40 oil DIC M27 objective (Carl Zeiss Microscopy, LLC., White Plains, NY, USA), Zen software (Carl Zeiss Microscopy, LLC) and a heated stage to maintain temperature at 37 °C. To visualize the recycling of transferrin receptors images were acquired at timepoints 0 min, 10 min, 20 min, and 30 min, after the initial 1 h incubation with fluorescent ligand.

## Fixed cell immunostaining and Airyscan microscopy

Freshly isolated ventricular mouse myocytes were plated onto poly-L-lysine and laminin-coated coverslips and subjected to control (PBS) or 8 min ISO stimulation in a 37 °C incubator prior to fixation with 4% paraformaldehyde solution with a cytoskeletal preserving buffer (PEM) containing 80 mM PIPES, 5 mM EGTA, and 2 mM MgCl$_2$ for 10 min. After PBS rinse, cells were permeabilized for 10 min in 0.5% Triton X-100 (Sigma–Aldrich) and, after several rinses with PBS, blocked with 20% SEA Block (Thermo Fisher Scientific) and 0.25% v/v Triton X-100 for 1 h at room temperature. Primary antibodies diluted in blocking solution for overnight incubation were: rabbit polyclonal IgG anti-$Ca_V1.2$ (ACC-003, Alomone Labs; 1:300), mouse monoclonal anti-EEA1 IgG1 (ab70521, Abcam, Waltham, MA, USA; 1:250), and mouse monoclonal IgG$_1$ anti-Bin1, clone 2F11 (NBP2-21675, Novus Biologicals,

Littleton, CO, USA; 1:125). Cells were then washed with PBS and incubated for 45 min at RT with the relevant Alexa Fluor conjugated secondary antibodies (Life Technologies; 1:1000). Secondaries utilized included Alexa Fluor 488 goat-anti-rabbit (A-11034), Alexa Fluor 647 goat-anti-mouse IgG$_1$ (A-21240) and Alexa Fluor 488 goat-anti-mouse IgG$_1$ (A-21121)(Invitrogen; 1:1000). Coverslips were mounted onto glass slides and examined on a Zeiss LSM 880 super-resolution microscope equipped with an Airyscan detector and a Plan-Apochromat 63×/1.40 oil DIC M27 objective and Zen software.

## Western blot

Mice were euthanized as described, then whole hearts were collected and flash-frozen in liquid nitrogen. Frozen hearts were then homogenized in RIPA buffer (R26200, Research Products International, Mount Prospect, IL USA) containing protease inhibitor, 4 μM microcystin (Insolution microcystin-LR, Thermo Fisher Scientific), and 1 mM NaF using tubes pre-filled with beads (Next Advance, Inc., Troy, NY, USA) for centrifugation at 4 °C for 20 min. The supernatant was then extracted, and protein concentration was determined by BCA assay. All protein samples were denatured in SDS sample buffer (Bolt SDS Sample Buffer, Invitrogen) at 70 °C for at least 10 min. The protein was fractionated by SDS-PAGE 4–12% gradient acrylamide gels and transferred onto polyvinylidene difluoride membranes (PVDF; Bio-Rad Laboratories, Hercules, CA, USA) using transfer buffer (95.9 mM glycine, 12.5 mM Tris, 150 ml MeOH, up to 1 L H$_2$0) at 50 V for 10 h. PVDF membranes were ponceau stained and imaged using a photo scanner (Epson Perfection V600 photo). The membrane was cut into separate bands of interest and incubated in a blocking buffer (BB) consisting of 150 mM NaCl, 10 mM Tris-HCl, pH 7.4 (TBS) with 0.2% Tween (TBST), and 5% milk for 1 h at RT. Next, they were incubated with primary antibodies in BB overnight at 4 °C. BIN1 was detected by anti-BIN1 (2F11, NBP2-21675, 1:250, Novus Biologicals; 99D, 1:100, sc-13575 Santa Cruz Biotechnology Inc; 99D, 1:500, 05-449-C Sigma–Aldrich). Total cTnI was detected by anti-troponin I, (TI-4, AB_10573815, 1:100; TI-4 was deposited by Schiaffino, S. to the Developmental Studies Hybridoma Bank, created by the NICHD of the NIH and maintained at The University of Iowa, Department of Biology, Iowa City, IA 52242). Phosphorylated cTnI (pS23/24) was detected by anti-phospho-troponin I (Cardiac) (Ser23/24) (4004S, Cell Signaling Technology, 1:1000). Total cMyBP-C was detected by anti-cMyBP-C 2-14 (kindly provided by Sakthivel Sadayappan, University of Cincinnati, 1:10,000). Phosphorylated cMyBP-C was detected by anti-Ser 273 (kindly provided by Sakthivel Sadayappan, University of Cincinnati, 1:2500).

Membranes were washed for at least 30 min with at least five exchanges of BB. Membranes were subsequently incubated with horseradish peroxidase-conjugated secondary antibodies: anti-Mouse IgG (H + L)-HRP Conjugate (1706516, Bio-Rad) and monoclonal mouse anti-rabbit IgG light chain specific HRP conjugate (211-032-171, Jackson ImmunoResearch) both diluted in BB at 1:10,000, rocking for 1 h at RT. Secondary antibodies were then washed with TBST for 90 min with at least 6 exchanges of solution. The membranes were developed using chemiluminescent reagents (Immobilon Classico, Sigma–Aldrich; Prometheus ProSignal Femto, Genesee Scientific, El Cajon, CA, USA) on autoradiography film (Amersham Hyperfilm ECL, Cytivia, Global Life Sciences Solutions USA, LLC., Marlborough, MA, USA) using a film developer to quantify protein expression. Multiple scans were taken over varying time periods to ensure signals were in the linear range and bands were non-saturated. Films were scanned using the photo scanner and immunoreactive bands were quantified using densitometry in Adobe Photoshop, normalizing to protein loading using the Ponceau scan. Note, total protein is the only acceptable normalization standard in aging studies since the protein levels of many, if not all, the standard 'housekeepers' change with aging[84,85]. Individual lysates (biological replicates) were probed 1–6 times on separately run gels constituting technical replicates. In some cases, lanes were excluded from our final

calculations because of artifacts on the ponceau, or incomplete/missing bands that made analysis impossible (see Source Data). Uncropped and unprocessed scans of Western blots in the figures are supplied in the Supplementary Information.

## Quantitative RT-PCR

Freshly isolated young and old mouse ventricular cardiomyocytes were diluted to $1.5–2 \times 10^6$ cells per mL. Total RNA was extracted using an RNeasy MINI kit (QIAGEN, Hilden, Germany) and reverse transcribed into cDNA using the SuperScript III First-Strand Synthesis System (Invitrogen). Quantitative real-time PCR reactions (final volume of 12 µl) containing SYBR-green indicator (ThermoFisher) were performed in a 384-well reaction plate on an Applied Biosystems ViiA 7 Real-Time PCR system (Applied Biosystems, Waltham, MA) using the following *bin1* transcript primer pairs[37]: *Pan-Bin1*: ACGAAGGACGAG-CAGTTTGA (FW), CAGAAGCCAGATAGGTCCGAA (RV); *Bin1*: CCCCAAGTCCCCATCTCAGAG (FW), CCCATTCACAGTTGCGGAGAA (RV); *Bin1 + 13*: GACCACCCCCTCCCAGAG (FW), AACAT-GAATCCCGGGGGCAG (RV); *Bin1 + 17*: CCTCCAGATGGCTCCCCT (FW), GCCTCTGCTGGCTGAGATG (RV); *Bin1 + 13 + 17*: TGACG-CATTTGTCCCTGAGA (FW), CTGTCTCCCCTGGCTCCT (RV); *Bin1 + 11(+17)*: CTGGTCAGCCTAGAGAAGCAG (FW), GCAGCCGTGAGAA-CAGTTT (RV)[37]. Quantitative RT-PCR conditions were as follows: 10 min at 95 °C (initial denaturation), followed by 40 cycles of 15 s at 95 °C (denaturation) and 1 min at 60 °C (annealing and extension), followed by a final dissociation step[37].

Transcript expression levels were normalized to the housekeeping control gene *gapdh*. The primers used for *gapdh* were: GAAGCTTGTCATCAACGGGAAG (FW), TTTGATGT-TAGTGGGGTCTCGC (RV). In agreement with prior studies of *Bin1* expression in mouse heart[28,37], the exon 11-containing skeletal muscle *Bin1* isoforms *Bin1 + 11* and *Bin1 + 11 + 17* were not detected in the reaction. Data was averaged from three separate PCR reactions, with each reaction running each N-number in triplicate, to constitute technical replicates. Individual data points presented in Fig. 5c represent the mean result from each animal averaged over three technical replicates of each of the three runs.

## T-tubule staining

Fresh ventricular myocytes were loaded with 10 µM di-8-ANEPPS (Thermo Fisher Scientific) for 30 min at room temperature in the dark. The stained cardiomyocytes were centrifuged at $9 \times g$ and resuspended in Tyrode's solution (140 mM NaCl, 5 mM KCl, 10 mM HEPES, 10 mM Glucose, 1 mM $MgCl_2$, and 2 mM $CaCl_2$, pH adjusted to 7.4 with NaOH). Live cells were then imaged at room temperature on a Zeiss LSM 880 super-resolution microscope equipped with an Airyscan detector and a Plan-Apochromat 63×/1.40 oil DIC M27 objective (Carl Zeiss Microscopy, LLC., White Plains, NY, USA) using Zen software (Carl Zeiss Microscopy, LLC). T-tubule organization was analyzed and quantified in ImageJ using the TTorg plugin[86].

## Adeno-associated virus (AAV9) short hairpin RNA (shRNA) for BIN1 knockdown

AAV9-GFP-U6-m-Bin1-shRNA (shRNA-mBIN1) and AAV9-GFP-scrmb-shRNA (shRNA-scrmb) were purchased from Vector Biolabs (Malvern, PA, USA). For BIN1 knockdown, 22–25-month-old mice were anesthetized with vaporized isoflurane and retro-orbitally injected with the shRNA-mBIN1 or shRNA-scrmb at a concentration of $5 \times 10^{11}$ vg/ml two weeks before experiments when animals were euthanized, and myocytes isolated, or hearts harvested for biochemistry.

## Echocardiography

Echocardiography was performed using a Vevo 2100 imaging system (VisualSonics, Fujifilm, Toronto, ON, Canada) and an MS 550D probe (22–55 MHz). Systolic function from conscious and systolic and diastolic function from unconscious (anesthetized with 2% isofluorane) NIA-sourced young and old mice was evaluated using M-mode echocardiography and pulsed-wave Doppler. Two-dimensional measurements were used to extract fractional shortening (FS), left ventricular (LV) mass, heart rate, left ventricular anterior wall dimensions in diastole and systole (LVAW;d and LVAW;s), left ventricular posterior wall dimensions in diastole and systole (LVPW;d and LVPW;s), left ventricular inner diameter dimensions in diastole and systole (LVID;d and LVID;s), mitral valve E/A wave ratio (MV E/A) and isovolumetric relaxation time (IVRT) values[87,88]. Intraperitoneal injection of 0.1 mg/kg ISO diluted in saline was used on unconscious mice to measure β-AR response.

For paired BIN1 knockdown results, echocardiography was performed on old mice prior to and two, four, and six weeks after retro-orbital injections of AAV9-shRNA-scramble or AAV9-shRNA-mBIN1.

## Statistical analysis

N represents the number of animals and n represents the number of cells. Data are reported as mean ± SEM. GraphPad Prism software (GraphPad Software Inc., La Jolla, CA, USA) was used to graph and compare datasets using paired or unpaired Student's t-tests, one-way ANOVAs, or two-way ANOVAs with multiple comparison post-hoc tests. $P < 0.05$ was considered statistically significant.

## Reporting summary

Further information on research design is available in the Nature Portfolio Reporting Summary linked to this article.

## Data availability

Data generated in this study are provided in the Source Data file and Supplementary Information. Source data are provided with this paper.

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

## Acknowledgements

We extend our thanks to Dr. Sakthivel Sadayappan for the generous gift of the total cMyBP-C antibody and the cMyBP-C Ser-273 phospho-site specific antibody[72]. We thank Dr. L. Fernando Santana for reading and providing critical suggestions on our manuscript. This work was supported by NIH R01AG063796 and R01HL159304 to R.E.D.; R35GM149211 and RF1NS131379 to E.J.D.; an NIGMS funded Pharmacology Training Program T32 GM099608 (T.L.V. and H.C.S.); an AHA Predoctoral Fellowship 827909 to T.L.V., a Postdoctoral Fellowship from NIH T32 Training Grant in Basic & Translational Cardiovascular Science HL086350, Harold S. Geneen Charitable Trust Awards Program for Coronary Heart Disease, AHA CDA 24CDA1276831, and NIH F32 HL149288 to P.N.T.; an NIH F31 HL165815 to H.C.S.; by NIH R56HL167932, AHA 23SFRNPCS1060482 and UC Davis Internal Medicine Chair's Research Award to P.S., and by NIH R01HL085727, R01HL085844, R01HL137228, R01HL152055, S10OD010389 Core Equipment Grant, and VA Merit Review Grant I01 BX000576 and I01 CX001490 to N.C. Figure 10 was created using BioRender.com.

## Author contributions

M.W., S.G.d.V., T.L.V., P.N.T., H.C.S., A.D.C., and P.S. performed the research; M.W., S.G.d.V., T.L.V., P.N.T., H.C.S., P.S., and R.E.D. analyzed

data and wrote the manuscript; N.C., E.J.D. and R.E.D. designed research. All authors read and approved the final manuscript.

## Competing interests

The authors declare no competing interests.
