## [Peer Review File · Nature Communications]

REVIEWER COMMENTS

Reviewer #1 (Remarks to the Author):

Summary

The author's report that a two-week treatment to restore youthful Bridging Integrator 1 (BIN1) levels in the hearts of 24-month-old mice "rejuvenated" cardiac function and substantially reversed the aging phenotype. The results indicate that age-associated overexpression of BIN1 occurs in the context of dysregulated endosomal recycling and disrupted trafficking of cardiac CaV1.2 and type 2 ryanodine receptors. In vivo echocardiography revealed reduced systolic function in old mice. BIN1 knockdown using an adeno-associated virus serotype 9 packaged shRNA-mBIN1 restored the nanoscale distribution and clustering plasticity of ryanodine receptors and recovered Ca²⁺ transient amplitudes and cardiac systolic function toward youthful levels. Enhanced systolic function correlated with increased phosphorylation of the myofilament protein cardiac myosin binding protein-C. The authors' interrupt their results to indicate that BIN1 knockdown may be a novel therapeutic strategy to "rejuvenate" the aging myocardium.

General Comment

The paper appears to be a tour-de-force, and is very well written, with explicit quality controls. But, in general, the story sounds kind of "too good to be true." It's hard to imagine that any single molecule (BIN1 or whatever) can "rejuvenate" the aging myocardium. It's possible that the increase in BIN1 and its link to RyR and Cav1.2 clustering in the aging heart are due to adaptive mechanisms that preserve a high Ca transient in the presence of other age-associated cardiac deterioration. Maneuvers that reduce adaptive mechanism may have longer term deleterious effects on basal heart function that could be life threatening, despite the short-term increases function in vivo such as those in the present study.

Specific Comments

Major Comments

1. How confident are you that your interpretation of age-associated changes in cardiomyocyte function are maladaptive? Is it possible that the age-associated differences of clustering of ICaL to the surface membrane occurs is an attempt to compensate for reduced efficacy of beta-adrenergic directed signaling? Is it possible that the LV also adjusts its Ca²⁺ cycling characteristics to adapt to reduction in heart rate that accompanies advancing age? In a well performed study (Luany-Kleintop et al., J. Cell.

Biochem. 116: 2541–2551, 2015) it was reported that that cardiomyocyte-specific loss of Bin1 causes an age-associated dilated cardiomyopathy (DCM) that develops over time beginning by 8–10 months of age, and that younger animals rapidly develop DCM if cardiac pressure overload was created by transverse aortic constriction. Bin1 loss increased interstitial fibrosis and mislocalization of the voltage-dependent Ca²⁺ channel Cav1.2, and the lipid raft scaffold protein caveolin-3, which normally complexes with Bin1 and Cav1.2 in cardiomyocyte membranes. Thus, this report indicates that cardiac deficiency in Bin1 function causes age- and stress-associated heart failure. These authors suggest that cardiac specific Bin1 causes a novel preclinical model of age-associated terminal cardiac disease rather than rejuvenating the heart.

2. Do you suppose that age-associated changes in beta-adrenergic receptor numbers or affinity could be related to trafficking ICaL to the membrane in response to isoproterenol? BAR adaptor proteins are pleiotropic regulators of membrane dynamics and nuclear functions with roles in endocytosis, and in vesicle fusion and trafficking. (c.f. Luary-Kleintop et al., J. Cell. Biochem. 116: 2541–2551, 2015)

3. Does “stranding” Cav1.2 at the plasma membrane occur because the channel cannot dissociate from the membrane or the process of recycling back to channels of endosomes is impaired. It would help the reader to briefly define the roles of different RABs in trafficking.

4. In my opinion, BIN1 Knock down in young mice is also required to fully interpret the results that you report.

5. Is the “rejuvenating” effect of BIN1 knock down that you report long lasting? You have only monitored it for 2 weeks. Given the long-term disastrous results observed in BIN1 KO with increasing age (Luary-Kleintop et al., J. Cell. Biochem. 116: 2541–2551, 2015), do you suppose that the “rejuvenating” you report may only be a transient phenomenon?

6. It was recently discovered that PKA dependent phosphorylation of small RGK G-protein Rad (a Ca²⁺ channel inhibitor) is essential for regulation of basal ICaL current and for beta-adrenergic augmentation of Ca²⁺ influx in cardiomyocytes (Papa et al. Nat Cardiovasc Res 2022; 1:1022–1038). 4SA-Rad mice, in which four evolutionarily conserved PKA-phosphorylated serine residues of the endogenous murine Rad locus were replaced by alanine residues, have no response to beta-AR stimulation with ISO. This absence of the increase in Ca²⁺ current in response to ISO in VM from these genetically manipulated 4SA-Rad mice suggests that trafficking and insertion of additional Ca²⁺ channels in plasma membrane after β-AR stimulation might not be a singular mechanism responsible for the augmented Ca²⁺ influx during β-AR stimulation.

7. You do not consider the potential role of SERCA regulation in relation to your rejuvenation of the old heart. SERCA2 expression and activity is known to be reduced with aging and has important implications for pumping of Ca²⁺ to the SR and on the SR Ca²⁺ load that is a crucial factor in the release of Ca²⁺ via RyRs. Prior literature has shown that gene therapy can rejuvenate the older heart via enhancing Ca²⁺ cycling (c.f. Roger Hajjar papers).

8. Removal of Ca²⁺ from the cytosol is an important factor in the relaxation of the myofilaments, in addition to myofilament Ca²⁺ sensitivity on which you seem to focus, it would be important, to provide the relaxation time or relaxation rate in the transients.

9. Discrepancies between basal amplitudes of I_{CaL} or Ca²⁺ transients in aged VM observed in the present study and other reports (see above) create concerns how universal and relevant to aging is the major hypothesis: “that old myocytes exhibit basal super-clustering of both CaV1.2 and RyR2, with no additional dynamic response to ISO” presented in this MS.

10. What is known about aging with respect to phosphorylation, of phospholamban and RyRs at different sites? This would be important with respect to your Ca²⁺ transient measurements. It seems odd that you assessed the phosphorylation of myofilament proteins but not that of Ca²⁺ cycling proteins. In order to ascertain myofilament Ca²⁺ sensitivity, some measure of contraction is required. I don't seem to find this information in your report.

11. Does the iso effect to increase cardiomyocyte performance require increased phosphorylation of I_{CaL} and RyR to the membrane in order to effectively affect their trafficking? Is de-phosphorylation required to remove I_{CaL} from the surface membrane and return the RyR Cluster to the pre-iso level? These sorts of issues seem to be important and require some information on the activities of kinases and phosphatases in the aging myocardium.

12. I would add that I could not find the time of application of ISO that caused remodeling of Cav and Ryr clusters. Do they really move in 2 minutes? “Recordings were obtained in control conditions and after 2-3 mins of perfusion with 60 100 nM isoproterenol (ISO; Sigma-Aldrich) in the external bath solution.” Why specifically 2-3 min? Neither a reference, nor an example of time course is given. Depending on the time course 2 or 3 minutes could be a big difference.

13. Line 112 in results: a maximal ISO-stimulated current was referred to; you report only 1 concentration in your result (100 nmol). Had you performed a dose response to isoproterenol to determine the concentration at which the maximal response is achieved, and whether this differs by age? Numerous prior studies have indeed demonstrated that sensitivity for maximum response to β-

adrenergic receptor agonists (e.g. mixed β -1 and β -2 agonists, like isoproterenol, or β -1 agonist in the presence of α -adrenergic receptor blockade differ as age advances).

14. Please provide time course (with all statistics) to justify the 2-3 min of measurements of I_{CaL} and transients. It's well known that kinetics of most reactions become slowed with aging, and often time the same maximum could be achieved but it takes a longer time. It's possible that the time course for old and young animals might be different. The same problem might be with assessment of clustering. The time course of clustering also differ by age could be also different and when to assess the clustering could be important. There is no assessment of SERCA function that could be also important for data interpretation.

15. Although voltage clamp is the classical method to inquire about I_{CaL} amplitude, Action Potential shapes appear to differ with aging, and this might result in age-associated differences in I_{CaL} that are not evident in 300 msec voltage steps. It would have been instructive to measure the action potential characteristics in your isolated cardiac myocytes.

16. In a related issue, the amplitude and characteristics of the cytosolic Ca²⁺ transient and contraction may differ by age with respect to the pacing frequency employed in experiments; and this effect of pacing rate also differs in the presence and absence of isoproterenol and may also differ by age. The temperature at which the experiments were performed is also an important factor, with respect to the biophysical measurements you performed. I couldn't seem to find information about the temperature at which the in situ cardiac myocytes studies were conducted. Given these uncertainties, how confident are you that you have captured the "reality" of age-associated changes in your isolated cell experiments?

17. With respect to in vivo experiments, a major premise on which you base your study, that aging is associated with a reduced response to stress, e.g. reduced responsiveness to BAR signaling, that renders the heart more vulnerable to stress. Your in vivo ECG recording certainly must be a stress on the conscience mouse. Because the rationale is that in advanced age the response to stress is reduced, renders the interpretation of the results of the in vivo ECG in conscious mice might be difficult. Specifically, the response to stress is greater in young than older mice, and this increased stress response elevates HR and contractility to a greater extent in young vs old mice. By eyeballing the figures in conscious vs. unconscious state, seems to indicate that the HR difference in the conscious vs. unconscious states, in HR for example, is greater in young than in old mice. In my opinion, an analysis of variance of the echo results for main effects of consciousness and age, and age/consciousness interactions is required.

18. Line 277-278: you state the resultant increase in preload in old mice would be expected to enhance ventricular output via the Frank-Starling mechanism to maintain stroke volume in the face of reduced contractility. It seems that enhance and maintain are contradictory, which one do you propose is

correct? Because your HR did not change in response to BIN1 Knockdown, any enhancement of cardiac output must occur via an increase in stroke volume. Because the cardiac output increased following BIN1 knockdown in old mice, the Frank-Starling mechanism would not only have to maintain stroke volume but would have to increase stroke volume. I don't see any stroke volume data in your results, what was the stroke volume in young and old and following BIN1 knockdown, and did this differ in the conscious vs. non-conscious states.

19. Line 394: One interpretation of your results is that the heart reserve capacity is lost with aging because it is already expended at rest. It seems to me that if reserve capacity is lost, the body would increase sympathetic input to the heart in an attempt to maintain homeostasis of cardiac output that is essential for body organ needs. This increase in sympathetic tone could be the cause of increased trafficking of ICaL to the membrane. In this case, treatment of BAR blockade, might reduce the excessive baseline ICaL trafficking to the surface membrane.

Minor Comments

1. Lines 133-134: "Increased sarcolemmal CaV1.2 channel expression and clustering at rest in aging cells could explain the age-dependent increase in basal ICa." Did you study expression of Cav1.2 protein in the aged VM?

2. Line 384: It would be helpful to the reader to list a few of these proteins.

3. Line 450: to which myosin heavy chain do you refer? This may be an important consideration as there is a switch from α to β myosin heavy chain in rodents as age advances?

4. Figure 5 legend, line 743: "h" should be replaced by "k".

Reviewer #2 (Remarks to the Author):

In the current ms by Westhoff and co-workers the authors describe experiments on two groups of mice (old and young) to determine whether the knockdown of BIN1 was able to "restore" systolic dysfunctions the authors observed in the old hearts.

The authors found intracellular Ca transients to be doubled in amplitude and decay rates to be substantially increased in the old hearts but Iso-responses to be diminished. Alongside these cellular findings the authors reported on a 50% reduction in FS in the old hearts when compared to the young group. Interestingly, they found a massive upregulation of BIN1 expression in the older group. As a consequence of that AAV-mediated downregulation of BIN1 expression “restored” the localization of BIN1 as well as its cycling and the Iso-mediated increases in Ca transient amplitude. These results were supported by the echocardiographic studies in these mice (old, young AAV-scrambled, AAV-BIN1).

From these data the authors concluded that BIN1 downregulation might represent a potential novel therapeutic strategy to “restore” a young cardiac phenotype in the aging heart.

After several careful readings of the ms this reviewer has identified the following criticisms:

- This reviewer was surprised by the following findings that contradict most (>80%) of the previous reports on the aging heart including:
 - o Increased Ca transient amplitude by more than 100%. Reports usually depict decreases or constant amplitudes.
 - o Increased speed of Ca removal. Reports usually depict a slowdown of the decay phase of Ca transients.
 - o Loss of EF by 50%. Reports usually depict slight decreases or constant EF or FS.
- BIN1’s primary RNA is well known to undergo extensive splicing with splice variants associated to different functions of the particular BIN1-isoform. The authors appear to neglect this fact by solely analyzing full length BIN1.
- Loss of BIN1 expression has been reported to cause substantial T-tubular disarray and uncoupling of Ca channels in the plasma membrane and the RyR clusters in the SR membrane. In this it would be very important to analyze distribution and morphology of the T-tubular system and coupling efficiency of EC-coupling in the young and old hearts as well in the cells with downregulated BIN1 expression.
- One of the most striking details is that old and young mice were NOT from the same source, despite both being Blck6 mice. While the young group originated from Jackson Laboratories directly, the old group was derived from the “National Institute on Aging Aged Rodent Colony”. Considering the unexpected findings detailed above and taking into account the fact that a lot of the aging studies have been designed with mice from a single source or even as a longitudinal endeavor, one might speculate to what degree these differences indeed originate from diverging mouse strains within the “Black6” strain rather than from age “alone”. The authors should therefore be rather cautious with the interpretation of their data.
- Minor points:
 - o This reviewer could not find information on the temperature used for the life cell experiments and sample speed of the linescan images.
 - o This reviewer was rather surprised by the excessive Fluo4-AM concentration (10 μ M). For fast confocal imaging myocytes are usually loaded at concentrations < 1 μ M in the reviewer’s lab. Is the Zeiss LSM 880 so inefficient in light collection?

o The statistical bases ought NOT to be the total number of cells but rather the number of animals because only the animals are statistically independent from one another. In this, the authors might also consider adjusting the sample sizes to be more equal and avoid sample size differences of 100%.

Reviewer #3 (Remarks to the Author):

The study by Westhoff, del Villar and colleagues reports on mechanisms of Ca channel and RyR2 regulation in youth and aged mice. I_{Ca,L} and Ca transients (CaT) were elevated in aged cardiomyocytes to a level not different from ISO-treated young cardiomyocytes. ISO of aged cardiomyocytes caused no further change in I_{Ca,L} or CaT. Bin1 knockdown did not alter I_{Ca,L} or CaV1.2 clustering but did alter RyR2 clustering and nanoscale distribution. The authors then measured in vivo heart function by echocardiography. Systolic function is improved from depressed levels in aged mice with Bin1 knockdown. It is unclear how increased basal I_{Ca,L} and CaT lead to depressed in vivo heart function. A probe of myofilament phosphorylation incompletely suggests changes myofilament Ca-sensitivity. Overall, this study presents new information linking Bin1 changes to aging related changes in function. The study has noteworthy findings and would be improved with one of two edits: 1. Provide sufficient sample size in Figure 8 to pose firmer hypothesis linking changes in CaV1.2 and RyR2 to in vivo heart function; and/or 2. Focus on the interesting mechanisms of action that are very well-developed in this study for how Bin1 influences I_{Ca,L} and CaT.

Details and Comments

1.

I_{Ca,L} density and CaT amplitudes are elevated along with CaV1.2 and RyR2 superclustering in baseline conditions for old relative to young. With ISO, amplitudes are not different. By contrast, echocardiography shows EF (FS) is greater in the young. The authors in the Results section clearly attribute the reduced dynamic range for the ISO response to an elevated 'floor.' Thus, it is misleading to repeatedly conclude that ISO responsiveness is reduced. For example, the representative CaT traces appear to exhibit accelerated decay, including for the aged ISO-treated sweep. The authors should report an operational measure of decay kinetics and if faster with ISO in the aged group, a Western blot of PLN-Ser16 is a reasonable index of cellular ISO – responsiveness. Put another way, is the reduced dynamic range to ISO limited (or restricted) to the Ca channel complex in aging?

2.

Related, and as mentioned in Discussion, other groups reported phosphorylation-dependent RyR2 clustering. Did the authors evaluate RyR2-phosphorylation in young, aged and aged with Bin1 knockdown?

3.

Bin1 is over-expressed in aged mice and t-tubule (but not crest) $I_{Ca,L}$ shows an elevated floor, including loss of dynamism. What is the relative t-tubule to crest $I_{Ca,L}$ in young versus aged? Does t-tubule structure change in aging, in this mouse strain, and with cBin1 knockdown? Can these interesting findings be attributed to a change in t-tubule architecture?

4.

What is the relationship between $I_{Ca,L}$ density and ISO responsiveness? Miriyala et al 2008, Circ Res. 102:e54–e64, published evidence for the concept of a functional reserve in the PKA regulation of $I_{Ca,L}$ whereby ISO (PKA) responsiveness is inversely related to $I_{Ca,L}$ density. For a plot of the data in Figures 1 and 5, do all conditions satisfy a single equation as in Miriyala 2008?

5.

For $I_{Ca,L}$, why does ISO cause a $V_{1/2}$ shift but not a fold-change in peak $I_{Ca,L}$ in cells from old mice? The dot plots of $V_{1/2}$ for activation should be added to Figure 1.

6.

What does Bin1 do to t-tubules in young vs old? Is the t-tubule organization in cardiomyocytes from young versus aged mice different?

7.

Figure 6. FS and EF are calculated from the same M-mode measures. Panels f and g are redundant, choose just one. While EDV is reported, it is notable that this measure is a calculation from a linear measure – cubing terms and assumptions of shape can distort interpretation. Please report the LV inner dimension in diastole and the wall thickness measures. Is there evidence for dilatation or wall hypertrophy from these direct measures?

8.

Figure 8. With $N=3$ and the scatter of that data along with inherent noise of quantification from Western blots, it is difficult to interpret the results. Either sample size should be increased or this Figure omitted.

9.

Discussion, p17 (lines 371-385) Paragraph on AD seems out of place and can be deleted.

10.

Discussion, p 18, line 403, "Echocardiography confirms the benefits of Bin1 knockdown ..." reiterates that echocardiography results reflect contractility loss in aging that is reversed by Bin1 knockdown. It is duly noted that mice at room temperature are under enhanced sympathetic tone (relative to humans). Under these conditions, extrapolating from Figures 1-4, we might have extended the common ISO levels in young and aged translating to no in vivo basal difference. This discordance is postulated to be found in myofilament Ca-sensitivity; however, Figure 8 is underpowered and so no conclusions can be reached.

Nevertheless, if there was a significantly reduced myofilament Ca-sensitivity would this be reflected as elevated diastolic Ca?

11.

Discussion. The authors acknowledge the puzzling dichotomous impact of Bin1 levels in prior studies of heart failure in distinction to the present effects on aging. Kudos to the authors for acknowledging but not over-speculating on why these differences in results are found. In this vein, Fu (reference 18) showed increased spontaneous Ca-release with reduced Bin1 levels. Do the authors observe spontaneous Ca-release for any of the models used (young, old, +/- Bin1 knockdown)?

Responses to the Reviewers Comments

We thank the reviewers for their constructive feedback and kind comments on our manuscript. We took these comments and critiques seriously and have revised the manuscript accordingly. In response to reviewer 1's critique we tripled the follow up time post-BIN1 knockdown from the initial two-week time-period featured in the first submission to now include *in vivo* echocardiography data gathered after two, four and six weeks of knockdown. These additional data strengthen our conclusions on the beneficial, long-lived effect of BIN1 knockdown. In a second major addition to the manuscript, this revised submission now includes several new datasets examining the time-course of the response to isoproterenol in old and young mice. These data include electrophysiological perforated patch recordings of I_{Ca} , Ca^{2+} transient recordings, and single molecule localization microscopic analysis of RyR2 and $Ca_v1.2$ localization and clustering. A third important addition is the inclusion of young NIA source matched mice to provide better comparisons to the old NIA mice used in this study. Finally, we used an exon 17-specific antibody as well as a pan-BIN1 isoform antibody to examine the BIN1 isoforms expressed in these animals and in agreement with prior reports from adult mice, we detected four BIN1 isoforms. We also include an analysis of t-tubule organization and an expanded discussion that puts our findings into context in the field. We highlight prior literature, some in agreement with our findings and some not. The inclusion of this new material has strengthened the conclusions of the study and enhanced its potential impact in the field. We hope the paper is now acceptable for publication in Nature Communications. We include here a point-by-point response to each of the issues raised by the three expert reviewers.

Reviewer 1's comments:

1. How confident are you that your interpretation of age-associated changes in cardiomyocyte function are maladaptive? Is it possible that the age-associated differences of clustering of ICaL to the surface membrane occurs is an attempt to compensate for reduced efficacy of beta-adrenergic directed signaling? Is it possible that the LV also adjusts its Ca^{2+} cycling characteristics to adapt to reduction in heartrate that accompanies advancing age? In a well preformed study (Luary-Kleintop et al., J. Cell. Biochem. 116: 2541–2551, 2015) it was reported that that cardiomyocyte-specific loss of Bin1 causes an age-associated dilated cardiomyopathy (DCM) that develops over time beginning by 8–10 months of age, and that younger animals rapidly develop DCM if cardiac pressure overload was created by transverse aortic constriction.

Bin1 loss increased interstitial fibrosis and mislocalization of the voltage-dependent Ca²⁺ channel Cav1.2, and the lipid raft scaffold protein caveolin-3, which normally complexes with Bin1 and Cav1.2 in cardiomyocyte membranes. Thus, this report indicates that cardiac deficiency in Bin1 function causes age- and stress-associated heart failure. These authors suggest that cardiac specific Bin1 causes a novel preclinical model of age-associated terminal cardiac disease rather than rejuvenating the heart.

We are well-acquainted with this landmark study and agree with the reviewer, there is clear evidence that BIN1 deficiency in the heart results in dilated cardiomyopathy and is associated with heart failure. However, our study clearly demonstrates that *too much* BIN1 expression is also detrimental to cardiac function in old mice. In our view, these findings are not in conflict but instead suggest that BIN1 levels must be tightly controlled, and that deviation in expression whether that be too much or too little, has negative consequences for heart health and function. Indeed, a similar conclusion has been drawn in the Alzheimer's disease field where too much or too little BIN1 expression in neurons are each associated with enhanced risk of disease^{1,2}.

It is noteworthy and perhaps serendipitous for us that our shRNA-BIN1 knockdown experiments reduced BIN1 levels in old hearts to young levels but did not shift them below that level (as shown in Fig. 5 d-e (copied here for convenience)). It is conceivable that there could be a critical lower level or threshold for BIN1 expression that we did not cross. Similarly, there could be a critical upper level or threshold of BIN1 expression that when exceeded, results in reduced systolic function. Given that cardiac BIN1 supplementation has been proposed as a therapeutic for heart failure, our findings provide an important cautionary note that warns of a potentially tight therapeutic window for this approach where too much BIN1 supplementation may trigger further deterioration of systolic function and worsen the disease. While beyond the scope of this current work, future studies should measure and define these thresholds.

Figure 5. BIN1 knockdown restores β -AR augmentation of Ca^{2+} transients and RyR2 clustering dynamics. **a**, western blot of BIN1 expression in whole heart lysates from young and old mice probed with 2F11 (top) and 99D (bottom). Total ponceau was used for normalization. **b**, histogram showing normalized BIN1 levels relative to young male for 2F11 and 99D (N = 6, average of 3-6 replicates). **c**, representative Airyscan images of young, old, and old transduced myocytes immunostained against BIN1. **d**, western blot of BIN1 expression in whole heart lysates from: young, old, shRNA-mBIN1 and shRNA-scrmb transduced mice probed with 2F11. Total ponceau was used for normalization. **e**, histogram showing normalized BIN1 levels relative to young (N = 3, average of 1-4 replicates). **f**, representative whole-cell currents from shRNA-scrmb and shRNA-mBIN1 myocytes before and during ISO application. **g**, fold change in peak I_{Ca} with ISO in young, old, shRNA-scrmb (N = 3, n = 7) and shRNA-mBIN1 (N = 3, n = 9) myocytes. **h**, representative Ca^{2+} transients recorded from old shRNA-scrmb and shRNA-mBIN1 myocytes before and after ISO. **i**, fold increase after ISO from young, old, shRNA-scrmb (N = 3, n = 12) and shRNA-mBIN1 (N = 5, n = 13) myocytes. **j**, SMLM localization maps showing $Ca_v1.2$ channel localization on t-tubules of myocytes from old shRNA-scrmb and shRNA-mBIN1, with or without

ISO-stimulation. Regions of interest are highlighted by yellow boxes. **k**, fold change in mean $Ca_v1.2$ channel cluster area with ISO in the old shRNA-scrmb (control: N = 3, n = 14; ISO: N = 3, n = 13) and shRNA-mBIN1 myocytes (control: N = 3, n = 12; ISO: N = 3, n = 11). **l** and **m**, show the same layout for RyR2 immunostained old shRNA-scrmb (control: N = 3, n = 12; ISO: N = 3, n = 13) and shRNA-mBIN1 myocytes (control: N = 3, n = 9; ISO: N = 3, n = 8). Old and young data points in g, i, k and m are reproduced from data in Figs. 1b, 1h, 2b and 2d respectively. Statistical analysis was performed on data in b using unpaired Student's t-tests, and on data in e, g, i, k and m using one-way ANOVAs.

In response to the reviewer's comment, we have expanded our discussion of these ideas to put our findings into context in the field as follows:

"It is important to mention that although our study demonstrates the detrimental effects of BIN1 upregulation on systolic function with aging, prior work has revealed that BIN1 downregulation is associated with heart failure^{3,4} and established that cardiomyocyte-specific loss of BIN1 causes dilated cardiomyopathy⁵. These findings are not in conflict but instead suggest that BIN1 levels must be tightly controlled so that deviation in expression whether that be too much or too little, has negative consequences for heart health and function. In Alzheimer's disease such a model has been presented where too much or too little BIN1 expression are both associated with enhanced risk of disease^{1,2}. Exogenous supplementation of a cardiac specific isoform of BIN1 has thus been proposed as a heart failure therapeutic and has been demonstrated to improve cardiac function in mice with pressure overload-induced heart failure⁶. Our findings provide an important cautionary note that warns of a potentially tight therapeutic window for this approach where too much BIN1 supplementation may trigger further deterioration of systolic function and worsen the disease".

2. Do you suppose that age-associated changes in beta-adrenergic receptor numbers or affinity could be related to trafficking ICaL to the membrane in response to isoproterenol? BAR adaptor proteins are pleiotropic regulators of membrane dynamics and nuclear functions with roles in endocytosis, and in vesicle fusion and trafficking. (c.f. Luary-Kleintop et al., J. Cell. Biochem. 116: 2541–2551, 2015)

We thank the reviewer for this thought-provoking comment. The endosomal pathway disruption we report in aging, likely affects the sarcolemmal expression and recycling of perhaps *all* membrane proteins including β -ARs. New data included in the revised manuscript supports this

idea, showing that transferrin receptor recycling is also significantly slowed in old cells compared to young (see Supplementary Fig. 6).

Supplementary Figure 6. Transferrin receptor recycling is slowed in old myocytes. **a**, representative images of young (left) and old (right) ventricular myocytes incubated with transferrin-Alexa 568 nm for 1 hr (time 0) and after washout (time 30 min). Yellow boxes show regions of interest zoomed-in below the images. **b**, quantification of transferrin-Alexa 568 nm % fluorescence change over a time course in young JAX (N = 3, n = 9), young NIA (N = 3, n = 9) and old (N = 4, n = 13) ventricular myocytes. Area fills indicate SEM.

It has been well documented that reduced β -AR responsiveness in the aging heart occurs despite the presence of enhanced levels of the endogenous activators of these receptors (norepinephrine and epinephrine), implying reduced affinity or desensitization of the receptors (reviewed in ⁷ and ⁸). Given our finding of age-associated endosomal traffic jams, another plausible explanation is impaired β -AR re-sensitization due to recycling deficiencies. If the endocytosis and/or recycling process is impaired, β -ARs may not undergo dephosphorylation and disengagement from β -arrestins rendering them functionally silent even if present at the membrane, and potentially stranded there and/or on endosomes. This could theoretically play a role in the age-dependent hypo-responsivity to β -AR-stimulation that occurs with aging and could be another factor that impairs Cav1.2 regulation by this signaling pathway in aging hearts. We now include this idea as a discussion point in the revised manuscript where we write:

"Another possibility that should be pursued is whether endosomal pathway disruption also affects the sarcolemmal expression and/or recovery from desensitization of β -ARs. Old hearts exhibit reduced expression of β -ARs and impaired recovery from desensitization compared to young^{9,10}. Normally, agonist binding to β -ARs triggers G-protein signaling and subsequent receptor phosphorylation, and recruitment of β -arrestin which halts receptor signaling by reducing functional coupling to heterotrimeric G-proteins^{11,12}. β -ARs are then endocytosed, dephosphorylated, β -arrestin dissociates, they resensitize, and recycle^{11,12}. If this process becomes impaired due to age-associated endosomal traffic jams this would

prolong β -AR desensitization, potentially contributing to the age-dependent hypo-responsivity to β -AR-stimulation."

3. Does "stranding" Cav1.2 at the plasma membrane occur because the channel cannot dissociate from the membrane or the process of recycling back to channels of endosomes is impaired. It would help the reader to briefly define the roles of different RABs in trafficking.

Both endocytosis and recycling of the Cav1.2 biosensor were significantly reduced in old myocytes compared to young as presented in Figure 3 (copied below; see Fig. 3c and d). Visual examination of the TIRF time series images revealed a strikingly large static population of Cav1.2 biosensor in old cells compared to young. In the revised manuscript we include a supplemental video file (Supplemental Movie 1) to provide the reader with a side-by-side comparison of the biosensor dynamics and permit appreciation of this clear difference. Examining the total PM Cav β_{2a} -paGFP revealed an age-dependent increase in the % of Cav1.2 biosensor that remained static in the footprint during the experimental time (Fig. 3e). Accordingly, 53.75 \pm 4.98 % of all the Cav1.2 biosensor detected in the TIRF footprint were static in old myocytes compared to just 33.97 \pm 2.92 % in young.

Figure 3. Dynamic TIRF imaging reveals age-associated trafficking deficits of Cav1.2.

a and **b**, Representative TIRF images of transduced Cav β_{2a} -paGFP young (JAX) (**a**) and old (**b**) ventricular myocytes before (top) and after ISO (bottom). Channel populations that were inserted (red), endocytosed (blue), or static (white) during the ISO treatment are represented to the right. **c** and **d**, dot-plots summarizing the quantification of inserted (**c**) and endocytosed (**d**) Cav β_{2a} -paGFP populations. **e**, is a compilation of the data to show the relative % of the channel pool that

is inserted, endocytosed, and static. **f**, is a summary plot showing the net % change of plasma membrane $\text{Ca}_v\beta_{2a}$ -paGFP after ISO for each cohort of young (N = 3, n = 10) and old (N = 3, n = 10) myocytes. Data were analyzed for statistical analysis using unpaired Student's t-tests.

The large static population are those that we refer to as appearing "stranded". This "stranding" likely occurs due to impaired endocytosis of channels. In support of that idea, we observed a similar stranding phenomenon in a prior study when we treated ventricular myocytes with an actin depolymerizing agent (latrunculin-A)¹³. Actin polymerization is an essential aspect of endocytosis, generating the lateral force to facilitate scission of the endocytic vesicle¹⁴. It seems possible that loss of endocytic processes could be an adaptive response in old myocytes to preserve sarcolemmal $\text{Ca}_v1.2$ in the face of impaired endosomal recycling. Thus, in answer to the reviewer's question, impaired endocytosis and recycling may both contribute to the stranding of $\text{Ca}_v1.2$ channels in old cardiomyocyte sarcolemmas. We have updated the discussion section to clarify these points as follows:

"Our data suggest the latter is true, revealing reduced ISO-induced insertions and endocytosis in old myocytes, an aggregation of channels on swollen endosomes, and a larger pool of static channels that appear stranded in the sarcolemma. Whether this pool of static channels reflects increased lifetime of $\text{Ca}_v1.2$ in the membrane remains to be established but it is reminiscent of our prior finding of an enhanced static population in myocytes treated with the actin depolymerizing agent latrunculin-A¹³. Actin polymerization is an essential aspect of endocytosis, generating the lateral force to facilitate scission of the endocytic vesicle¹⁴. It is possible that loss of endocytic processes is an adaptive response to preserve sarcolemmal $\text{Ca}_v1.2$ in the face of impaired endosomal recycling in old myocytes."

As to the roles of different Rabs in trafficking, we are restricted by the journal's word count allowance but for more details on this we refer the reader to our prior work, " β -Adrenergic control of sarcolemmal $\text{Ca}_v1.2$ abundance by small GTPase Rab proteins" (Del Villar et al PNAS 2021; PMID: 33558236)¹³ as follows:

"As we have previously reported, β -AR activation stimulates the recycling of a portion of the early endosome $\text{Ca}_v1.2$ to the plasma membrane via the Rab4-choreographed fast endosomal recycling pathway¹³."

4. In my opinion, BIN1 Knock down in young mice is also required to fully interpret the results that you report.

On this point, we direct the reviewer toward published studies in which BIN1 has been genetically knocked down or ablated in young mice^{3,5,15,16}. We reference these studies in the manuscript so readers are aware that BIN1 knockdown below the levels normally observed in young mice, is associated with dilated cardiomyopathy and heart failure. Given this prior work, we do not believe that performing this additional set of experiments and time-consuming analyses would change the conclusions of our study.

5. Is the “rejuvenating” effect of BIN1 knock down that you report long lasting? You have only monitored it for 2 weeks. Given the long-term disastrous results observed in BIN1 KO with increasing age (Luary-Kleintop et al., J. Cell. Biochem. 116: 2541–2551, 2015), do you suppose that the “rejuvenating” you report may only be a transient phenomenon?

We thank the reviewer for posing this question as our pursuit of experiments to address this comment has strengthened the conclusions of our paper and improved the study. In the resubmission we include new *in vivo* echocardiography data that tracks cardiac function for six weeks after shRNA-mediated BIN1 knockdown or shRNA-scramble control. In the initial submission we examined cardiac function before and two weeks after shRNA retro-orbital injections. We have now expanded that dataset to include measurements made before, two-weeks after, four weeks after, and six weeks after shRNA retro-orbital injections. Excitingly, we report that the rejuvenating effects of BIN1 knockdown on cardiac function persist for this six-week time-period, with mice that received the shRNA-mBIN1 treatment displaying and retaining a significant improvement in cardiac systolic function over age-matched mice that received the control shRNA-scramble treatment. These expanded datasets feature in Figure 6 (copied below), Figure 7 (copied below), and Supplementary Figures 12-15. To put these findings into context with the results of Luary-Kleintop et al, we refer the reviewer to the comments made in response to their first query above.

Figure 6. BIN1 knockdown improves cardiac contractility in old mice. **a**, representative M-mode echocardiogram images from conscious young and old mice. **b**, summary dot-plots showing fractional shortening (FS) in young (N = 10) and old (N = 30) mice are shown. **c** and **d**, representative M-mode echocardiogram images from conscious old mice before and two, four and six weeks after RO-injection of shRNA-scrmb (**c**) and shRNA-mBIN1 (**d**). **e**, summary dot-plots for FS showing paired results before and two weeks after RO-injection for shRNA-scrmb (N = 14) and shRNA-mBIN1 (N = 14). **f**, summary dot-plots for FS showing paired results before and two, four and six weeks after RO-injection for shRNA-scrmb (N = 5) and shRNA-mBIN1 (N = 5). Statistical analysis was performed on data in **b** using unpaired Student's t-tests, on data in **e** using paired Student's t-tests, and on data in **f** using one-way ANOVAs.

Figure 7. BIN1 knockdown does not recover age-related diastolic dysfunction. **a** and **b**, Representative pulsed-wave (PW) doppler images from unconscious young (**a**) and old (**b**) mice before and two, four and six weeks after RO-injection of shRNA-scrmb (left) and shRNA-mBIN1 (right). Summary dot-plots for young (N = 10) and old (N = 19) mice, paired results before and two weeks after RO-injection of old mice with shRNA-scrmb (N = 10) and shRNA-mBIN1 (N = 9), and paired results before and two, four and six weeks after RO-injection of old mice with shRNA-scrmb (N = 4) and shRNA-mBIN1 (N = 4) for the following diastolic measurements are displayed: **c-e**, isovolumetric relaxation time (IVRT) and, **f-h**, mitral valve E/A ratio (MV E/A). Statistical

analysis was performed on data in c and f using unpaired Student's t-tests, on data in d and g using paired Student's t-tests, and on data in e and h using one-way ANOVAs.

6. *It was recently discovered that PKA dependent phosphorylation of small RGK G-protein Rad (a Ca²⁺ channel inhibitor) is essential for regulation of basal I_{CaL} current and for beta-adrenergic augmentation of Ca²⁺ influx in cardiomyocytes (Papa et al. Nat Cardiovasc Res 2022; 1:1022–1038). 4SA-Rad mice, in which four evolutionarily conserved PKA-phosphorylated serine residues of the endogenous murine Rad locus were replaced by alanine residues, have no response to beta-AR stimulation with ISO. This absence of the increase in Ca²⁺ current in response to ISO in VM from these genetically manipulated 4SA-Rad mice suggests that trafficking and insertion of additional Ca²⁺ channels in plasma membrane after β-AR stimulation might not be a singular mechanism responsible for the augmented Ca²⁺ influx during β-AR stimulation.*

We are familiar with this work from Steven Marx' group and indeed worded our introduction carefully to provide a balanced view. Nowhere do we state that ISO-stimulated recycling of additional Ca_v1.2 channels into the sarcolemma is a singular mechanism responsible for the augmented Ca²⁺ influx during β-AR stimulation. Instead, we cite the Marx Nature paper from 2021 in that section alongside our own work where we reported that cytoskeletal disruption in ventricular myocytes abrogates ISO-stimulated Ca_v1.2 recycling and results in loss of I_{Ca} augmentation.

An alternative explanation for the absence of the increase in Ca²⁺ current in response to ISO in myocytes isolated from 4SA-Rad mice is that Rad phosphorylation and consequent detachment from Ca_vβ subunits is necessary for the augmented trafficking response and this contributes to a large proportion of the dynamic adrenergic increase in Ca²⁺ current. In support of that idea, the reviewer is referred to a well-performed 2007 Circ. Res. manuscript by Yada et al (PMID: 17525370)¹⁷. In that report, adenoviral transduction of cardiomyocytes with a mutant S105N-Rad incapable of Ca_vβ binding resulted in enhanced expression of Ca_vα_{1C} in the sarcolemma. The enhanced *N* and accompanying functional upregulation of the channels contributed to a long QT syndrome phenotype in transgenic mice that expressed S105N-Rad in a cardiac specific manner. These results support a model where both Marx's phospho-Rad regulation model and our trafficking model co-exist and are in fact linked.

A quick literature search reveals many additional prior works in which RGK proteins (the family of small GTP-binding proteins to which Rad belongs) were reported to inhibit Ca_v currents by

interfering with their trafficking and restricting their membrane expression. We refer the reviewer to several of those references listed below. One stand-out work reported that phosphorylation of Rem (another RGK) by PKD, downstream of α 1-adrenergic receptor activation relieved the inhibitory effects of Rem on L-type calcium channels by releasing the brake on their trafficking (Jhun et al. *Circ Res.* 2012; PMID: PMC4232192). Thus, as stated above one may hypothesize that phosphorylation of Rad during adrenergic stimulation may also be an important contributor to the enhanced t-tubular expression of $Ca_v1.2$. The important point here is that our channel recycling/trafficking hypothesis is not at odds with the work of Marx and his group.

For the reviewer's information we include here a selection of works reporting RGK protein disruption of voltage-gated calcium channel trafficking:

- Yang T, Xu X, Kernan T, Wu V and Colecraft HM. Rem, a member of the RGK GTPases, inhibits recombinant $Ca_v1.2$ channels using multiple mechanisms that require distinct conformations of the GTPase. *J Physiol.* 2010;588:1665-81.
- Xu X, Marx SO and Colecraft HM. Molecular mechanisms, and selective pharmacological rescue, of Rem-inhibited $Ca_v1.2$ channels in heart. *Circ Res.* 2010;107:620-30.
- Murata M, Cingolani E, McDonald AD, Donahue JK and Marban E. Creation of a genetic calcium channel blocker by targeted Gem gene transfer in the heart. *Circ Res.* 2004;95:398-405.
- Yada H, Murata M, Shimoda K, Yuasa S, Kawaguchi H, Ieda M, Adachi T, Murata M, Ogawa S, Fukuda K. Dominant negative suppression of Rad leads to QT prolongation and causes ventricular arrhythmias via modulation of L-type Ca^{2+} channels in the heart. *Circ Res.* 2007 Jul 6;101(1):69-77. doi: 10.1161/CIRCRESAHA.106.146399. Epub 2007 May 24. PMID: 17525370.
- Fukuda K. Dominant negative suppression of Rad leads to QT prolongation and causes ventricular arrhythmias via modulation of L-type Ca^{2+} channels in the heart. *Circ Res.* 2007;101:69-77.
- Beguin P, Mahalakshmi RN, Nagashima K, Cher DH, Ikeda H, Yamada Y, Seino Y and Hunziker W. Nuclear sequestration of beta-subunits by Rad and Rem is controlled by 14-3-3 and calmodulin and reveals a novel mechanism for Ca^{2+} channel regulation. *Journal of molecular biology.* 2006;355:34-46.
- Bannister RA, Colecraft HM and Beam KG. Rem inhibits skeletal muscle EC coupling by reducing the number of functional L-type Ca^{2+} channels. *Biophys J.* 2008;94:2631-8.

- Jhun BS, J OU, Wang W, Ha CH, Zhao J, Kim JY, Wong C, Dirksen RT, Lopes CMB and Jin ZG. Adrenergic signaling controls RGK-dependent trafficking of cardiac voltage-gated L-type Ca^{2+} channels through PKD1. *Circ Res.* 2012;110:59-70.

7. *You do not consider the potential role of SERCA regulation in relation to your rejuvenation of the old heart. SERCA2 expression and activity is known to be reduced with aging and has important implications for pumping of Ca^{2+} to the SR and on the SR Ca^{2+} load that is a crucial factor in the release of Ca^{2+} via RyRs. Prior literature has shown that gene therapy can rejuvenate the older heart via enhancing Ca^{2+} cycling (c.f. Roger Hajjar papers).*

The reviewer's point is well taken and while it is true that several studies have reported reduced SERCA expression with aging¹⁸⁻²¹, this is not a universal finding. Several studies have reported unchanged expression of SERCA in the aging heart²²⁻²⁵. Indeed, we are only aware of two published studies in which SERCA expression was examined in the same animal model as us, i.e., C57BL/6 mice, and in both instances, SERCA levels were reported to be unaltered by aging^{24,26}. It is also true that in many instances SERCA/phospholamban balance is affected by aging but that is not a universal finding either²⁴. We are aware of the Hajjar work and his finding that diastolic function can be restored in aging rats by enhancing the buffering capacity of the SR via *in vivo* gene transfer of parvalbumin. We did not see any significant rescue of diastolic function with BIN1 knockdown (**Figure 7**) and since there is no published evidence of BIN1 association with SERCA or influence over its function, we chose not to investigate SERCA expression or function in this study.

8. *Removal of Ca^{2+} from the cytosol is an important factor in the relaxation of the myofilaments, in addition to myofilament Ca^{2+} sensitivity on which you seem to focus, it would be important, to provide the relaxation time or relaxation rate in the transients.*

To address this concern, we now include analysis of the Ca^{2+} transient decay rate from our young and old mouse cohorts to provide a measure of the rate of removal of Ca^{2+} from the cytosol. These data appear in **Supplementary Figure 3** in the revised manuscript. Briefly, contrary to prior findings by some other groups, we report that basal Ca^{2+} transient decay rate was significantly faster than that of young myocytes and little additional acceleration occurred with ISO in contrast to the significant ISO-induced decay acceleration in young cells. Notably, in shRNA-mBIN1

treated cells Ca^{2+} transient decay kinetics retained the old phenotype, remaining at an accelerated level in the transduced cells with and without ISO (**Supplementary Figure 10**). These data reveal that BIN1 knockdown in old myocytes does not rejuvenate Ca^{2+} extrusion rates.

Supplementary Figure 3. Time course of calcium transient response to ISO in young and old myocytes. **a** and **c**, dot-plots showing the fold-increase in Ca^{2+} transient amplitude after 1 min (N = 5, n = 12), 2 min (N = 5, n = 12), 3 min (N = 5, n = 12), and 5 min (N = 5, n = 8) of ISO in young myocytes (**a**) and 1 min (N = 3, n = 10), 2 min (N = 3, n = 10), 3 min (N = 3, n = 10), and 5 min (N = 3, n = 7) of ISO in old myocytes (**c**). **b** and **d**, dot-plots summarizing decay tau during the same time course in young (**b**) and old (**d**) myocytes. Statistical analyses were performed using one-way ANOVAs and post-hoc multiple comparison tests.

Supplementary Figure 10. Further characterization of BIN1 knockdown in old mice. **a**, dot-plots showing peak I_{Ca} in control (left) and ISO (right) conditions for young, old, shRNA-scrambled ($N = 3, n = 7$) and shRNA-mBIN1 ($N = 3, n = 9$) transduced old myocytes. Old and young data points are reproduced from Fig. 1c. **b**, dot-plots summarizing Ca^{2+} transient amplitude in control (left) and ISO (right) conditions from young, old, shRNA-scrambled ($N = 3, n = 12$) and shRNA-mBIN1 ($N = 5, n = 14$) transduced old myocytes. Old and young data points are reproduced from Fig. 1i. **c**, dot-plots showing Ca^{2+} transient decay (ms) in control (left) and ISO (right) from young, old, shRNA-scrambled ($N = 3, n = 12$) and shRNA-mBIN1 ($N = 5, n = 13$) transduced old myocytes. Young and old data points are reproduced from Supplementary Fig. 3b and d. **d**, dot-plots summarizing the mean $Ca_v1.2$ channel cluster areas in control (left) and ISO (right) conditions for young, old, shRNA-scrambled (control: $N = 3, n = 14$; ISO: $N = 3, n = 13$) and shRNA-mBIN1 (control: $N = 3, n = 12$; ISO: $N = 3, n = 11$) transduced old myocytes. Old and young data points are reproduced from Fig. 2b. **e**, dot-plots summarizing the mean RyR2 channel cluster areas in control (left) and ISO (right) conditions for young, old, shRNA-scrambled (control: $N = 3, n = 12$; ISO: $N = 3, n = 13$) and shRNA-mBIN1 (control: $N = 3, n = 9$; ISO: $N = 3, n = 8$) transduced old myocytes. Old and young data points are reproduced from Fig. 2d. Statistical analyses were performed using one-way ANOVAs with multiple comparison post-hoc tests. Young data presented in a is a combination of data from NIA young and JAX young myocytes, and in d and e is from JAX young myocytes.

9. Discrepancies between basal amplitudes of $ICaL$ or Ca^{2+} transients in aged VM observed in the present study and other reports (see above) create concerns how universal and relevant to

aging is the major hypothesis: "that old myocytes exhibit basal super-clustering of both CaV1.2 and RyR2, with no additional dynamic response to ISO" presented in this MS.

Aging is frequently accompanied by complicating age-associated morbidities that can contribute to frailty. This, along with differences in selected ages and animal model likely underlie the large variation observed in aging studies. Read any review on the aging heart and you will find that there is rarely universal agreement on any parameters measured, oftentimes even within species. It is also notable that the cardiac aging field is narrow. This is perhaps because housing mice or any animal until they reach advanced age is time consuming and expensive which makes this somewhat of a niche field. The NIA aged rodent colony makes aging studies more feasible and that is what we have utilized in this study. We do concede though that more studies are necessary to see if this beneficial effect of BIN1 knockdown with aging is true across species. A statement to that effect now appears at the conclusion of the manuscript where we state:

"Future studies should determine whether the cardiovascular benefits of BIN1 knockdown persists across species. This study sets the groundwork for that future work and raises the hope that a similar approach could restore exercise tolerance and capacity to endure acute stress in elderly patients."

10. What is known about aging with respect to phosphorylation, of phospholamban and RyRs at different sites? This would be important with respect to your Ca²⁺ transient measurements. It seems odd that you assessed the phosphorylation of myofilament proteins but not that of Ca²⁺ cycling proteins. In order to ascertain myofilament Ca²⁺ sensitivity, some measure of contraction is required. I don't seem to find this information in your report.

To our knowledge, there is little published information on the phosphorylation state of phospholamban (at Ser17 or Thr16 sites) and RyR2 (at the S2808 or S2814 sites) in aging hearts/ventricles. One paper that we now cite in the revised manuscript reports phospholamban (PLB) phosphorylation is enhanced in myocytes from old mice compared to young²⁷. If there is enhanced PLB phosphorylation this would be expected to enhance transient decay rates assuming no change in SERCA2a expression, and could explain the enhanced transient decay rates we observe in old versus young myocytes. With regards to RyR2, we found just one paper examining RyR2 phosphorylation state with aging in female rabbit ventricles and they reported no

age-dependent change in expression of RyR2-S2809 or RyR2-S2815 (the rabbit CaMKII- and PKA-phospho sites on RyR2)²².

Based on our observations of enhanced age-dependent clustering of RyR2 and Ca_v1.2, we predict that RyR2 are likely to exhibit enhanced phosphorylation, given that prior work has established that phosphorylation of these channels leads to enhanced cluster sizes (see response to comment 11 below).

[Redacted]

Furthermore, I_{Ca} recorded from old myocytes displays a significant basal leftward shift compared to currents recorded from young myocytes (Figure 1e). This leftward shift is a fingerprint of PKA-phosphorylation of Ca_v1.2 channels and supports the idea that they are more phosphorylated at rest in old cells. Future studies should directly examine the phosphorylation state of these channels and other relevant proteins including PLB but we consider this beyond the scope of the present work.

In the revised manuscript we now state:

"Ca²⁺ transient decay rate is an indicator of the effectiveness of Ca²⁺ extrusion mechanisms, mainly SERCA2a, that reinstate Ca²⁺ gradients between beats. PKA-mediated phosphorylation of phospholamban relieves an inhibitory brake on SERCA2a and accelerates Ca²⁺ extrusion and lusitropy during β-AR stimulation. Accordingly, in

young myocytes ISO significantly accelerated T_{decay} (Supplementary Fig. 3b) within 2 mins of application however old myocytes exhibited an already enhanced T_{decay} that displayed little change with ISO (Supplementary Fig. 3d). These results suggest that old myocytes may exhibit enhanced basal phosphorylation of phospholamban as has been reported by others²⁷, and further support a reduced β -AR responsiveness in old hearts and diminished myocardial responses to acute stress."

11. *Does the iso effect to increase cardiomyocyte performance require increased phosphorylation of ICaL and RyR to the membrane in order to effectively affect their trafficking? Is de-phosphorylation required to remove ICaL from the surface membrane and return the RyR Cluster to the pre-iso level? These sorts of issues seem to be important and require some information on the activities of kinases and phosphatases in the aging myocardium.*

Excellent point. Our group has previously reported that the ISO-stimulated augmentation of Ca_v1.2 clustering is a PKA-dependent phenomenon²⁸. We showed this in SMLM experiments where the ISO-induced super-clustering effect was abolished in ventricular myocytes pre-treated with the PKA inhibitors H-89 (10 μ M) or PKI (5 μ M). For the reviewer's convenience, this previously published data is **included below**. In that and subsequent work¹³ from our group, including the present study, we have established that the ISO-stimulated superclustering of Ca_v1.2 occurs due to enhanced trafficking of these channels from an endosomal reservoir. Thus, in answer to the reviewer's question, increased phosphorylation is required to effectively affect Ca_v1.2 channel trafficking to the sarcolemma. Preventing their phosphorylation by inhibiting PKA precludes this effect. Similarly, other groups have shown that phosphorylation of RyR2 triggers their enhanced clustering on the junctional SR^{29,30} while pharmacological inhibition of phosphorylation via either CaMKII or PKA was found to reverse the effects³⁰. Accordingly, phosphorylation-triggered mobilization of Ca_v1.2 and RyR2 into larger clusters is well-established. In our revised manuscript we now include time courses of the clustering responses to ISO for both RyR2 and Ca_v1.2. We did attempt to washout ISO (1hr) to determine whether this would result in cluster dispersal/reversal of super-clustering but this led to problems with cell adhesion to the coverslip and thus we were unable to retain cells for imaging after washout. We also attempted to treat cells with ISO for 8 mins followed by 1 hr incubation with the PKA inhibitor PKI but encountered the same issues.

Figure 2 ISO-induced ‘super-clustering’ of $Ca_v1.2$ is mediated by PKA. **(A)** Diffraction-limited TIRF (left column) and GSD (right column) images of a control (top row) or 100 nM ISO-stimulated (bottom row), fixed, adult mouse ventricular cardiomyocyte immunostained to examine $Ca_v1.2$ channel distribution. Images were pseudocolored ‘red hot’ and received a 1 pixel median filter for display purposes. Yellow boxes indicate the location of the zoomed-in regions displayed on far right. **(B and C)** The same layout format in myocytes pretreated with 10 μ M H-89 **(B)** or 5 μ M PKI **(C)**. **(D)**: Aligned dot plot showing mean $Ca_v1.2$ channel cluster areas in control (black circles) and ISO-stimulated (blue circles) myocytes, in H-89 pre-treated myocytes under control (black squares) or ISO-stimulated (blue squares) conditions, and in PKI pre-treated myocytes control (black triangles) or ISO-stimulated (blue triangles) conditions. Red lines indicate the mean for each dataset and error bars indicate the SEM. Figure reproduced from Ito et al. *J Physiol.* 2019²⁸ under the terms of the Creative Commons Attribution 4.0 International Public License (CC-BY-NC 4.0; <https://creativecommons.org/licenses/by/4.0/>) associated with that article.

12. would add that I could not find the time of application of ISO that caused remodeling of Cav and Ryr clusters. Do they really move in 2 minutes? "Recordings were obtained in control conditions and after 2-3 mins of perfusion with 60 100 nM isoproterenol (ISO; Sigma-Aldrich) in the external bath solution." Why specifically 2-3 min? Neither a reference, nor an example of time course is given. Depending on the time course 2 or 3 minutes could be a big difference.

The revised methods and figure legends have been updated to include the requested information. In answer to the question "Do they really move in 2 minutes?" the answer is yes for both Cav1.2 channels and RyR2 (see **Supplementary Figure 4**). In the resubmission we now include time courses of the response to ISO as measured in: a) perforated patch recordings of I_{Ca} (**Supplementary Figure 2**); b) Ca^{2+} transient recordings (**Supplementary Figure 3**); c) SMLM experiments examining Cav1.2 channel localization and clustering (**Supplementary Figure 4a-b**); and d) SMLM experiments examining RyR2 localization and clustering (**Supplementary Figure 4c-d**). Since we only observed a significant difference in the cluster size of Cav1.2 and RyR2 in young (not old) myocytes at the 8 min ISO treatment timepoint (**Figure 2**), we focused on examining the intervening timepoints (1 min, 2 min, 3 min, and 5 min) in young myocytes finding that the superclustering response became significant at the 2 min timepoint and reached a plateau thereafter. Likewise, the I_{Ca} and Ca^{2+} transient responses to 100 nM ISO became significant at 2 - 3 mins and plateaued thereafter (**Supplemental Figures 2 and 3**).

Supplementary Figure 2. Time course of I_{Ca} response to ISO in young and old myocytes. a, representative perforated patch clamp currents elicited from young (left) and old (right) ventricular myocytes before (control; black traces) and during application of ISO (blue traces). **b**, diary plot of normalized I_{Ca} density for young ($N = 3$, $n = 6$) and old ($N = 3$, $n = 8$) myocytes. **c** and **d**, dot-plots summarizing the time course of the fold-change in I_{Ca} with ISO for young (**c**) and old (**d**) myocytes. Statistical analyses on data in **c** and **d** were performed using one-way ANOVAs with post-hoc multiple comparison tests.

Supplementary Figure 3. Time course of calcium transient response to ISO in young and old myocytes. **a** and **c**, dot-plots showing the fold-increase in Ca^{2+} transient amplitude after 1 min ($N = 5$, $n = 12$), 2 min ($N = 5$, $n = 12$), 3 min ($N = 5$, $n = 12$), and 5 min ($N = 5$, $n = 8$) of ISO in young myocytes (**a**) and 1 min ($N = 3$, $n = 10$), 2 min ($N = 3$, $n = 10$), 3 min ($N = 3$, $n = 10$), and 5 min ($N = 3$, $n = 7$) of ISO in old myocytes (**c**). **b** and **d**, dot-plots summarizing decay tau during the same time course in young (**b**) and old (**d**) myocytes. Statistical analyses were performed using one-way ANOVAs and post-hoc multiple comparison tests.

a SMLM of Ca_v1.2

c SMLM of RyR2

Supplementary Figure 4. Time course of ISO-stimulated Ca_v1.2 and RyR2 clustering in young myocytes. **a**, SMLM localization maps showing Ca_v1.2 channel localization and distribution in the t-tubules of young myocytes treated with 0, 1, 2, 3, 5 and 8 minutes of ISO. **b**, dot-plots summarizing the mean Ca_v1.2 cluster areas in young myocytes for the various time points (0 min ISO: N = 3, n = 22; 1 min ISO: N = 3, n = 15; 2 min ISO: N = 3, n = 20; 3 min ISO: N = 3, n = 18; 5 min ISO: N = 3, n = 17; 8 min ISO: N = 3, n = 22). **c** and **d**, show the same layout for RyR2 cluster areas in young myocytes (0 min ISO: N = 3, n = 15; 1 min ISO: N = 3, n = 16; 2 min ISO: N = 3, n = 16; 3 min ISO: N = 3, n = 18; 5 min ISO: N = 3, n = 14; 8 min ISO: N = 3, n = 15). Statistical analyses were performed using one-way ANOVAs and post-hoc multiple comparisons tests.

Figure 2. Basal super-clustering, impaired β -AR responsiveness, and enhanced proximity of Ca_v1.2 and RyR2 in aged myocytes. **a**, SMLM localization maps showing Ca_v1.2 channel localization and distribution in the t-tubules of young and old ventricular myocytes with or without ISO-stimulation. Yellow boxes indicate the location of the regions of interest magnified in the top right of each image. **b**, dot-plots showing mean Ca_v1.2 channel cluster areas in young (control: N = 3, n = 9; ISO: N = 3, n = 9) and old (control: N = 4, n = 11; ISO: N = 4, n = 10) myocytes. **c** and **d**, show the same for RyR2 in young (control: N = 3, n = 17; ISO: N = 3, n = 11) and old (control: N = 3, n = 12; ISO: N = 3, n = 11) myocytes. **e**, Representative fluorescence PLA (red)/DAPI (blue) images of myocytes with and without ISO-stimulation. **f**, dot-plot summarizing the cellular density of PLA fluorescent puncta normalized to the density in young control cells (young control: N = 3, n = 19; young ISO: N = 3, n = 16; old control: N = 3, n = 14; old ISO: N = 3, n = 16). Statistical analyses on data summarized in **b**, **d**, and **f** were performed using two-way ANOVAs with multiple comparison post-hoc tests.

13. Line 112 in results: a maximal ISO-stimulated current was referred to; you report only 1 concentration in your result (100 nmol). Had you performed a dose response to isoproterenol to determine the concentration at which the maximal response is achieved, and whether this differs by age? Numerous prior studies have indeed demonstrated that sensitivity for maximum response to β -adrenergic receptor agonists (e.g. mixed β -1 and β -2 agonists, like isoproterenol, or β -1 agonist in the presence of α -adrenergic receptor blockade differ as age advances).

Good point, we have rephrased this sentence to remove any ambiguity. It now reads:

The maximal current amplitude stimulated with 100 nM ISO was similar in young and old myocytes (Fig. 1c and d), suggesting that the “ceiling” was intact but that the “floor” or baseline shifted with aging leaving less scope for current augmentation with acute β -adrenergic stress.

Our rationale for using a single, 100 nM concentration of isoproterenol (ISO) is two-fold. Firstly, we wanted to imitate the physiological activation of this pathway which is initiated by ligand binding to the receptors. Arguably 100 nM is already a supra-physiological dose, as in adult human ventricular myocyte sarcomere shortening experiments ISO reportedly has an EC₅₀ of ~10 nM (see PMID: 32376974). Thus, by using 100 nM ISO we are already demonstrating the receptors are less responsive. Secondly, using a potent adenylyl cyclase activator like forskolin invites the critique that one is eliciting a large global response that occurs independently of the receptors and their localization. We would suggest that localization matters and thus, while we may see more channel phosphorylation with forskolin, deciphering what that would mean in the physiological setting of aging would be challenging.

We also wish to clarify that the idea that β -adrenergic hypo-responsivity occurs with aging is not something we conceived. This is well-established in the field of cardiac aging as demonstrated by several of the references we cite in the manuscript and listed below for the reviewer's convenience:

- White M, Roden R, Minobe W, Khan MF, Larrabee P, Wollmering M, Port JD, Anderson F, Campbell D, Feldman AM and et al. Age-related changes in β -adrenergic neuroeffector systems in the human heart. *Circulation*. 1994;90:1225-38.
- Lakatta EG. Deficient Neuroendocrine Regulation of the Cardiovascular-System with Advancing Age in Healthy Humans. *Circulation*. 1993;87:631-636.
- Stratton JR, Cerqueira MD, Schwartz RS, Levy WC, Veith RC, Kahn SE and Abrass IB. Differences in cardiovascular responses to isoproterenol in relation to age and exercise training in healthy men. *Circulation*. 1992;86:504-12.
- Davies CH, Ferrara N and Harding SE. β -adrenoceptor function changes with age of subject in myocytes from non-failing human ventricle. *Cardiovascular research*. 1996;31:152-6.

- Lakatta EG, Gerstenblith G, Angell CS, Shock NW and Weisfeldt ML. Diminished inotropic response of aged myocardium to catecholamines. *Circ Res.* 1975;36:262-9.
- Xiao RP, Tomhave ED, Wang DJ, Ji X, Boluyt MO, Cheng H, Lakatta EG and Koch WJ. Age-associated reductions in cardiac β 1- and β 2-adrenergic responses without changes in inhibitory G proteins or receptor kinases. *J Clin Invest.* 1998;101:1273-82.
- Xiao RP, Spurgeon HA, O'Connor F and Lakatta EG. Age-associated changes in β -adrenergic modulation on rat cardiac excitation-contraction coupling. *J Clin Invest.* 1994;94:2051-9.
- Cerbai E, Guerra L, Varani K, Barbieri M, Borea PA and Mugelli A. β -adrenoceptor subtypes in young and old rat ventricular myocytes: a combined patch-clamp and binding study. *Br J Pharmacol.* 1995;116:1835-42.
- Strait JB and Lakatta EG. Aging-Associated Cardiovascular Changes and Their Relationship to Heart Failure. *Heart Failure Clinics.* 2012;8:143-164.

Notably, in Xiao *et al.* *JCI* 1994, similar effects on $\text{Ca}_v1.2$ channel responsivity were reported using an alternative β -agonist, namely norepinephrine. There they used a range of concentrations of that agonist from 10^{-8} M to 10^{-6} M and stated:

“...the efficacy of beta AR stimulation to increase I_{Ca} was significantly reduced with aging”.

14. Please provide time course (with all statistics) to justify the 2-3 min of measurements of I_{CaL} and transients. It's well known that kinetics of most reactions become slowed with aging, and often time the same maximum could be achieved but it takes a longer time. It's possible that the time course for old and young animals might be different. The same problem might be with assessment of clustering. The time course of clustering also differ by age could be also different and when to assess the clustering could be important. There is no assessment of SERCA function that could be also important for data interpretation

The time course data requested is included in the revised manuscript. We do not see any evidence that the old mouse myocytes were simply responding with slower kinetics compared to

the young mice. Instead, we see that the I_{Ca} and transient responses plateau after 2-3 mins treatment with 100 nM ISO. Please see the responses to comments 7, 8 and 12 above.

15. Although voltage clamp is the classical method to inquire about I_{CaL} amplitude, Action Potential shapes appear to differ with aging, and this might result in age-associated differences in I_{CaL} that are not evident in 300 msec voltage steps. It would have been instructive to measure the action potential characteristics in your isolated cardiac myocytes.

The reviewer's point is well-taken, however given the numerous lines of evidence in favor of our hypothesis in the current study, we do not believe that performing this additional set of experiments and time-consuming analyses would change the conclusions of our study.

16. In a related issue, the amplitude and characteristics of the cytosolic Ca²⁺ transient and contraction may differ by age with respect to the pacing frequency employed in experiments; and this effect of pacing rate also differs in the presence and absence of isoproterenol and may also differ by age. The temperature at which the experiments were performed is also an important factor, with respect to the biophysical measurements you performed. I couldn't seem to find information about the temperature at which the in situ cardiac myocytes studies were conducted. Given these uncertainties, how confident are you that you have captured the "reality" of age-associated changes in your isolated cell experiments?

In the SMLM experiments, we performed the 100 nM ISO and control incubation periods at 37 °C thus maintaining physiological temperature, albeit in unpaced cells. I_{Ca} and Ca^{2+} transients were performed at room temperature as is standard in the field. As the reviewer correctly points out, in any study where one isolates a cell from its native environment disrupting connections with neighboring cells, one gets an imperfect view of those cells that reflects their new isolated reality. We believe readers recognize that caveat of this and most other electrophysiological and functional studies. One of the strengths of our study is that it includes datasets ranging from the nanoscale all the way up to the intact animal. We are certain that we have "captured the reality of the age-associated changes" in the *in vivo* echocardiography and doppler imaging as here the hearts are being internally paced at physiological temperature and there you can see clear age-associated differences in systolic and diastolic function. Our other datasets are vertically

integrated with this *in vivo* data, and the conclusions are complementary and support the age-associated changes we observe in the isolated cell experiments.

17. *With respect to in vivo experiments, a major premise on which you base your study, that aging is associated with a reduced response to stress, e.g. reduced responsiveness to BAR signaling, that renders the heart more vulnerable to stress. Your in vivo ECG recording certainly must be a stress on the conscience mouse. Because the rationale is that in advanced age the response to stress is reduced, renders the interpretation of the results of the in vivo ECG in conscious mice might be difficult. Specifically, the response to stress is greater in young than older mice, and this increased stress response elevates HR and contractility to a greater extent in young vs old mice. By eyeballing the figures in conscious vs. unconscious state, seems to indicate that the HR difference in the conscious vs. unconscious states, in HR for example, is greater in young than in old mice. In my opinion, an analysis of variance of the echo results for main effects of consciousness and age, and age/consciousness interactions is required.*

The reviewer's point is well taken. It is well known that performing echocardiograms is stressful to conscious mice it is equally well known that anesthetizing mice to obtain unconscious recordings suppresses sympathetic activity. These effects likely explain the enhanced heart rates and fractional shortening (FS) seen in young conscious (likely stressed) mice versus young unconscious (sympathetically suppressed) mice but do not explain the differences when comparing young to old mice who were both subjected to the same conscious and unconscious recordings. Importantly, BIN1 knockdown in old mice significantly rejuvenated FS toward youthful levels in both conscious and unconscious mice thus the beneficial effects of BIN1 knockdown are quite clear in these data sets.

18. *Line 277-278: you state the resultant increase in preload in old mice would be expected to enhance ventricular output via the Frank-Starling mechanism to maintain stroke volume in the face of reduced contractility. It seems that enhance and maintain are contradictory, which one do you propose is correct? Because your HR did not change in response to BIN1 Knockdown, any enhancement of cardiac output must occur via an increase in stroke volume. Because the cardiac output increased following BIN1 knockdown in old mice, the Frank-Starling mechanism would not only have to maintain stroke volume but would have to increase stroke volume. I don't see any*

stroke volume data in your results, what was the stroke volume in young and old and following BIN1 knockdown, and did this differ in the conscious vs. non-conscious states.

This statement has been removed from the revised manuscript along with all reported measurements of volume (including EF, EDV, ESV, SV, and CO) since as Reviewer 3 correctly points out, these measures involve, "calculation from a linear measure – cubing terms and assumptions of shape can distort interpretation". We thus decided to report only direct measures in our resubmission.

19. Line 394: One interpretation of your results is that the heart reserve capacity is lost with aging because it is already expended at rest. It seems to me that if reserve capacity is lost, the body would increase sympathetic input to the heart in an attempt to maintain homeostasis of cardiac output that is essential for body organ needs. This increase in sympathetic tone could be the cause of increased trafficking of I_{CaL} to the membrane. In this case, treatment of BAR blockade, might reduce the excessive baseline I_{CaL} trafficking to the surface membrane.

This is an astute point from the reviewer and indeed we believe this is likely why the reserve capacity is already expended at rest. We will report this in a second manuscript currently in preparation.

[Figure Redacted]

Minor

1. Lines 133-134: "Increased sarcolemmal Ca_v1.2 channel expression and clustering at rest in aging cells could explain the age-dependent increase in basal I_{Ca}." Did you study expression of Cav1.2 protein in the aged VM?

We did not perform surface biotinylation experiments to quantify the sarcolemmal Ca_v1.2 channel expression in young and old cells however this statement still stands as we simply suggest that this could be a potential explanation for the enhanced basal I_{Ca}. Later in the paragraph in question, we detail SMLM experiments that support the idea that aging VMs have increased sarcolemmal Ca_v1.2 channel expression and clustering compared to young cells.

For a direct link between superclustering and electrophysiology we refer the reviewer to prior published work from our group (Ito et al. J. Physiol. 2019; **PMID: 30714156**) where superclustering of Ca_v1.2 channels was found to facilitate cooperative gating between adjacent channels within the cluster and to promote amplification of calcium influx. In that publication we demonstrated this link between the nanoscale arrangement of Ca_v1.2 and channel activity with multiple techniques including single channel electrophysiology, whole cell patch clamp, super-resolution microscopy, and stepwise photobleaching. The idea that ion channel clustering can have functional consequences and implications for the efficiency of Ca²⁺ release in cardiomyocytes is an emerging concept but one that is supported by an accumulating body of literature. For a recent review on the topic, we refer the reviewer to **PMCID: PMC8769284**. There the reviewer will find many examples from groups including that of Bill Louch, Edwin Moore, Don Bers, Daisuke Sato, and Clive Orchard that support the idea that clustering of cardiac Ca_v1.2 channels and RyR2 channels is altered by phosphorylation and by heart failure, with profound functional consequences.

2. Line 384: It would be helpful to the reader to list a few of these proteins.

We are unfortunately limited by strict word count restrictions Nature Communications which is why we decided to refer the reader to the following review manuscript, stating:

"Several other proteins have been implicated as sources or contributors to endosomal dysfunction and swelling in AD neurons as reviewed".

Small, S. A., Simoes-Spassov, S., Mayeux, R. & Petsko, G. A. Endosomal Traffic Jams Represent a Pathogenic Hub and Therapeutic Target in Alzheimer's Disease. *Trends Neurosci* **40**, 592-602, doi:10.1016/j.tins.2017.08.003 (2017).

For the reviewer's benefit, the "other proteins" include but are not limited to SORL1, retromer complex proteins, PICALM, APOE4, CD2AP, ABCA7 and BACE1.

3. Line 450: to which myosin heavy chain do you refer? This may be an important consideration as there is a switch from α to β myosin heavy chain in rodents as age advances?

This statement does not refer to our work but to a published manuscript from another group. That manuscript does not specify the heavy chain but does refer to using an anti-myosin heavy chain antibody called MF20. A search for this reveals it is an antibody that is specific for type II myosin (which is a larger complex that contains β -myosin heavy chain). This myosin heavy chain is the dominant isoform in fetal hearts and is an established marker of aging where it becomes re-expressed.

4. Figure 5 legend, line 743: "h" should be replaced by "k".

Thanks. Updated.

Reviewer #2 (Remarks to the Author):

1. *After several careful readings of the ms this reviewer has identified the following criticisms:*

- *This reviewer was surprised by the following findings that contradict most (>80%) of the previous reports on the aging heart including:*

- o Increased Ca transient amplitude by more than 100%. Reports usually depict decreases or constant amplitudes.*
- o Increased speed of Ca removal. Reports usually depict a slowdown of the decay phase of Ca transients.*
- o Loss of EF by 50%. Reports usually depict slight decreases or constant EF or FS.*

The reviewer's point is well-taken and we agree that the mentioned findings are different from the majority (but not all) prior work in the field. We go to lengths in the discussion to put our findings in context with the rest of the cardiac aging field where we acknowledge:

"Prior studies have shown that basal $Ca_v1.2$ activity and β -AR-mediated I_{Ca} enhancement are altered with aging³¹. However, the effects on basal $Ca_v1.2$ currents appear to vary depending on sex, species, and possibly even strain with various groups reporting enhanced^{26,31,32}, unchanged³³, or reduced³⁴⁻³⁶ currents in aged models. In our hands, myocytes from ~24-month-old C57BL/6 mice had enhanced basal $Ca_v1.2$ currents compared to young (~3-month-old). Old myocytes also exhibited basally enhanced Ca^{2+} transient amplitudes. Prior findings on Ca^{2+} transient alterations in aging cardiomyocytes range from basal amplitude augmentation^{32,37} and faster decay kinetics³⁸ to no or little reported change³⁶, to basal reduction in Ca^{2+} transient amplitude and slower rate of decay^{26,39}. This echoes the variability in age-associated alterations of I_{Ca} and reflects the complexity of aging. Importantly though, and in line with our results, there is agreement that the ability of β -AR-signaling to augment Ca^{2+} transient amplitude and tune EC-coupling and inotropy is diminished with aging^{9,40-46}.

While we have now removed the EF parameter at the request of Reviewer 3 (see their 7th comment) and instead focus on FS measurements, the reviewer's point remains well taken and understood. With regards to prior findings on systolic function and how our work aligns/misaligns to those, we state:

"Echocardiography confirmed the benefits of BIN1 knockdown on the aging heart with systolic function substantially improved after just a two-week knockdown of BIN1. In humans, resting systolic function is generally preserved with aging, however it becomes significantly impaired during exercise and acute stress⁴⁷⁻⁴⁹. The resting systolic dysfunction observed in the current study may reflect the difference between human and mouse heart's basal autonomic balance at room temperature. Mice have relatively more sympathetic nervous system activity at rest than humans. With aging, autonomic balance

becomes even more tilted toward sympathetic activity as parasympathetic input appears reduced^{50,51}. In this way, a resting mouse heart is not necessarily a fair comparison to a resting human heart and may instead be a better reflection of an exercising human heart. Thus, BIN1 knockdown may present a therapeutic option to enhance systolic function in aging humans during acute exercise and stress".

Prior work using the same aging animal model as us (C57BL/6 mice) reported a significant (22 %) reduction in FS with aging in this strain of mouse but preserved systolic function in DBA/2⁵².

Additionally, with permission I can reveal that in personal communications with Dr. Ed Lakatta, the Chief of the Laboratory for Cardiovascular Science at the National Institute on Aging and world-renowned expert in cardiovascular aging, his group also observes a significant age-associated decrease in EF and FS in C57BL/6 mice and plan to report those findings in an upcoming longitudinal study.

2. BIN1's primary RNA is well known to undergo extensive splicing with splice variants associated to different functions of the particular BIN1-isoform. The authors appear to neglect this fact by solely analyzing full length BIN1.

We thank the reviewer for this helpful comment. Full length BIN1 contains 20 exons, and the reviewer is correct, there are greater than ten known splice variants of BIN1 based on alternate combinations of exons⁵³. Of those variants, only four have been identified in the mouse heart^{15,54}. In response to the reviewers very valid comment, we have now probed western membranes with 2F11 anti-BIN1 (considered a pan-BIN1 antibody although it will not detect the exon 7 containing neuronal isoform) and with 99D anti-BIN1 which detects the smaller number of isoforms that contain exon 17⁵⁵. Using this approach, we have been able to identify at least three of the four BIN1 variants known to be expressed in mouse hearts. Two of these variants have extremely similar molecular weights thus we believe the middle of the three bands we observe is likely a combination of BIN1+13 and BIN1+17. In the revised manuscript this section has been updated as follows:

" Endosomal enlargement has been linked to endosomal dysfunction and deficiencies in membrane protein recycling in Alzheimer's disease (AD)⁵⁶. Endosome swelling results from an imbalance in cargo coming into and leaving endosomes, sometimes referred to

as endosomal traffic jams⁵⁷. BIN1/amphiphysin II has been implicated as playing a role in the development of endosomal traffic jams in AD-stricken neurons⁵⁷. In the heart, BIN1 is better known for its role in targeted delivery of Cav1.2^{4,16}, t-tubule biogenesis⁵⁸, micro-folding¹⁵, and maintenance³, and in dyad formation⁵⁹. However, in neurons, where there are no t-tubules, BIN1 is known to play a role in mediating endosomal membrane curvature and tubule formation required for cargo exit and recycling from endosomes^{60,61}. BIN1 overexpression has been linked to aging and neurodegeneration in the brain^{1,62,63} and has been seen to produce endosomal expansion and traffic jams^{57,60,61}. Furthermore, BIN1 expression levels are known to affect ion channel trafficking in both neurons⁶⁴ and ventricular myocytes^{4,16}. We thus examined BIN1 expression and localization in young and old myocytes and found an age-associated upregulation (Fig. 5a-b) and redistribution of BIN1 with old myocytes exhibiting a vesicular, endosomal pattern of BIN1-staining alongside the expected t-tubule population (Fig. 5c; left). Four BIN1 splice variants are known to be expressed in the hearts of young mice^{15,54} (Supplementary Fig. 8a). We observed each of those variants in young and old hearts. Probing western blots with a pan-BIN1 antibody (2F11) revealed three bands in young and old mice and based on their molecular weight these are believed to represent BIN1 (~48 kDa), a combination of BIN1+13 and BIN1+17 (~58 kDa), and BIN1+13+17 (~65 kDa) (Supplementary Fig. 8b). Probing with an exon 17 specific anti-BIN1 (99D) revealed the presence of two exon 17 containing isoforms, surmised to be BIN1+17 (~58 kDa) and BIN1+13+17 (~65 kDa) (Supplementary Fig. 8c). Unfortunately, the relatively high BIN1 expression levels in old lysates made it impossible to capture well-separated bands for young and old mice using our approach and thus the relative isoform expression profile of each cohort could not be defined".

The relevant two figures **Figure 5** and **Supplementary Figure 8** are copied below for the Reviewer's convenience.

Figure 5. BIN1 knockdown restores β -AR augmentation of Ca^{2+} transients and RyR2 clustering dynamics. **a**, western blot of BIN1 expression in whole heart lysates from young and old mice probed with 2F11 (top) and 99D (bottom). Total ponceau was used for normalization. **b**, histogram showing normalized BIN1 levels relative to young male for 2F11 and 99D ($N = 6$, average of 3-6 replicates). **c**, representative Airyscan images of young, old, and old transduced myocytes immunostained against BIN1. **d**, western blot of BIN1 expression in whole heart lysates from: young, old, shRNA-mBIN1 and shRNA-scrmb transduced mice probed with 2F11. Total ponceau was used for normalization. **e**, histogram showing normalized BIN1 levels relative to young ($N = 3$, average of 1-4 replicates). **f**, representative whole-cell currents from shRNA-scrmb and shRNA-mBIN1 myocytes before and during ISO application. **g**, fold change in peak I_{Ca} with ISO in young, old, shRNA-scrmb ($N = 3$, $n = 7$) and shRNA-mBIN1 ($N = 3$, $n = 9$) myocytes. **h**, representative Ca^{2+} transients recorded from old shRNA-scrmb and shRNA-mBIN1 myocytes before and after ISO. **i**, fold increase after ISO from young, old, shRNA-scrmb ($N = 3$, $n = 12$) and shRNA-mBIN1 ($N = 5$, $n = 13$) myocytes. **j**, SMLM localization maps showing $Ca_v1.2$ channel

localization on t-tubules of myocytes from old shRNA-scrmb and shRNA-mBIN1, with or without ISO-stimulation. Regions of interest are highlighted by yellow boxes. **k**, fold change in mean $Ca_v1.2$ channel cluster area with ISO in the old shRNA-scrmb (control: $N = 3$, $n = 14$; ISO: $N = 3$, $n = 13$) and shRNA-mBIN1 myocytes (control: $N = 3$, $n = 12$; ISO: $N = 3$, $n = 11$). **l** and **m**, show the same layout for RyR2 immunostained old shRNA-scrmb (control: $N = 3$, $n = 12$; ISO: $N = 3$, $n = 13$) and shRNA-mBIN1 myocytes (control: $N = 3$, $n = 9$; ISO: $N = 3$, $n = 8$). Old and young data points in **g**, **i**, **k** and **m** are reproduced from data in Figs. 1b, 1h, 2b and 2d respectively. Statistical analysis was performed on data in **b** using unpaired Student's t-tests, and on data in **e**, **g**, **i**, **k** and **m** using one-way ANOVAs.

Supplementary Figure 8. BIN1 isoforms present in the heart. **a**, diagram illustrating the domains and exons present in full length BIN1 as well as the four isoforms present in the heart (BIN1, BIN1+13, BIN1+17, BIN1+13+17) with their respective predicted molecular weights. Antibody epitopes for 2F11 and 99D antibodies are shown on the bottom. **b**, example western blot showing the presence of three bands with 2F11, corresponding to BIN1, BIN1+13 and BIN1+17 combined, and BIN1+13+17. **c**, example western blot showing the presence of two bands with 99D, corresponding to BIN1+17 and BIN1+13+17.

3. Loss of BIN1 expression has been reported to cause substantial T-tubular disarray and uncoupling of Ca channels in the plasma membrane and the RyR clusters in the SR membrane. In this it would be very important to analyze distribution and morphology of the T-tubular system

and coupling efficiency of EC-coupling in the young and old hearts as well in the cells with downregulated BIN1 expression.

To address this concern, we examined the t-tubule network in isolated young, old, shRNA-mBIN1 treated old, and shRNA-scramble treated old ventricular myocytes, staining them with the membrane dye di-8-ANEPPS. T-tubule density and organization was found to decrease with aging with many tubules adopting a longitudinal orientation (see **Supplemental Figure 9**). Interestingly, myocytes isolated from old mice two weeks post-retro-orbital injection with AAV9-shRNA-mBIN1 displayed youthful t-tubule organization and density with fewer longitudinally arranged tubules. This effect was not observed in shRNA-scramble transduced old myocytes and thus it appears to occur as a direct result of the BIN1 knockdown. As the reviewer is likely aware, BIN1 is thought to facilitate t-tubule development and growth in neonatal mouse hearts. It is possible that upregulation of BIN1 in the aging heart is an adaptive effort to re-grow the t-tubule network that has become damaged with age. Why this enhanced BIN1 expression does not facilitate t-tubule network growth and maintenance is unclear at present but Bill Louch's group have recently reported that different BIN1 variants generate t-tubules with variable geometries. For example, BIN1+13+17 was shown to produce the widest diameter t-tubules of all the cardiac isoforms, while BIN1+13 produces lower density, shorter t-tubules. It is possible that the shRNA-mBIN1 changes the balance of BIN1 variants to facilitate restoration of the network.

In the revised manuscript we report:

Given the role of BIN1 in t-tubule development^{3,54}, we also examined the t-tubule network in isolated young, old, shRNA-mBIN1 treated old, and shRNA-scramble treated old ventricular myocytes, staining them with the membrane dye di-8-ANEPPS (Supplementary Fig. 9). T-tubule density and organization decreased with aging with many tubules adopting a longitudinal orientation. Interestingly, myocytes isolated from old mice two weeks post-retro-orbital injection with AAV9-shRNA-mBIN1 displayed youthful t-tubule organization and density with fewer longitudinally arranged tubules. This effect was not observed in shRNA-scramble transduced old myocytes and thus this rejuvenation of the t-tubule network appears to occur as a direct result of the BIN1 knockdown.

Supplementary Figure 9. Youthful t-tubule network organization is recovered in old myocytes upon BIN1 knockdown. **a**, representative images of t-tubules in cardiomyocytes labeled with di-8-ANEPPS isolated from young, old, old shRNA-scrmb and old shRNA-mBIN1 mice. Graphs representing **b**, t-tubule distance (μm) versus t-tubule intensity (AIU) data from the indicated ROIs for young and old and **c**, transduced shRNA-scrmb and shRNA-mBIN1 myocytes. Histograms representing **d**, t-tubule organization power (AIU) and **e**, periodicity (μm) for each group of JAX young ($N = 3$, $n = 24$), old ($N = 3$, $n = 26$), old shRNA-scrmb ($N = 3$, $n = 24$) and old shRNA-mBIN1 ($N = 3$, $n = 25$). Statistical analyses were performed using one-way ANOVAs and post-hoc multiple comparison tests. Note there was no significant difference in t-tubule organization when JAX and NIA mice were compared (see Supplementary Fig. 1m-n) but to maintain equal-sized groups for statistical comparison, data in this figure is from JAX young mice.

4. One of the most striking details is that old and young mice were NOT from the same source, despite both being Blk6 mice. While the young group originated from Jackson Laboratories directly, the old group was derived from the “National Institute on Aging Aged Rodent Colony”. Considering the unexpected findings detailed above and taking into account the fact that a lot of the aging studies have been designed with mice from a single source or even as a longitudinal endeavor, one might speculate to what degree these differences indeed originate from diverging

mouse strains within the “Black6” strain rather than from age “alone”. The authors should therefore be rather cautious with the interpretation of their data.

We thank the reviewer for raising this important point. We erroneously assumed that one C57Bl/6J mouse was like another but having researched we found that the NIA maintain C57Bl/6JN mice. With the "J" meaning that the foundation stock was originally obtained from The Jackson Laboratory and the "N" indicating that they were bred in the NIA colony. Accordingly, we performed new electrophysiology, SMLM immunolabelling of Cav1.2 and RyR2, endosome analysis, western analyses, and t-tubule di-8-ANEPPS staining to directly compared JAX sourced young mice with NIA sourced young mice (see **Supplementary Figure 1**). Our *in vivo* echocardiograms, doppler imaging, Ca²⁺ transient datasets, and transferrin recycling assay datasets contain only NIA sourced young mice and isolated myocytes. Results from JAX-sourced young mice were statistically compared to those from NIA-sourced young mice and in all cases, we found no significant difference between the two cohorts thus our conclusions are unchanged. However, this has undoubtedly enhanced the rigor of our study. In the revised manuscript we transparently identify datasets collected from JAX young mice in the figure legends. The supplemental methods of the revised manuscript also states:

"C57BL/6JN 3-5-month-old (referred to as “young”) and 21-25-month-old mice (referred to as “old”) were sourced from the NIA Aged Rodent Colony (Charles River Laboratories) unless otherwise stated. In some experiments young mice sourced from The Jackson Laboratory (JAX; C57BL/6J; Sacramento, CA, USA) were utilized. Data from JAX young and NIA young mice are statistically compared in Supplementary Figure 1 and no significant differences were detected in any of the measured parameters. Figure legends specify which data were obtained from JAX-sourced young mice."

Supplementary Figure 1. Comparison between young C57BL/6 mice sourced from the Jackson Laboratory (JAX; C57BL/6J) and the National Institute on Aging (NIA; C57BL/6JN). a-e, whole cell patch clamp electrophysiology data. a, plots showing the voltage dependence of I_{Ca} density for JAX young ($N = 6$, $n = 8$) and NIA young ($N = 3$, $n = 5$) male mice. b, voltage-dependence of the normalized conductance (G/G_{max}) fit with Boltzmann functions. c-e, Dot-plots showing fold change in I_{Ca} with ISO (c), peak I_{Ca} density (d), and $V_{1/2}$ of activation (e) for all groups. f and g, SMLM data showing dot-plots for mean $Ca_v1.2$ (f) and RyR2 (g) cluster area for JAX young ($Ca_v1.2$: control: $N = 3$, $n = 9$; ISO: $N = 3$, $n = 9$; RyR2: control: $N = 3$, $n = 17$; ISO: $N = 3$, $n = 11$) and NIA young ($Ca_v1.2$: control: $N = 3$, $n = 16$; ISO: $N = 3$, $n = 16$; RyR2: control: $N = 3$, $n = 15$; ISO: $N = 3$, $n = 15$). h and i, endosome analysis with dot-plots summarizing % colocalization between EEA1 and $Ca_v1.2$ (h), and EEA1 positive endosome areas (i) in JAX young (control: $N = 3$, $n = 16$; ISO: $N = 3$, $n = 16$) and NIA young (control: $N = 3$, $n = 17$; ISO: $N = 3$, $n = 16$) myocytes. j-l, western blot analysis. j, western blot of BIN1 expression in whole heart lysates from JAX young, NIA young, and old mice probed with 2F11 (top) and 99D (bottom). Total ponceau was used for normalization. Histograms showing normalized BIN1 levels relative to JAX young for western blots probed with 2F11 (k) ($N = 4$, average of 1-3 replicates) and 99D (l) ($N = 4$, average of 2-3 replicates). m and n, t-tubule analysis with dot-plots summarizing t-tubule organization (m) and periodicity (n) for JAX young ($N = 3$, $n = 24$) and NIA young ($N = 3$, $n = 12$) cardiomyocytes. Whole cell data from JAX and NIA young mice is merged in Figs. 1b-f. SMLM JAX young data from panels f and g is reproduced from Figs. 2b and 2d respectively. Endosomal JAX young data in panels h and i is reproduced from Figs. 4b and 4c respectively. T-tubule JAX young data in panels j and k is reproduced in Figs. S9d and S9e respectively. Statistical analyses

were performed using unpaired Student's t-test in c, m and n, two-way ANOVAs on d, e, f, g, h and i, and one-way ANOVAs on k and l with multiple comparison post-hoc tests.

Minor points:

1. *This reviewer could not find information on the temperature used for the life cell experiments and sample speed of the linescan images.*

Thank you for bringing this to our attention. The methods have now been updated with this information.

2. *This reviewer was rather surprised by the excessive Fluo4-AM concentration (10 μ M). For fast confocal imaging myocytes are usually loaded at concentrations < 1 μ M in the reviewer's lab. Is the Zeiss LSM 880 so inefficient in light collection?*

For these experiments we followed an established protocol published by Long-Sheng Song's group (<https://europepmc.org/article/pmc/3233356#FN4>) and used the Ca^{2+} indicator concentration recommended there⁶⁵.

3. *The statistical bases ought NOT to be the total number of cells but rather the number of animals because only the animals are statistically independent from one another. In this, the authors might also consider adjusting the sample sizes to be more equal and avoid sample size differences of 100%.*

In the revised manuscript we made efforts to have more equal sample sizes. All datasets include measurements obtained from >3 animals (*N*) and in our echocardiogram, doppler imaging, and western analyses statistics are performed comparing individual animals/heart lysates. We acknowledge the merits of comparing *Ns* over *ns* but would argue that standard practice in the field is to perform statistical analyses on cells, but to declare the number of animals they represent. Below we highlight just a few recently published articles that performed statistical analyses on cells. This small selection is representative of the field and features highly respected cardiovascular physiology-focused labs publishing in reputable peer-reviewed journals:

1. Hutchings DC, Madders GWP, Niort BC, Bode EF, Waddell CA, Woods LS, Dibb KM, Eisner DA, **Trafford AW**. Interaction of background Ca²⁺ influx, sarcoplasmic reticulum threshold and heart failure in determining propensity for Ca²⁺ waves in sheep heart. *J Physiol*. 2022 Jun;600(11):2637-2650. doi: 10.1113/JP282168. Epub 2022 Mar 20. PMID: 35233776; PMCID: PMC9310721.
2. Hutchings DC, Pearman CM, Madders GWP, Woods LS, Eisner DA, Dibb KM, **Trafford AW**. PDE5 Inhibition Suppresses Ventricular Arrhythmias by Reducing SR Ca²⁺ Content. *Circ Res*. 2021 Sep 3;129(6):650-665. doi: 10.1161/CIRCRESAHA.121.318473. Epub 2021 Jul 12. PMID: 34247494; PMCID: PMC8409902.
3. Liu G, Papa A, Katchman AN, Zakharov SI, Roybal D, Hennessey JA, Kushner J, Yang L, Chen BX, Kushnir A, Dangas K, Gygi SP, Pitt GS, Colecraft HM, Ben-Johny M, Kalocsay M, **Marx SO**. Mechanism of adrenergic Ca_v1.2 stimulation revealed by proximity proteomics. *Nature*. 2020 Jan;577(7792):695-700. doi: 10.1038/s41586-020-1947-z. Epub 2020 Jan 22. PMID: 31969708; PMCID: PMC7018383.
4. Papa A, Zakharov SI, Katchman AN, Kushner JS, Chen BX, Yang L, Liu G, Jimenez AS, Eisert RJ, Bradshaw GA, Dun W, Ali SR, Rodrigues A, Zhou K, Topkara V, Yang M, Morrow JP, Tsai EJ, Karlin A, Wan E, Kalocsay M, Pitt GS, Colecraft HM, Ben-Johny M, **Marx SO**. Rad regulation of Ca_v1.2 channels controls cardiac fight-or-flight response. *Nat Cardiovasc Res*. 2022 Nov;1(11):1022-1038. doi: 10.1038/s44161-022-00157-y. Epub 2022 Nov 14. PMID: 36424916; PMCID: PMC9681059.
5. Mira Hernandez J, Ko CY, Mandel AR, Shen EY, Baidar S, Christensen AR, Hellgren K, Morotti S, Martin JL, Hegyi B, Bossuyt J, **Bers DM**. Cardiac Protein Kinase D1 ablation alters the myocytes β-adrenergic response. *J Mol Cell Cardiol*. 2023 Jul;180:33-43. doi: 10.1016/j.yjmcc.2023.05.001. Epub 2023 May 4. PMID: 37149124.

Reviewer #3 (Remarks to the Author):

1. *ICa,L density and CaT amplitudes are elevated along with CaV1.2 and RyR2 superclustering in baseline conditions for old relative to young. With ISO, amplitudes are not different. By contrast, echocardiography shows EF (FS) is greater in the young. The authors in the Results section clearly attribute the reduced dynamic range for the ISO response to an elevated 'floor.'* Thus, it is misleading to repeatedly conclude that ISO responsiveness is reduced. For example, the representative CaT traces appear to exhibit accelerated decay, including for the aged ISO-treated

sweep. The authors should report an operational measure of decay kinetics and if faster with ISO in the aged group, a Western blot of PLN-Ser16 is a reasonable index of cellular ISO – responsiveness. Put another way, is the reduced dynamic range to ISO limited (or restricted) to the Ca channel complex in aging?

We agree that the ISO-stimulated amplitude of the currents and transients is similar in young and old cells. It was not our intention to mislead. The response to ISO is clearly diminished in old cells as can be appreciated by the fold-change in currents and transients in response to ISO shown in **Figure 1**, copied below for convenience. The reduced response to ISO also extends to the decay kinetics of the transients as shown in the revised manuscript **Supplemental Figure 3**. We have not measured PLB phosphorylation as we believe that investigation of age-dependent changes in phosphorylation in the aging heart would require a separate study. Please also see our response to Reviewer 1's similar comment #13 above.

Figure 1. β -AR stimulated augmentation of I_{Ca} and Ca^{2+} transients is diminished in aging. **a**, representative whole-cell currents elicited from young and old ventricular myocytes before (control; black) and during application of ISO (blue). **b**, dot-plots showing the fold change in peak current with ISO in young ($N = 9, n = 13$) and old ($N = 5, n = 11$) myocytes. **c**, dot-plots showing peak I_{Ca} density before and after ISO. **d**, plots showing the voltage dependence of I_{Ca} density for both groups before and after ISO. **e**, voltage-dependence of the normalized conductance (G/G_{max}) fit with Boltzmann functions and **f**, dot-plots showing the $V_{1/2}$ of activation for each group. **g**,

representative Ca^{2+} transients recorded before and after ISO from paced young ($N = 5$, $n = 12$) and old ($N = 3$, $n = 10$) myocytes. **h**, dot-plots showing the fold-increase in Ca^{2+} transient amplitude after ISO and **i**, Ca^{2+} transient amplitude before and after ISO. Unpaired Student's *t*-tests were performed on data sets displayed in **b** and **h**. Two-way ANOVAs with multiple comparison post-hoc tests were performed on data displayed in **c**, **f** and **i**.

Supplementary Figure 3. Time course of calcium transient response to ISO in young and old myocytes. a and c, dot-plots showing the fold-increase in Ca^{2+} transient amplitude after 1 min ($N = 5$, $n = 12$), 2 min ($N = 5$, $n = 12$), 3 min ($N = 5$, $n = 12$), and 5 min ($N = 5$, $n = 8$) of ISO in young myocytes (**a**) and 1 min ($N = 3$, $n = 10$), 2 min ($N = 3$, $n = 10$), 3 min ($N = 3$, $n = 10$), and 5 min ($N = 3$, $n = 7$) of ISO in old myocytes (**c**). **b and d**, dot-plots summarizing decay tau during the same time course in young (**b**) and old (**d**) myocytes. Statistical analyses were performed using one-way ANOVAs and post-hoc multiple comparison tests.

2. Related, and as mentioned in Discussion, other groups reported phosphorylation-dependent RyR2 clustering. Did the authors evaluate RyR2-phosphorylation in young, aged and aged with Bin1 knockdown?

Please see our response to the similar question posed by Reviewer 1 in their comment #11 above.

3. Bin1 is over-expressed in aged mice and t-tubule (but not crest) ICa_L shows an elevated floor, including loss of dynamism. What is the relative t-tubule to crest ICa_L in young versus aged?

Does *t*-tubule structure change in aging, in this mouse strain, and with *cBin1* knockdown? Can these interesting findings be attributed to a change in *t*-tubule architecture?

We thank the reviewer for raising this point. While we have not de-tubulated myocytes to examine I_{Ca} in the *t*-tubules versus crest, we have visualized $Ca_v1.2$ populations in the crest and *t*-tubule regions using SMLM and found no significant difference in the crest population with aging (see **Supplementary Figure 5**) while the *t*-tubule population of channels was increased.

Supplementary Figure 5. $Ca_v1.2$ clustering at the sarcolemmal crest is unaltered by aging. **a**, SMLM localization maps showing $Ca_v1.2$ channel localization and distribution in the sarcolemmal crest of young and old ventricular myocytes with or without ISO-stimulation. Yellow boxes indicate the location of the regions of interest magnified in the top right of each image. **b**, dot-plots summarizing the mean $Ca_v1.2$ cluster areas in JAX young (control: $N = 4$, $n = 11$; ISO: $N = 5$, $n = 8$) and old (control: $N = 3$, $n = 9$; ISO: $N = 3$, $n = 7$) myocytes. Statistical analyses were performed using a two-way ANOVA and post-hoc multiple comparison test.

Additional data collected for the resubmission suggests that *t*-tubular architecture may play a role. Please see our response to a similar question posed by Reviewer 2 in their comment #3 above.

4. What is the relationship between $I_{Ca,L}$ density and ISO responsiveness? Miriyala et al 2008, *Circ Res.* 102:e54–e64, published evidence for the concept of a functional reserve in the PKA regulation of $I_{Ca,L}$ whereby ISO (PKA) responsiveness is inversely related to $I_{Ca,L}$ density. For a plot of the data in Figures 1 and 5, do all conditions satisfy a single equation as in Miriyala 2008?

We thank the reviewer for bringing this paper to our attention. The idea that the accessible functional reserve revealed by PKA regulation of I_{Ca} is inversely related to I_{Ca} density could be interpreted as being in support of our model whereby PKA-phosphorylation of $Ca_v1.2$ channels enhances their recycling to augment their sarcolemmal expression and that this increase in the number of functional channels at least partially underlies the PKA-mediated enhancement of I_{Ca} during β -AR stimulation. If there is already a large number of channels in the sarcolemma generating a high current density, then there are likely fewer channels in the endosomes that can

be mobilized to the membrane upon PKA activation. We plotted the data in Figures 1 and 5 as requested and see a similar inverse relationship between the accessible functional reserve and current density. Those plots are included below:

5. For $I_{Ca,L}$, why does ISO cause a $V_{1/2}$ shift but not a fold-change in peak $I_{Ca,L}$ in cells from old mice? The dot plots of $V_{1/2}$ for activation should be added to Figure 1.

The suggested $V_{1/2}$ plots are included in the revised **Figure 1** with accompanying I-V plots. As the reviewer points out, old cells display a small (relative to the young cells) but significant depolarizing shift in their voltage dependence. This results in a larger driving force for Ca^{2+} entry with channels more likely to open at potentials further from E_{Ca} and should therefore result in a larger peak current. In line with that prediction, peak current does increase in old cells treated with ISO as seen in the fold change plot in **Figure 1b** where the fold-change in peak I_{Ca} with ISO is not 1 (unchanged), but 1.2 (indicating a 20 % increase in current with ISO). However, this fold change is significantly reduced compared to the young cell response.

Figure 1. β -AR stimulated augmentation of I_{Ca} and Ca^{2+} transients is diminished in aging. **a**, representative whole-cell currents elicited from young and old ventricular myocytes before (control; black) and during application of ISO (blue). **b**, dot-plots showing the fold change in peak current with ISO in young ($N = 9$, $n = 13$) and old ($N = 5$, $n = 11$) myocytes. **c**, dot-plots showing peak I_{Ca} density before and after ISO. **d**, plots showing the voltage dependence of I_{Ca} density for both groups before and after ISO. **e**, voltage-dependence of the normalized conductance (G/G_{max}) fit with Boltzmann functions and **f**, dot-plots showing the $V_{1/2}$ of activation for each group. **g**, representative Ca^{2+} transients recorded before and after ISO from paced young ($N = 5$, $n = 12$) and old ($N = 3$, $n = 10$) myocytes. **h**, dot-plots showing the fold-increase in Ca^{2+} transient amplitude after ISO and **i**, Ca^{2+} transient amplitude before and after ISO. Unpaired Student's t-tests were performed on data sets displayed in **b** and **h**. Two-way ANOVAs with multiple comparison post-hoc tests were performed on data displayed in **c**, **f** and **i**.

6. What does *Bin1* do to *t-tubules* in young vs old? Is the *t-tubule* organization in cardiomyocytes from young versus aged mice different?

Data included in the revised manuscript indicate that *t-tubule* organization is reduced in old myocytes compared to young. For a more detailed response please see our response to a similar question posed by Reviewer 2 in their comment #3 above.

7. Figure 6. FS and EF are calculated from the same M-mode measures. Panels f and g are redundant, choose just one. While EDV is reported, it is notable that this measure is a calculation from a linear measure – cubing terms and assumptions of shape can distort interpretation. Please

report the LV inner dimension in diastole and the wall thickness measures. Is there evidence for dilatation or wall hypertrophy from these direct measures?

In **Figure 6** we have retained the FS measurements and now omit the EF data. We have also removed the EDV, ESV, SV, and CO measurements from the Supplementary datasets as we agree with the reviewer that these measurements involve assumptions about shape that render them unreliable. The requested wall thickness measurements are now included in **Supplemental Figure 13**. We do not see any clear evidence of dilatation or hypertrophy with BIN1 knockdown but our LV inner dimension measurements from both diastole and systole appear benefit from this treatment with measurements becoming more youthful after 2-6 weeks of transduction.

Figure 6. BIN1 knockdown improves cardiac contractility in old mice. **a**, representative M-mode echocardiogram images from conscious young and old mice. **b**, summary dot-plots showing fractional shortening (FS) in young ($N = 10$) and old ($N = 30$) mice are shown. **c** and **d**, representative M-mode echocardiogram images from conscious old mice before and two, four and six weeks after RO-injection of shRNA-scrmb (**c**) and shRNA-mBIN1 (**d**). **e**, summary dot-plots for FS showing paired results before and two weeks after RO-injection for shRNA-scrmb ($N = 14$) and shRNA-mBIN1 ($N = 14$). **f**, summary dot-plots for FS showing paired results before and two, four and six weeks after RO-injection for shRNA-scrmb ($N = 5$) and shRNA-mBIN1 ($N = 5$). Statistical analysis was performed on data in **b** using unpaired Student's t-tests, on data in **e** using paired Student's t-tests, and on data in **f** using one-way ANOVAs.

CONSCIOUS Dimensions

Supplementary Figure 13. Left ventricular dimensions of young and old mice from conscious M-mode echocardiography. Summary dot-plots for conscious young ($N = 10$) and old ($N = 30$) mice, paired results before and two weeks after RO-injection of old mice with shRNA-scrambled ($N = 14$) and shRNA-mBIN1 ($N = 14$), and paired results before and after two, four and six weeks RO-injection of old mice with shRNA-scrambled ($N = 5$) and shRNA-mBIN1 ($N = 5$) for the following measurements are displayed: **a-c**, left ventricular anterior wall in diastole (LVAW;d), **d-f**, left ventricular anterior wall in systole (LVAW;s), **g-i**, left ventricular posterior wall in diastole (LVPW;d), **j-l**, left ventricular posterior wall in systole (LVPW;s), **m-o**, left ventricular inner diameter in diastole (LVID;d), **p-r**, left ventricular inner diameter in systole (LVID;s). Unpaired Student's t-tests were performed on data displayed in a, d, g, j, m and p. Paired Student's t-tests were performed on data displayed in b, e, h, k, n and q. One-way ANOVAs were performed on data displayed in c, f, i, l, o and r.

8. Figure 8. With $N=3$ and the scatter of that data along with inherent noise of quantification from Western blots, it is difficult to interpret the results. Either sample size should be increased or this Figure omitted.

We believe these data provide valuable information that links the cellular data to the *in vivo* data. The data were obtained from multiple technical and biological replicates as indicated below.

9. Discussion, p17 (lines 371-385) Paragraph on AD seems out of place and can be deleted.

We understand the reviewers point but politely disagree. We believe citing the literature on the role of BIN1 in AD is important as it provides context on endosomal traffic jams. This concept is

new to the cardiovascular field and we believe that providing some background on that will benefit the reader.

10. Discussion, p 18, line 403, “Echocardiography confirms the benefits of Bin1 knockdown ...” reiterates that echocardiography results reflect contractility loss in aging that is reversed by Bin1 knockdown. It is duly noted that mice at room temperature are under enhanced sympathetic tone (relative to humans). Under these conditions, extrapolating from Figures 1-4, we might have extended the common ISO levels in young and aged translating to no in vivo basal difference. This discordance is postulated to be found in myofilament Ca-sensitivity; however, Figure 8 is underpowered and so no conclusions can be reached. Nevertheless, if there was a significantly reduced myofilament Ca-sensitivity would this be reflected as elevated diastolic Ca?

We have not measured diastolic Ca²⁺ in this study as we did not use ratiometric Ca²⁺ dyes. Future studies should investigate this.

11. Discussion. The authors acknowledge the puzzling dichotomous impact of Bin1 levels in prior studies of heart failure in distinction to the present effects on aging. Kudos to the authors for acknowledging but not over-speculating on why these differences in results are found. In this vein, Fu (reference 18) showed increased spontaneous Ca-release with reduced Bin1 levels. Do the authors observe spontaneous Ca-release for any of the models used (young, old, +/- Bin1 knockdown)?

Unfortunately, when we recorded Ca²⁺ transients and observed events that looked like SCR, those recordings were discontinued, and a new cell was located to record from. We thus do not have a complete dataset from which to extract this information.

References cited

- 1 Chapuis, J. *et al.* Increased expression of BIN1 mediates Alzheimer genetic risk by modulating tau pathology. *Mol Psychiatry* **18**, 1225-1234, doi:10.1038/mp.2013.1 (2013).
- 2 Calafate, S., Flavin, W., Verstreken, P. & Moechars, D. Loss of Bin1 Promotes the Propagation of Tau Pathology. *Cell Rep* **17**, 931-940, doi:10.1016/j.celrep.2016.09.063 (2016).
- 3 Caldwell, J. L. *et al.* Dependence of Cardiac Transverse Tubules on the BAR Domain Protein Amphiphysin II (BIN-1). *Circulation Research* **115**, 986-996, doi:10.1161/circresaha.116.303448 (2014).
- 4 Hong, T. T. *et al.* BIN1 is reduced and Cav1.2 trafficking is impaired in human failing cardiomyocytes. *Heart Rhythm* **9**, 812-820, doi:10.1016/j.hrthm.2011.11.055 (2012).
- 5 Laury-Kleintop, L. D. *et al.* Cardiac-specific disruption of Bin1 in mice enables a model of stress- and age-associated dilated cardiomyopathy. *J Cell Biochem* **116**, 2541-2551, doi:10.1002/jcb.25198 (2015).
- 6 Li, J., Agvanyan, S., Zhou, K., Shaw, R. M. & Hong, T. Exogenous Cardiac Bridging Integrator 1 Benefits Mouse Hearts With Pre-existing Pressure Overload-Induced Heart Failure. *Front Physiol* **11**, 708, doi:10.3389/fphys.2020.00708 (2020).
- 7 Strait, J. B. & Lakatta, E. G. Aging-Associated Cardiovascular Changes and Their Relationship to Heart Failure. *Heart Failure Clinics* **8**, 143-164, doi:10.1016/j.hfc.2011.08.011 (2012).
- 8 Howlett, L. A. & Lancaster, M. K. Reduced cardiac response to the adrenergic system is a key limiting factor for physical capacity in old age. *Exp Gerontol* **150**, 111339, doi:10.1016/j.exger.2021.111339 (2021).
- 9 White, M. *et al.* Age-related changes in beta-adrenergic neuroeffector systems in the human heart. *Circulation* **90**, 1225-1238 (1994).
- 10 Xiao, R. P. *et al.* Age-associated reductions in cardiac beta(1)- and beta(2)-adrenergic responses without changes in inhibitory G proteins or receptor kinases. *Journal of Clinical Investigation* **101**, 1273-1282, doi:10.1172/Jci1335 (1998).
- 11 Hanyaloglu, A. C. & von Zastrow, M. Regulation of GPCRs by endocytic membrane trafficking and its potential implications. *Annual review of pharmacology and toxicology* **48**, 537-568, doi:10.1146/annurev.pharmtox.48.113006.094830 (2008).
- 12 Yudowski, G. A., Puthenveedu, M. A., Henry, A. G. & von Zastrow, M. Cargo-mediated regulation of a rapid Rab4-dependent recycling pathway. *Mol Biol Cell* **20**, 2774-2784, doi:10.1091/mbc.E08-08-0892 (2009).
- 13 Del Villar, S. G. *et al.* beta-Adrenergic control of sarcolemmal CaV1.2 abundance by small GTPase Rab proteins. *Proc Natl Acad Sci U S A* **118**, doi:10.1073/pnas.2017937118 (2021).
- 14 Galletta, B. J. & Cooper, J. A. Actin and endocytosis: mechanisms and phylogeny. *Current opinion in cell biology* **21**, 20-27, doi:10.1016/j.ceb.2009.01.006 (2009).
- 15 Hong, T. *et al.* Cardiac BIN1 folds T-tubule membrane, controlling ion flux and limiting arrhythmia. *Nature medicine* **20**, 624-632, doi:10.1038/nm.3543 (2014).

- 16 Hong, T. T. *et al.* BIN1 localizes the L-type calcium channel to cardiac T-tubules. *PLoS Biol* **8**, e1000312, doi:10.1371/journal.pbio.1000312 (2010).
- 17 Yada, H. *et al.* Dominant negative suppression of Rad leads to QT prolongation and causes ventricular arrhythmias via modulation of L-type Ca²⁺ channels in the heart. *Circ Res* **101**, 69-77, doi:10.1161/CIRCRESAHA.106.146399 (2007).
- 18 Taffet, G. E. & Tate, C. A. CaATPase content is lower in cardiac sarcoplasmic reticulum isolated from old rats. *Am J Physiol* **264**, H1609-1614, doi:10.1152/ajpheart.1993.264.5.H1609 (1993).
- 19 Schmidt, U. *et al.* In vivo gene transfer of parvalbumin improves diastolic function in aged rat hearts. *Cardiovascular research* **66**, 318-323, doi:S0008-6363(04)00285-8 [pii], 10.1016/j.cardiores.2004.06.028 (2005).
- 20 Lompre, A. M., Lambert, F., Lakatta, E. G. & Schwartz, K. Expression of sarcoplasmic reticulum Ca(2+)-ATPase and calsequestrin genes in rat heart during ontogenic development and aging. *Circ Res* **69**, 1380-1388, doi:10.1161/01.res.69.5.1380 (1991).
- 21 Cain, B. S. *et al.* Human SERCA2a levels correlate inversely with age in senescent human myocardium. *Journal of the American College of Cardiology* **32**, 458-467, doi:10.1016/s0735-1097(98)00233-2 (1998).
- 22 Cooper, L. L. *et al.* Redox modification of ryanodine receptors by mitochondria-derived reactive oxygen species contributes to aberrant Ca²⁺ handling in ageing rabbit hearts. *J Physiol* **591**, 5895-5911, doi:10.1113/jphysiol.2013.260521 (2013).
- 23 Kandilci, H. B., Tuncay, E., Zeydanli, E. N., Sozmen, N. N. & Turan, B. Age-related regulation of excitation-contraction coupling in rat heart. *J Physiol Biochem* **67**, 317-330, doi:10.1007/s13105-011-0077-3 (2011).
- 24 Slack, J. P. *et al.* The enhanced contractility of the phospholamban-deficient mouse heart persists with aging. *J Mol Cell Cardiol* **33**, 1031-1040 (2001).
- 25 Lim, C. C., Liao, R., Varma, N. & Apstein, C. S. Impaired lusitropy-frequency in the aging mouse: role of Ca(2+)-handling proteins and effects of isoproterenol. *Am J Physiol* **277**, H2083-2090, doi:10.1152/ajpheart.1999.277.5.H2083 (1999).
- 26 Isenberg, G., Borschke, B. & Rueckschloss, U. Ca²⁺ transients of cardiomyocytes from senescent mice peak late and decay slowly. *Cell Calcium* **34**, 271-280, doi:10.1016/s0143-4160(03)00121-0 (2003).
- 27 Mougnot, N. *et al.* Cardiac adenylyl cyclase overexpression precipitates and aggravates age-related myocardial dysfunction. *Cardiovascular research* **115**, 1778-1790, doi:10.1093/cvr/cvy306 (2019).
- 28 Ito, D. W. *et al.* beta-adrenergic-mediated dynamic augmentation of sarcolemmal CaV 1.2 clustering and co-operativity in ventricular myocytes. *J Physiol* **597**, 2139-2162, doi:10.1113/JP277283 (2019).
- 29 Fu, Y. *et al.* Isoproterenol Promotes Rapid Ryanodine Receptor Movement to Bridging Integrator 1 (BIN1)-Organized Dyads. *Circulation* **133**, 388-397, doi:10.1161/CIRCULATIONAHA.115.018535 (2016).
- 30 Asghari, P. *et al.* Cardiac ryanodine receptor distribution is dynamic and changed by auxiliary proteins and post-translational modification. *Elife* **9**, doi:10.7554/eLife.51602 (2020).

- 31 Josephson, I. R., Guia, A., Stern, M. D. & Lakatta, E. G. Alterations in properties of L-type Ca channels in aging rat heart. *J Mol Cell Cardiol* **34**, 297-308, doi:10.1006/jmcc.2001.1512 (2002).
- 32 Dibb, K. M., Rueckschloss, U., Eisner, D. A., Isenberg, G. & Trafford, A. W. Mechanisms underlying enhanced cardiac excitation-contraction coupling observed in the senescent sheep myocardium. *J Mol Cell Cardiol* **37**, 1171-1181, doi:10.1016/j.yjmcc.2004.09.005 (2004).
- 33 Walker, K. E., Lakatta, E. G. & Houser, S. R. Age associated changes in membrane currents in rat ventricular myocytes. *Cardiovascular research* **27**, 1968-1977, doi:10.1093/cvr/27.11.1968 (1993).
- 34 Grandy, S. A. & Howlett, S. E. Cardiac excitation-contraction coupling is altered in myocytes from aged male mice but not in cells from aged female mice. *Am J Physiol Heart Circ Physiol* **291**, H2362-2370, doi:10.1152/ajpheart.00070.2006 (2006).
- 35 Liu, S. J., Wyeth, R. P., Melchert, R. B. & Kennedy, R. H. Aging-associated changes in whole cell K(+) and L-type Ca(2+) currents in rat ventricular myocytes. *Am J Physiol Heart Circ Physiol* **279**, H889-900, doi:10.1152/ajpheart.2000.279.3.H889 (2000).
- 36 Kong, C. H. T. *et al.* The Effects of Aging on the Regulation of T-Tubular ICa by Caveolin in Mouse Ventricular Myocytes. *J Gerontol A Biol Sci Med Sci* **73**, 711-719, doi:10.1093/gerona/glx242 (2018).
- 37 Rueckschloss, U., Villmow, M. & Klockner, U. NADPH oxidase-derived superoxide impairs calcium transients and contraction in aged murine ventricular myocytes. *Exp Gerontol* **45**, 788-796, doi:10.1016/j.exger.2010.05.002 (2010).
- 38 Francis Stuart, S. D. *et al.* Age-related changes in cardiac electrophysiology and calcium handling in response to sympathetic nerve stimulation. *J Physiol* **596**, 3977-3991, doi:10.1113/JP276396 (2018).
- 39 Janczewski, A. M., Spurgeon, H. A. & Lakatta, E. G. Action potential prolongation in cardiac myocytes of old rats is an adaptation to sustain youthful intracellular Ca²⁺ regulation. *J Mol Cell Cardiol* **34**, 641-648, doi:10.1006/jmcc.2002.2004 (2002).
- 40 Lakatta, E. G. Deficient Neuroendocrine Regulation of the Cardiovascular-System with Advancing Age in Healthy Humans. *Circulation* **87**, 631-636 (1993).
- 41 Stratton, J. R. *et al.* Differences in cardiovascular responses to isoproterenol in relation to age and exercise training in healthy men. *Circulation* **86**, 504-512 (1992).
- 42 Davies, C. H., Ferrara, N. & Harding, S. E. Beta-adrenoceptor function changes with age of subject in myocytes from non-failing human ventricle. *Cardiovascular research* **31**, 152-156 (1996).
- 43 Lakatta, E. G., Gerstenblith, G., Angell, C. S., Shock, N. W. & Weisfeldt, M. L. Diminished inotropic response of aged myocardium to catecholamines. *Circ Res* **36**, 262-269 (1975).
- 44 Xiao, R. P. *et al.* Age-associated reductions in cardiac beta1- and beta2-adrenergic responses without changes in inhibitory G proteins or receptor kinases. *J Clin Invest* **101**, 1273-1282 (1998).
- 45 Xiao, R. P., Spurgeon, H. A., O'Connor, F. & Lakatta, E. G. Age-associated changes in beta-adrenergic modulation on rat cardiac excitation-contraction coupling. *J Clin Invest* **94**, 2051-2059 (1994).

- 46 Cerbai, E. *et al.* Beta-adrenoceptor subtypes in young and old rat ventricular myocytes: a combined patch-clamp and binding study. *Br J Pharmacol* **116**, 1835-1842 (1995).
- 47 Schulman, S. P. *et al.* Age-related decline in left ventricular filling at rest and exercise. *Am J Physiol* **263**, H1932-1938, doi:10.1152/ajpheart.1992.263.6.H1932 (1992).
- 48 Fleg, J. L. & Lakatta, E. G. in *Cardiovascular disease in the elderly* 21-64 (CRC Press, 2008).
- 49 Dai, D. F., Chen, T., Johnson, S. C., Szeto, H. & Rabinovitch, P. S. Cardiac aging: from molecular mechanisms to significance in human health and disease. *Antioxid Redox Signal* **16**, 1492-1526, doi:10.1089/ars.2011.4179 (2012).
- 50 Comelli, M. *et al.* Rhythm dynamics of the aging heart: an experimental study using conscious, restrained mice. *Am J Physiol Heart Circ Physiol* **319**, H893-H905, doi:10.1152/ajpheart.00379.2020 (2020).
- 51 Piantoni, C. *et al.* Age-Related Changes in Cardiac Autonomic Modulation and Heart Rate Variability in Mice. *Front Neurosci* **15**, 617698, doi:10.3389/fnins.2021.617698 (2021).
- 52 Kiper, C., Grimes, B., Van Zant, G. & Satin, J. Mouse strain determines cardiac growth potential. *PLoS One* **8**, e70512, doi:10.1371/journal.pone.0070512 (2013).
- 53 Taga, M. *et al.* BIN1 protein isoforms are differentially expressed in astrocytes, neurons, and microglia: neuronal and astrocyte BIN1 are implicated in tau pathology. *Mol Neurodegener* **15**, 44, doi:10.1186/s13024-020-00387-3 (2020).
- 54 Perdreau-Dahl, H. *et al.* BIN1, Myotubularin, and Dynamin-2 Coordinate T-Tubule Growth in Cardiomyocytes. *Circ Res* **132**, e188-e205, doi:10.1161/CIRCRESAHA.122.321732 (2023).
- 55 De Rossi, P. *et al.* Predominant expression of Alzheimer's disease-associated BIN1 in mature oligodendrocytes and localization to white matter tracts. *Mol Neurodegener* **11**, 59, doi:10.1186/s13024-016-0124-1 (2016).
- 56 Cataldo, A. M. *et al.* Endocytic pathway abnormalities precede amyloid beta deposition in sporadic Alzheimer's disease and Down syndrome: differential effects of APOE genotype and presenilin mutations. *Am J Pathol* **157**, 277-286, doi:10.1016/s0002-9440(10)64538-5 (2000).
- 57 Small, S. A., Simoes-Spassov, S., Mayeux, R. & Petsko, G. A. Endosomal Traffic Jams Represent a Pathogenic Hub and Therapeutic Target in Alzheimer's Disease. *Trends Neurosci* **40**, 592-602, doi:10.1016/j.tins.2017.08.003 (2017).
- 58 Lee, E. *et al.* Amphiphysin 2 (Bin1) and T-tubule biogenesis in muscle. *Science* **297**, 1193-1196, doi:10.1126/science.1071362 (2002).
- 59 De La Mata, A. *et al.* BIN1 Induces the Formation of T-Tubules and Adult-Like Ca(2+) Release Units in Developing Cardiomyocytes. *Stem cells (Dayton, Ohio)* **37**, 54-64, doi:10.1002/stem.2927 (2019).
- 60 Pant, S. *et al.* AMPH-1/Amphiphysin/Bin1 functions with RME-1/Ehd1 in endocytic recycling. *Nat Cell Biol* **11**, 1399-1410, doi:10.1038/ncb1986 (2009).
- 61 Lambert, E. *et al.* The Alzheimer susceptibility gene BIN1 induces isoform-dependent neurotoxicity through early endosome defects. *Acta Neuropathol Commun* **10**, 4, doi:10.1186/s40478-021-01285-5 (2022).

- 62 Tan, M. S., Yu, J. T. & Tan, L. Bridging integrator 1 (BIN1): form, function, and Alzheimer's disease. *Trends in molecular medicine* **19**, 594-603, doi:10.1016/j.molmed.2013.06.004 (2013).
- 63 Wang, H. F. *et al.* Bridging Integrator 1 (BIN1) Genotypes Mediate Alzheimer's Disease Risk by Altering Neuronal Degeneration. *J Alzheimers Dis* **52**, 179-190, doi:10.3233/JAD-150972 (2016).
- 64 Schurmann, B. *et al.* A novel role for the late-onset Alzheimer's disease (LOAD)-associated protein Bin1 in regulating postsynaptic trafficking and glutamatergic signaling. *Mol Psychiatry* **25**, 2000-2016, doi:10.1038/s41380-019-0407-3 (2020).
- 65 Guatimosim, S., Guatimosim, C. & Song, L. S. Imaging calcium sparks in cardiac myocytes. *Methods Mol Biol* **689**, 205-214, doi:10.1007/978-1-60761-950-5_12 (2011).

REVIEWER COMMENTS

Reviewer #1 (Remarks to the Author):

The author's have more than adequately addressed any concerns that I had in regards to their manuscript.

I am very excited to see the work they do in the future, I would encourage them to establish specific mechanisms of the (rapid) clustering and how the clustering modulates the function (including numerical modeling studies).

Reviewer #2 (Remarks to the Author):

This reviewer thanks the authors for their effort to address all reviewers' comment to increase the quality of their ms.

Despite this, this reviewer still has major reservations with respect to the following points.

My major criticisms are still not addressed appropriately.

BIN1-expression/Splicing. This reviewer does still not understand why the authors went through the difficult path of WB without demonstrating the splice-variant specificity of their ABs themselves (!). Why didn't they perform PCR analysis? This part is still rather weak, especially also because they only show "letter box" WB without giving transparent full data (the whole WB lane)? Here some of the contradictions in their own data e.g. in Figure 5. While the WB in a clearly depicts two distinct bands in young mice with the 2F11 AB, the "same" sample probed with the same AB in d shows only 1. While the analysis in a for the old mice displays a large blob of immunoreactivity, the same sample in d depicts two distinct bands. This appears rather curious if not disturbing. These data do not increase the confidence in the results presented in Suppl.Fig. 8. (i) Full-length lanes ought to be presented. The authors themselves state that the Abs used will NOT capture all splice variants. Why didn't they do the analysis with PCR?

The authors need to explain the different results when analysing T-tubular organisation with BIN1 Abs in fig. 5 (vertical stripes, no clear punctuated structures) and suppl.Fig. 9 using Di-8-ANEPPS, in which they clearly depict the expected punctuated appearance of the sliced T-tubules. In suppl.Fig.9 exemplified traces of the power spectra ought to be presented.

One of the major criticisms still unaddressed and remaining is the use of two different mouse strains rather than littermates. The animals of the BIN1-KO group are given a long time for developing possible compensatory mechanisms. The use of two different strains from two different sources (!) in my humble opinion is a KO-criterion. The "proper" way to address this point is the use of inducible KOs and the employment of proper littermates from the same strain under the same conditions. All the data

presented by the authors do NOT address the core of the problem including possible differences in gene regulation.

Application of 10 μm Fluo. The arguments of the authors are rather handwaving since I could (but I spare, a simple search can be performed by the authors themselves) present an even longer list of respectful papers in the highest journals employing much, much less Fluo loading. As the authors should be aware of, such high Fluo concentration will (i) massively increase the contribution of cytosolic fast calcium buffers (calcium shuttling), (ii) increase the Fluo concentrations in important calcium handling organelles such as SR and mitochondria.

Reviewer #3 (Remarks to the Author):

This is a responsive re-submission. I have no further comments. I recommend publication of this very interesting work.

Responses to the Reviewers Comments

We thank the reviewers for their time and effort in reviewing the first revision of our manuscript. **We were excited and gratified to read that Reviewers 1 and 3 considered our revision to be "a responsive re-submission" and that they are both recommending publication of the study with Reviewer 3 calling it "very interesting work" and Reviewer 1 noting that they are "very excited to see the work (we) do in the future"**. In response to Reviewer 2's remaining concerns we have diligently revised the manuscript a second time. We have performed quantitative RT-PCR to examine *Bin1* splice variant transcript expression in young and old cardiomyocytes. The results of those experiments agree with the analysis previously performed by Bill Louch's group¹ and by Robin Shaw's group² and show the transcriptional expression of four BIN1 splice variants in mouse cardiomyocytes. Furthermore, these results support and complement the extensive western analysis of BIN1 protein expression in young and old hearts. Below we provide specific point-by-point responses to that and each of Reviewer 2's other remaining concerns.

Reviewer 1's comments:

The author's have more than adequately addressed any concerns that I had in regards to their manuscript. I am very excited to see the work they do in the future, I would encourage them to establish specific mechanisms of the (rapid) clustering and how the clustering modulates the function (including numerical modeling studies).

We thank the reviewer for their kind comments on our work and agree with their suggestion for an interesting line of study that should be pursued in future work. We look forward to collaborations with our modeling colleagues to answer those important questions.

Reviewer 2's comments:

1) This reviewer thanks the authors for their effort to address all reviewers' comment to increase the quality of their ms.

We thank the reviewer for their kind comments on our work and their recognition that we undertook many new experiments to address the constructive reviews and we are confident those additions have improved the study and strengthened the conclusions of the manuscript.

2) BIN1-expression/Splicing. This reviewer does still not understand why the authors went through the difficult path of WB without demonstrating the splice-variant specificity of their ABs themselves.

The commercially available antibodies we use against BIN1 (i.e. 99D and 2F11) have been validated by multiple other labs for use on both heart cells and/or lysates¹⁻³ and are widely used in neuroscience, cancer, and cell biology labs^{4,5}. Most recently Bill Louch's lab extensively validated the 2F11 BIN1 antibody in their 2023 *Circulation Research* manuscript¹. This monoclonal antibody was originally generated by George Prendergast's lab and was raised against the N-terminal BAR domain of BIN1³. The second monoclonal BIN1 antibody we use in our westerns is clone 99D which was also generated by Prendergast's group and is specific for exon 17-containing isoforms^{3,6}. These antibodies have been well validated and found to not cross-react with other BAR-domain containing proteins including BIN2 or amphiphysin³. Perdreau-Dahl *et al* (Louch group) recently confirmed the identity of the four BIN1 bands that 2F11 reveals in heart lysates using custom splice variant specific antibodies that they commissioned from Genscript. These rigorous experiments confirmed that the 2F11 antibody detects the four splice variants known to be expressed in the heart, as reported by at least three independent groups including the present work^{1,2}. Those splice variants are BIN1, BIN1+13, BIN1+17, and BIN1+13+17. Perdreau-Dahl *et al* performed further quantitative RT-PCR that confirmed the presence of transcript for those specific variants in mouse cardiomyocytes. Prior to that, Robin Shaw's group had reported the presence of the same four splice variants in adult and neonatal mouse cardiomyocytes in his group's 2014 *Nature Medicine* article using western blots and RT-PCR². Thus, when we observed four bands with the expected molecular weights of the four known cardiac splice variants in both young and old heart lysates using the well validated 99D and 2F11 antibodies, based on the strength of the prior literature, we felt that it was not necessary to redo what had already been well demonstrated by two independent groups.

3) Why didn't they perform PCR analysis?

In response to Reviewer 2's suggestion, total RNA was extracted from ventricular myocytes isolated from young and old mouse hearts and reverse transcribed into cDNA. Quantitative RT-PCR was performed using a previously described protocol and primers against total *Bin1* transcript, the known cardiac isoforms *Bin1*, *Bin1+13*, *Bin1+17*, and *Bin1+13+17*¹, and *gapdh*. Using a pan-*Bin1* primer pair against the ubiquitous exon 2, we observed a 2.32 ± 0.53 -fold increase in total *Bin1* transcript expression in old cells compared to young. Isoform specific primer pairs confirmed the expression of all four established *Bin1* isoforms and similar fold changes were observed in each of the *Bin1* isoforms with aging (*Bin1*: 2.20 ± 0.58 ; *Bin1+13*: 1.88 ± 0.33 ; *Bin1+17*: 2.57 ± 0.60 ; *Bin1+13+17*: 2.03 ± 0.38). In both young and old cells mRNA for *Bin1* and *Bin1+17* was most abundantly expressed while *Bin1+13* and *Bin1+13+17* transcripts were scarcer. These data are included in Figure 5c of the second revision of the manuscript (relevant panels copied below). By confirming the transcriptional expression of the four known cardiac isoforms of *Bin1*, these results add further support to our western analyses.

Figure 5. BIN1 knockdown restores β -AR augmentation of Ca^{2+} transients and RyR2 clustering dynamics. **a**, western blot of BIN1 expression in whole heart lysates from young and old mice probed with 2F11 (*top*) and 99D (*bottom*). Short and long exposures are displayed. Total ponceau was used for normalization. **b**, histogram showing normalized BIN1 levels relative to young male for 2F11 and 99D ($N = 6$, average of 3-6 replicates). **c**, quantitative RT-PCR analysis of *pan-bin1*, *bin1*, *bin1+13*, *bin1+17* and *bin1+13+17* transcripts for young and old myocytes are displayed normalized to *gapdh* ($N = 3$ per group, samples from each N were ran in triplicate through three separate PCR runs). **d**, representative Airyscan images of young, old, and old transduced myocytes immunostained against BIN1. **e**, western blot of BIN1 expression in whole heart lysates from: young, old, shRNA-mBIN1 and shRNA-scrmb transduced mice probed with 2F11. Short and long exposures are displayed. Total ponceau was used for normalization. **f**, histogram showing normalized BIN1 levels relative to young ($N = 3$, average of 1-4 replicates).

4) Here some of the contradictions in their own data e.g. in Figure 5. While the WB in a clearly depicts two distinct bands in young mice with the 2F11 AB, the “same” sample probed with the same AB in d shows only 1. While the analysis in a for the old mice displays a large blob of immunoreactivity, the same sample in d depicts two distinct bands. These data do not increase the confidence in the results presented in Suppl.Fig. 8. (i).

We agree with the reviewer that in the prior form of Figure 5 panels a and d, those data may have appeared contradictory since two different exposures were used to image the two membranes. Figure 5a featured an image that was obtained using a long exposure (10 mins), while Figure 5d featured an image taken with a shorter exposure time (5 sec). In the re-revised manuscript, we now include each of the exposures so the reader can see that there are at least two BIN1 positive bands in the young lanes (see **Figure 5** excerpt in the response to concern 3 where the relevant panels are a and e). However, since BIN1 expression levels are so much higher in old heart lysates in comparison to young, to resolve the multiple bands in the young lanes, a longer exposure is required and that makes the old lanes appear to have that "large blob" of immunoreactivity to which the reviewer refers. Please note that datasets that were statistically compared to other datasets were always imaged at the same exposure.

5) The authors need to explain the different results when analysing T-tubular organisation with BIN1 Abs in fig. 5 (vertical stripes, no clear punctuated structures) and suppl.Fig. 9 using Di-8-ANEPPS, in which they clearly depict the expected punctuated appearance of the sliced T-tubules. In suppl.Fig.9 exemplified traces of the power spectra ought to be presented.

In the re-revised manuscript, we now include the requested exemplified traces in the relevant figure. In the re-revised manuscript the data presented in the original Supplemental Figure 9 now appears in Supplemental Figure 8. To clarify, the anti-BIN1 staining in Figure 5 indicates that BIN1 appears to adopt a predominantly striated, z-line localization. A similar z-line pattern of localization has been reported by at least two other independent groups^{1,7,8}. The most striking thing about the experiments in Figure 5 is the apparent redistribution of BIN1 from that youthful z-line staining pattern in young cardiomyocytes, to include a pronounced punctate staining pattern in old cardiomyocytes where we also see BIN1 expressional upregulation. Knockdown of BIN1 expression in old mice using shRNA-mBIN1 restores the youthful distribution.

In the di-8-ANEPPS stained cells in Supplemental Figure 8, we are looking at the plasma membrane of these cells. The reviewer is correct that there we can see t-tubular staining that sometimes appears punctate depending on the positioning of the confocal slice. In the revised text we have updated the results to clearly point out the z-line localization of BIN1 in young and in old shRNA-mBIN1 transduced cardiomyocytes stating:

"While BIN1 was observed to undergo upregulation with aging, those experiments did not provide any information as to the spatial localization of the excess protein. To examine that we immunostained fixed young and old ventricular myocytes against BIN1. These experiments revealed a mislocalization or redistribution of BIN1 in old myocytes where a vesicular, endosomal pattern of BIN1-staining was observed alongside the expected z-line population (Fig. 5d; *top*)."

Supplementary Figure 8. Youthful t-tubule network organization is recovered in old myocytes upon BIN1 knockdown. a-d, representative images of t-tubules in cardiomyocytes labeled with di-8-ANEPPS isolated from young (a), old (b), old shRNA-scrmb transduced (c), and old shRNA-mBIN1 transduced (d) mice. Plotted below are graphs representing the power spectrum obtained from a fast fourier transform (FFT) analysis performed on the representative cells. e-f: Histograms representing e, t-tubule organization power (AU) and f, periodicity (μm) for each group of young ($N = 3$, $n = 20$), old ($N = 3$, $n = 26$), old shRNA-scrmb ($N = 3$, $n = 24$) and old shRNA-mBIN1 ($N = 3$, $n = 25$). Young data was obtained from NIA-sourced young mouse myocytes.

6) One of the major criticisms still unaddressed and remaining is the use of two different mouse strains rather than littermates. The animals of the BIN1-KO group are given a long time for developing possible compensatory mechanisms. The use of two different strains from two different sources (!!) in my humble opinion is a KO-criterion. The “proper” way to address this point is the use of inducible KOs and the employment of proper littermates from the same strain under the same conditions. All the data presented by the authors do NOT address the core of the problem including possible differences in gene regulation.

We did not use BIN1KO mice as the reviewer states. Instead, we knockdown (not knockout) BIN1 in WT old mice using an AAV9-mediated shRNA approach. The transduction period for the shRNA treatments was as short as 2 weeks. We disagree with the reviewer that this constitutes a long time for developing possible compensatory mechanisms. In all cases we include the appropriate controls for the experiments (more details below) and indeed in many cases used same animal controls.

As clearly stated, multiple times in the MS, we use old mice that were transduced (via retro-orbital injection) with an AAV9 containing an shRNA-mBIN1 to knockdown but not knockout BIN1. For example, in line 257 we state:

“Since BIN1 became upregulated and mislocalized with aging with a more endosome-like pattern of expression, we hypothesized that knocking down BIN1 to young levels might resolve the endosomal traffic jams and restore β -AR responsivity to aging hearts. To test this, we transduced live mice with AAV9-GFP-mBIN1-shRNA (shRNA-mBIN1) or AAV9-GFP-scramble-shRNA (shRNA-scrmb; control) via retro-orbital injection. Two weeks post-injection, hearts were harvested, and successful shRNA-mediated knockdown of BIN1 was confirmed via both immunostaining and western blot analysis. BIN1 knockdown restored youthful localization of BIN1 to the t-tubules (Fig. 5d; *bottom*), and reduced BIN1 expression levels in old hearts to young heart levels (Fig. 5e-f). Control shRNA-scrmb transduced old hearts had similar BIN1 expression levels to non-transduced old hearts (Fig. 5e-f) and in all experiments, cells isolated from shRNA-scrmb transduced hearts maintained an old phenotype (Fig. 5d, 5g-n, Supplementary Fig. 7, and Supplementary Table 1).”

Additionally, in the Supplementary Material our detailed methods includes the following section:

“Adeno-associated virus (AAV9) short hairpin RNA (shRNA) for BIN1 knockdown AAV9-GFP-U6-m-Bin1-shRNA (shRNA-mBIN1) and AAV9-GFP-scrmb-shRNA (shRNA-scrmb) were purchased from Vector Biolabs (Malvern, PA, USA). For BIN1 knockdown, 22–25-month-old mice were anesthetized with vaporized isoflurane and retro-orbitally

injected with the shRNA-mBIN1 or shRNA-scramb at a concentration of 5×10^{11} vg/ml two weeks before experiments when animals were euthanized, and myocytes isolated, or hearts harvested for biochemistry.”

The appropriate control in these experiments is the parallel use of old mice (same source, same strain, same age) transduced with an AAV9-shRNA-scramble which we diligently examined in parallel with the shRNA-mBIN1 mice. Going a step further, in our echocardiography experiments we even use the same mice before and after BIN1 knockdown or scramble treatment (Figures 6 and 7, and Supplementary Figures 10-13) and show the paired comparisons with lines connecting the same mouse before and after the AAV9 treatment.

In the original submission we had performed our experiments on young mice on C57Bl/6 mice obtained from Jackson Laboratories (JAX) while the experiments on old mice were performed on C57Bl/6 mice obtained from the NIA (maintained by Charles River Laboratories). In our first resubmission we responded to justified and helpful reviewer concerns about this point and as stated in our first "Responses to the Reviewers":

"Accordingly, we performed new electrophysiology, SMLM immunolabelling of $Ca_v1.2$ and RyR2, endosome analysis, western analyses, and t-tubule di-8-ANEPPS staining to directly compared JAX sourced young mice with NIA sourced young mice (see **Supplementary Figure 1**). Our *in vivo* echocardiograms, doppler imaging, Ca^{2+} transient datasets, and transferrin recycling assay datasets contain only NIA sourced young mice and isolated myocytes. Results from JAX-sourced young mice were statistically compared to those from NIA-sourced young mice and in all cases, we found no significant difference between the two cohorts thus our conclusions are unchanged. However, this has undoubtedly enhanced the rigor of our study. In the revised manuscript we transparently identify datasets collected from JAX young mice in the figure legends."

We have now gone a step further and have omitted several datasets that were obtained from exclusively JAX-sourced young mice. Those datasets include proximity ligation assays between $Ca_v1.2$ and RyR2 originally featured in Figure 2. We have further omitted SMLM experiments examining the crest population of $Ca_v1.2$ in young and old myocytes originally featured in Supplementary Figure 5. SMLM of $Ca_v1.2$ and RyR2 in Figure 2 now only feature data from NIA-sourced young and old mice. Additionally, we removed all JAX-sourced young mouse data from the t-tubule analysis featured in the revised Supplementary Figure 8 and instead include only data obtained from NIA sourced mice. We also omitted recycling and late endosome data and associated figures and analysis, instead focusing on early endosomes. Importantly all omitted datasets were not critical to the conclusions of the paper.

Remaining datasets that still contain a portion of data obtained from JAX-sourced young mice were statistically compared to the same experiments performed on NIA-sourced young mice data. In all cases there was no statistically significant difference between data from JAX-sourced and NIA-sourced C57Bl/6 young mice as detailed in Supplementary Figure 1 and thus data were pooled. In Figure 3 dynamic TIRF experiments were performed on young myocytes from JAX-sourced mice and compared to old myocytes isolated from NIA-sourced mice. This data remains in the paper as it was a technically challenging dataset to obtain and analyze and the results are supported by a redundant set of experiments in which we performed transferrin receptor recycling assays on young and old myocytes isolated from exclusively NIA-sourced mice, obtaining similar conclusions (Supplementary Figure 5).

Supplementary Figure 1. Comparison between young C57Bl/6 mice sourced from the Jackson Laboratory (JAX) and the National Institute on Aging (NIA). **a-e**, whole cell patch clamp electrophysiology data. **a**, plots showing the voltage dependence of I_{Ca} density for JAX young ($N = 6$, $n = 8$) and NIA young ($N = 3$, $n = 5$) male mice. **b**, voltage-dependence of the normalized conductance (G/G_{max}) fit with Boltzmann functions. **c-e**, dot-plots showing fold change in I_{Ca} with ISO (**c**), peak I_{Ca} density (**d**), and $V_{1/2}$ of activation (**e**) for all groups. **f** and **g**, SMLM data showing dot-plots for mean $Ca_v1.2$ (**f**) and RyR2 (**g**) cluster areas for JAX young ($Ca_v1.2$: control: $N = 3$, $n = 9$; ISO: $N = 3$, $n = 9$; RyR2: control: $N = 3$, $n = 17$; ISO: $N = 3$, $n = 11$) and NIA young ($Ca_v1.2$: control: $N = 3$, $n = 16$; ISO: $N = 3$, $n = 16$; RyR2: control: $N = 3$, $n = 15$; ISO: $N = 3$, $n = 15$). **h** and **i**, endosome analysis with dot-plots summarizing % colocalization between EEA1 and $Ca_v1.2$ (**h**), and EEA1 positive endosome areas (**i**) in JAX young (control: $N = 3$, $n = 16$; ISO: $N = 3$, $n = 16$) and NIA young (control: $N = 3$, $n = 17$; ISO: $N = 3$, $n = 16$) myocytes. **j-l**, BIN1 western blot analysis. **j**, western blot of BIN1 expression in whole heart lysates from JAX young, NIA young, and old mice probed with 2F11 (*top*) and 99D (*bottom*). Total ponceau was used for normalization. Histograms showing normalized BIN1 levels relative to JAX young for western blots probed with 2F11 (**k**) ($N = 4$, average of 4-6 replicates) and 99D (**l**) ($N = 4$, average of 2-3

replicates). **m-o**, cTnI western blot analysis. **m**, western blot of whole heart lysates from JAX-sourced young and NIA-sourced young mice showing total (*top*) and pS23/24 (*bottom*) cTnI expression. Total ponceau was used for normalization. Histograms showing normalized total (**n**) and pS23/24/total (**o**) cTnI levels relative to JAX-sourced young ($N = 4$, average of 3-4 replicates). **p-r**, cMyBP-C western blot analysis. **p**, western blot of whole heart lysates from JAX-sourced young and NIA-sourced young mice showing total (*top*) and pS273 (*bottom*) cMyBP-C expression. Total ponceau was used for normalization. Histograms showing normalized total (**q**) and pS273/total (**r**) cMyBP-C levels relative to JAX-sourced young ($N = 4$, average of 4 replicates). Whole cell data from JAX-sourced and NIA-sourced young mice is merged in Figs. 1b-f. SMLM data from NIA-sourced young myocytes in panels f and g is reproduced from Figs. 2b and 2d respectively. Endosomal JAX-sourced young data in panels h and i is reproduced from Figs. 4b and 4c respectively. Statistical analyses were performed using unpaired Student's t-tests in c, n, o, q and r, two-way ANOVAs on d-i, and one-way ANOVAs on k and l with multiple comparison post-hoc tests.

7) Application of 10 μM Fluo. The arguments of the authors are rather handwaving since I could (but I spare, a simple search can be performed by the authors themselves) present an even longer list of respectful papers in the highest journals employing much, much less Fluo loading. As the authors should be aware of, such high Fluo concentration will (i) massively increase the contribution of cytosolic fast calcium buffers (calcium shuttling), (ii) increase the Fluo concentrations in important calcium handling organelles such as SR and mitochondria.

Please note that we removed Fluo-4 acquired data from the manuscript during the first revision to compare all cohorts (young, old, old shRNA-scrmb and old shRNA-mBIN1) utilizing the same Ca^{2+} indicator (Rhod-2). In response to this concern, we have added a sentence to the re-revised manuscript stating: "We cannot rule out that our 20-minute loading of cells with 10 μM Rhod2-AM at room temperature may have affected cytosolic Ca^{2+} buffering or Ca^{2+} handling by organelles such as the SR or mitochondria. However, the same loading protocol was used in all Ca^{2+} transient experiments so all cells across all cohorts experienced the same conditions".

Reviewer #3 (Remarks to the Author):

This is a responsive re-submission. I have no further comments. I recommend publication of this very interesting work.

We thank the reviewer for their encouraging words and kind comments about our manuscript.

References

- 1 Perdreau-Dahl, H. *et al.* BIN1, Myotubularin, and Dynamin-2 Coordinate T-Tubule Growth in Cardiomyocytes. *Circ Res* **132**, e188-e205, doi:10.1161/CIRCRESAHA.122.321732 (2023).
- 2 Hong, T. *et al.* Cardiac BIN1 folds T-tubule membrane, controlling ion flux and limiting arrhythmia. *Nature medicine* **20**, 624-632, doi:10.1038/nm.3543 (2014).
- 3 DuHadaway, J. B. *et al.* Immunohistochemical analysis of Bin1/Amphiphysin II in human tissues: diverse sites of nuclear expression and losses in prostate cancer. *J Cell Biochem* **88**, 635-642, doi:10.1002/jcb.10380 (2003).
- 4 De Rossi, P. *et al.* Predominant expression of Alzheimer's disease-associated BIN1 in mature oligodendrocytes and localization to white matter tracts. *Mol Neurodegener* **11**, 59, doi:10.1186/s13024-016-0124-1 (2016).
- 5 Chapuis, J. *et al.* Increased expression of BIN1 mediates Alzheimer genetic risk by modulating tau pathology. *Mol Psychiatry* **18**, 1225-1234, doi:10.1038/mp.2013.1 (2013).
- 6 Wechsler-Reya, R., Elliott, K., Herlyn, M. & Prendergast, G. C. The putative tumor suppressor BIN1 is a short-lived nuclear phosphoprotein, the localization of which is altered in malignant cells. *Cancer research* **57**, 3258-3263 (1997).
- 7 Zhou, J. *et al.* Phosphatidylinositol-4,5-Bisphosphate Binding to Amphiphysin-II Modulates T-Tubule Remodeling: Implications for Heart Failure. *Front Physiol* **12**, 782767, doi:10.3389/fphys.2021.782767 (2021).
- 8 Hong, T. T. *et al.* BIN1 localizes the L-type calcium channel to cardiac T-tubules. *PLoS Biol* **8**, e1000312, doi:10.1371/journal.pbio.1000312 (2010).